# Topological Generalization Bounds for Discrete-Time Stochastic Optimization Algorithms

**Rayna Andreeva**[*1]**, Benjamin Dupuis**[*2,3]**, Rik Sarkar**[1]**, Tolga Birdal**[†4]**, Umut Şimşekli**[†2,3]

[1] School of Informatics, University of Edinburgh, UK
[2] INRIA, France
[3] CNRS, Ecole Normale Supérieure PSL Research University, France
[4] Department of Computing, Imperial College London, UK

[*†] indicate equal contributions.

## Abstract

We present a novel set of rigorous and computationally efficient topology-based complexity notions that exhibit a strong correlation with the generalization gap in modern deep neural networks (DNNs). DNNs show remarkable generalization properties, yet the source of these capabilities remains elusive, defying the established statistical learning theory. Recent studies have revealed that properties of training trajectories can be indicative of generalization. Building on this insight, state-of-the-art methods have leveraged the topology of these trajectories, particularly their fractal dimension, to quantify generalization. Most existing works compute this quantity by assuming continuous- or infinite-time training dynamics, complicating the development of practical estimators capable of accurately predicting generalization without access to test data. In this paper, we respect the discrete-time nature of training trajectories and investigate the underlying topological quantities that can be amenable to topological data analysis tools. This leads to a new family of reliable topological complexity measures that provably bound the generalization error, eliminating the need for restrictive geometric assumptions. These measures are computationally friendly, enabling us to propose simple yet effective algorithms for computing generalization indices. Moreover, our flexible framework can be extended to different domains, tasks, and architectures. Our experimental results demonstrate that our new complexity measures correlate highly with generalization error in industry-standards architectures such as transformers and deep graph networks. Our approach consistently outperforms existing topological bounds across a wide range of datasets, models, and optimizers, highlighting the practical relevance and effectiveness of our complexity measures.

## 1 Introduction

Generalization, a hallmark of model efficacy, is one of the most fundamental attributes for certifying any machine learning model. Modern deep neural networks (DNN) display remarkable generalization abilities that defy the current wisdom of machine learning (ML) theory [85, 86]. The notion can be formalized through the *risk* minimization problem, which consists of minimizing the function:

$$\mathcal{R}(w) := \mathbb{E}_{z \sim \mu_z} \left[ \ell(w, z) \right], \tag{1}$$

38th Conference on Neural Information Processing Systems (NeurIPS 2024).

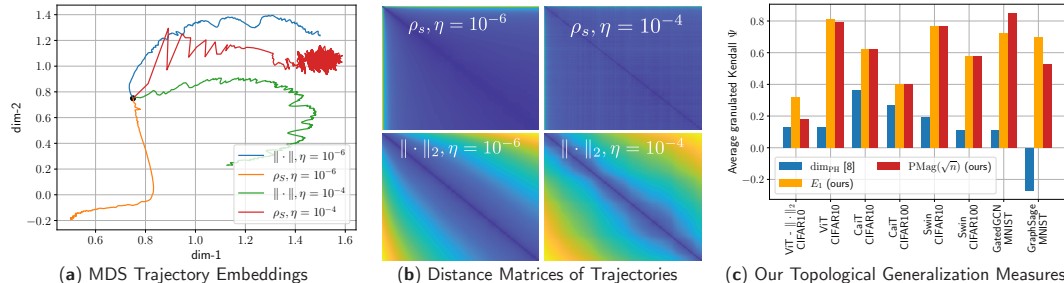

(a) MDS Trajectory Embeddings     (b) Distance Matrices of Trajectories     (c) Our Topological Generalization Measures

Figure 1: We devise a novel class of complexity measures that capture the topological properties of discrete training trajectories. These generalization bounds correlate highly with the test performance for a variety of deep networks, data domains and datasets. Figure shows different trajectories (**a**) embedded using multi-dimensional scaling based on the distance-matrices (**b**) computed using either the Euclidean distance ($\|\cdot\|_2$) between weights as in [10] or via the loss-induced pseudo-metric ($\rho_S$) as in [21]. (**c**) plots the *average granulated Kendall coefficients* for two of our generalization measures ($\boldsymbol{E}_\alpha$ and $\mathbf{PMag}(\sqrt{n})$) in comparison to the state-of-the-art persistent homology dimensions [10, 21] for a range of models, datasets, and domains, revealing significant gains and practical relevance.

where $z \in \mathcal{Z} := \mathcal{X} \times \mathcal{Y}$ denotes the data, distributed according to a probability distribution $\mu_z$ on the data space $\mathcal{Z}$. In practice, as $\mu_z$ is unknown, ML algorithms focus on minimizing the empirical risk,

$$\widehat{\mathcal{R}}_S(w) = \frac{1}{n}\sum_{i=1}^{n}\ell(w, z_i), \tag{2}$$

where $S := (z_1, \ldots, z_n) \sim \mu_z^{\otimes n} := \mu_z \otimes \cdots \otimes \mu_z$, which means that $(z_1, \ldots, z_n)$ are independent samples from $\mu_z$. In many applications, the minimization of (2) is achieved by discrete stochastic optimization algorithms, such as stochastic gradient descent (SGD) or the ADAM [40] method. Such algorithms generate a sequence of iterates in $\mathbb{R}^d$, denoted $\mathcal{W}_S := \{w_k\}_{k \geq 0}$, which depends on the data $S$, the initialization $w_0 \in \mathbb{R}^d$, and some additional randomness $U$, *e.g.*, the random batch indices in SGD. The *generalization error* characterizing the model's performance is then defined as:

$$G_S(w_k) := \mathcal{R}(w_k) - \widehat{\mathcal{R}}_S(w_k). \tag{3}$$

The empirical risk (2) typically has numerous local minima, which raises the question of how to characterize their generalization properties. Recently, training trajectories (*cf.*, Figure 1a) have been shown to be paramount to answer this question [84, 28]. Indeed, these trajectories can quantify the quality of a local minimum in a compact way, because they depend simultaneously on the algorithm, the hyperparameters, and the data, which is crucial for obtaining satisfactory bounds [37]. A wide family of trajectory-dependent bounds has been developed [84, 28, 50, 4, 36]. For instance, several results on stochastic gradient Langevin dynamics [57, 64, 49], continuous Langevin dynamics [57] and SGD [59] take into account the impact of the whole trajectory on the generalization error.

Parallel to these developments, several studies have brought to light the empirical links between topological properties of DNNs and their generalization performance [58, 52, 66, 70, 83], hereby making new connections with topological data analysis (TDA) tools [2]. These studies focus on the structural changes across the different layers of the network [51] or on the final trained network [66, 70, 83], and are almost exclusively empirical. This partially inspired a new class of trajectory-dependent bounds focusing on topological properties of the trajectories. In particular, recent studies [78, 21, 35, 22, 10, 3] have proposed to relate the generalization error to various kinds of intrinsic *fractal* dimensions [26, 53] that characterize the learning trajectory. Informally, these bounds provide the guarantee that with probability at least $1 - \zeta$, we have:[1]

$$\sup_{w \in \mathcal{W}_S} G_S(w) \lesssim \sqrt{\frac{\dim(\mathcal{W}_S) + \mathrm{IT} + \log(1/\zeta)}{n}}, \tag{4}$$

where $\dim(\mathcal{W}_S)$ denotes various equivalent fractal dimensions, in particular the persistent homology dimension (PH-dim) [10, 21] and the magnitude dimension [3]. The term $\mathrm{IT}$ is an information-theoretic quantity that takes different forms among different studies. Despite providing rigorous links

---

[1]We use $\lesssim$ in informal statements to indicate that absolute constants and/or small terms are missing.

between the topology of the trajectory and generalization, these bounds have major drawbacks. First and foremost, as noted in [75, 76, 13], fractal-trajectory bounds, such as Equation (4), do not apply to discrete-time algorithms. This creates a discrepancy between these theoretical results and the TDA-inspired methods to numerically evaluate them on commonly used discrete algorithms [10, 21, 3]. Additionally, existing bounds rely on intricate geometric assumptions, such as Ahlfors-regularity [78, 35] or geometric stability [21], that are not realistic in a practical, discrete setting.

Previous attempts were made to address this discretization issue. Specifically, under the assumption that the training dynamics possess a stationary measure $\mu_{w|S}^\infty$ for $T \to \infty$ ($T$ is the number of iterations), it was shown in [13] that with probability $1 - \zeta$ over $S \sim \mu_z^{\otimes n}$ and $w \sim \mu_{w|S}^\infty$, we have:

$$G_S(w) \lesssim \sqrt{\frac{\dim(\mu_{w|S}) + \mathrm{IT} + \log(1/\zeta)}{n}},$$ (5)

where $\dim(\mu_{w|S})$ corresponds to the fractal dimension of the measure $\mu_w$ (see [67] for formal definitions). While this was an important step, this bound only becomes practically relevant when the number of iterations grows to infinity, which is never attained in real-life experiments. Other attempts make use of so-called finite fractal dimensions [71] or fine properties of the Markov transition kernels associated with the dynamics [35]. However, these studies also rely on impractical assumptions and involve intricate quantities which make them not amenable to numerical evaluation.

Despite the theoretical limitations of existing topology-dependent generalization bounds, TDA-inspired tools have been developed to numerically estimate the proposed intrinsic dimensions in practical settings. Two particular methods have emerged and successfully demonstrate correlation with the generalization error, based on *persistent homology* [10, 21] (PH-dim) and *metric space magnitude* [3] (magnitude dimension); these two dimensions are equivalent for compact metric spaces [3]. Because of the limitations discussed above, existing theories do not account for these experiments, conducted with finite-time discrete algorithms. Moreover, existing empirical studies [10, 21, 3, 78] only consider very simple models and small (image) datasets. Because of their lack of theoretical foundations, it is not clear whether they could be extended to more practical setups.

**Contributions**. In this paper, we investigate the building blocks of PH and magnitude dimensions, in order to propose new topology-inspired generalization bounds that rigorously apply to widely used discrete-time stochastic optimization algorithms, and experimentally test our new topological complexities[2] on practically relevant DNN architectures. Our detailed contributions are as follows:

- We start by establishing the first theoretical links between generalization and a new kind of computationally thrifty topological complexity measure, the *$\alpha$-weighted lifetime sums* [73, 74].

- We propose and elaborate on another novel topological complexity, *positive magnitude* (**PMag**), a slightly modified version of magnitude [46, 55]. We rigorously link **PMag** with the generalization error, by relying on a new proof technique. Overall, our generalization bounds, rooted in TDA, admit the following generic form:

$$\sup_{w \in \mathcal{W}_S} G_S(w) \lesssim \sqrt{\frac{(\text{Topological complexity}) + \mathrm{IT} + \log(1/\zeta)}{n}}.$$

- We then provide a flexible computational implementation based upon dissimilarity measures between neural nets (Figure 1b), which enables quantifying generalization across different architectures and models, without the need for domain or problem-specific analysis as done in [39, 8].

- Unlike existing trajectory-based studies [10, 21] operating on small models, our experimental evaluation is extensive. We consider several vision transformers [20] and graph neural networks (GNN) [30] trained on multiple datasets spanning regular and irregular data domains (*cf.* Figure 1c). Our results demonstrate that the novel measures we introduce correlate strongly with the test performance across different architectures, hyperparameters and data modalities.

All the proofs of the main results are presented in the appendix, along with additional experiments. We will make our entire implementation publicly available under: https://github.com/rorondre/TDAGeneralization.

---

[2]Our term "topological complexity" should not be confused with the homonym topological invariant.

## 2   Technical Background

Our generalization indicators will be based upon $\alpha$-weighted lifetime sums and magnitude, capturing different topological features, as we shortly dicsuss below. Let $(X, \rho)$ be a finite pseudometric space.

$\alpha$**-weighted lifetime sums**. Persistent homology (PH) is an important concept in the analysis of geometric complexes [11]. We focus on the persistent homology of degree $0$ ($\mathrm{PH}^0$). Informally, it consists in tracking the "connected components" of a finite set at different scales. We provide in Sections A.3 and A.4 exact definitions of this notion. For simplicity, we present here an equivalent formulation of the $\alpha$-weighted lifetime sums based on minimum spanning trees (MST) [42, 73].

A tree over $X$ is a connected acyclic undirected graph (a set of edges) whose vertices are the points in $X$. Given an edge $e$ linking the points $a$ and $b$, we define its *cost* as $|e| := \rho(a, b)$. An MST $\mathcal{T}$ on $X$ is a tree minimizing the *total cost* $\sum_{e \in \mathcal{T}} |e|$. The $\alpha$-weighted lifetime sums $\boldsymbol{E}_\alpha^\rho$ are then written as:

$$\forall \alpha \geq 0, \ \boldsymbol{E}_\alpha^\rho(X) := \sum_{e \in \mathcal{T}} |e|^\alpha.$$

The celebrated *persistent homology dimension* (PH-dim) [1], of a compact pseudometric space $(A, \rho)$ is then defined as $\dim_{\mathrm{PH}}^\rho(A) = \inf_{\alpha \geq 0} \{ \exists C > 0, \forall Y \subset A \text{ finite}, \boldsymbol{E}_\alpha(Y) \leq C \}$. The PH-dim has been proven to be related to generalization error for different pseudometrics $\rho$ [10, 21].

**Magnitude**. Magnitude is a recently introduced topological invariant [46] which encodes many important invariants from geometric measure theory and integral geometry [46, 55, 56]. Magnitude can be interpreted as the effective number of distinct points in a space [46]. For $s > 0$, we define a *weighting* of the modified space $(X, s\rho)$ as a map $\beta : X \to \mathbb{R}$, such that $\forall a \in X, \ \sum_{b \in X} e^{-s\rho(a,b)} \beta(b) = 1$. Given such a weighting $\beta$, the magnitude function of $(X, s\rho)$ is defined as

$$\mathbf{Mag}^\rho(sX) := \sum_{a \in X} \beta(a). \tag{6}$$

The parameter $s > 0$ should be interpreted as a "scale" through which we look at the set $(X, \rho)$. We present in Appendix A.5 additional properties of this function. Note that magnitude is usually defined in metric spaces; we show in Appendix B.2 that we can seamlessly extend it to the pseudometric setting. Magnitude can be extended to (infinite) compact spaces [46, 55] and, as for PH, an intrinsic dimension, the *magnitude dimension*, can be defined from magnitude by the formula $\dim_{\mathrm{Mag}}^\rho(A) = \lim_{s \to \infty} \frac{\log \mathbf{Mag}(sA)}{\log(s)}$. It is known that $\dim_{\mathrm{PH}}^\rho$ and $\dim_{\mathrm{Mag}}^\rho$ coincide for compact metric spaces [56, 73, 3]. As a result, $\dim_{\mathrm{Mag}}^\rho$ has also been proposed as a topological generalization indicator [3].

**Total mutual information**. Prior intrinsic dimension-based studies relied on "mixing" assumptions ([78, Assumption H5], [10, Assumption H1], [76, 13]) or various mutual information terms [35, 21] to take into account the statistical dependence between the data and the training trajectory. Recently, a new framework was proposed in [22] to unify these approaches by proving data-dependent uniform generalization bounds using simpler and smaller information-theoretic (IT) terms. By leveraging these methods, we derive new generalization bounds involving the same IT terms for all our introduced topological complexities. More precisely, they take the form of a *total mutual information* between the data $S$ and the training trajectory $\mathcal{W}_S$. This term is denoted $\mathrm{I}_\infty(S, \mathcal{W}_S)$ and measures the dependence between $S$ and $\mathcal{W}$. We refer to Appendix A.1 and [35, 81] for exact definitions.

## 3   Main Theoretical Results

We now introduce our learning-theoretic setup (section 3.1) before delving into our main theoretical results in Sections 3.2 and 3.3.

### 3.1   Mathematical setup

**Random trajectories**. The primary goal of our theory is to prove uniform[3] generalization bounds over the training trajectory $\{w_k, \ k \geq 0\}$. We are mostly interested in the behavior near local minima of $\widehat{\mathcal{R}}_S$. To this end, we observe the trajectory between iterations $t_0$ and $T$, where $t_0 \in \mathbb{N}$ is the number of iterations before reaching (near) a local minimum and $T \geq t_0$ is the total number of iterations.

---

[3]By "uniform", we mean the worst error over a set; it should not be confused with uniform convergence.

Therefore, we consider the set $\mathcal{W}_{t_0 \to T} := \{w_i, \ t_0 \leq i \leq T\}$, which we call the *random trajectory*. Note that $\mathcal{W}_{t_0 \to T}$ is a *set*, *i.e.*, it does not contain any information about the time-dependence. Moreover, our setup allows the random times $t_0$ and $T$ to depend on the data $S$ through the choice of a stopping criterion as opposed to being fixed predetermined times.

**General Lipschitz conditions**. The topological quantities described in section 2, as well as the intrinsic dimensions introduced in prior works [78, 10, 3, 21, 22], require a notion of distance between parameters (in $\mathbb{R}^d$) to be computed. In the case of fractal-based generalization bounds, two cases have already been considered: the Euclidean distance [78] and the data-dependent pseudometric defined in [21]. In our work, we emphasize that both examples are particular cases of a more general family of pseudometrics on the parameter space $\mathbb{R}^d$. In order to fully characterize this family of pseudometrics, we define the data-dependent map $\boldsymbol{L}_S : \mathbb{R}^d \longrightarrow \mathbb{R}^n$ by $\boldsymbol{L}_S(w) = (\ell(w, z_1), \dots, \ell(w, z_n))$. To fit into our framework, a pseudometric must satisfy the following general Lipschitz condition.

**Definition 3.1** $((q, L, \rho)$-Lipschitz continuity). For any pseudo-metric $\rho$ on $\mathbb{R}^d$ and $q \geq 1$, we will say that $\ell$ is $(q, L, \rho)$-Lipschitz in $w$ when $\forall w, w' \in \mathbb{R}^d$, $\|\boldsymbol{L}_S(w) - \boldsymbol{L}_S(w')\|_q \leq L n^{1/q} \rho(w, w')$.

A wide variety of distances have been proposed to compare the weights of two DNNs [19]. The above condition restricts our analysis to a family of pseudometrics containing the following examples.

*Example* 3.2 (Data-dependent pseudometrics). For any $p \geq 1$, we define the pseudometrics $\rho_S^{(p)}(w, w') := n^{-1/p} \|\boldsymbol{L}_S(w) - \boldsymbol{L}_S(w')\|_p$. The case $\rho_S^{(1)}$ corresponds to the "data-dependent pseudometric" used in [21]; we will denote it $\rho_S := \rho_S^{(1)}$.

*Example* 3.3 (Euclidean distance). If $\ell(w, z)$ is $L$-Lipschitz continuous in $w$, *i.e.*, $|\ell(w, z) - \ell(w', z)| \leq L \|w - w'\|$ for all $z$, then $\ell$ is $(p, L, \|\cdot\|_2)$-Lipschitz continuous for every $p \geq 1$.

**Assumptions**. Given an $(q, L, \rho)$-Lipschitz continuous (pseudo-)metric, our approach relies only on a single assumption of a bounded loss function. For the case of the pseudometric $\rho_S$ (Example 3.2), this assumption is already made in [21, 22].

**Assumption 1.** We assume that the loss $\ell$ is bounded in $[0, B]$, with $B > 0$ a constant.

The boundedness of $\ell$ is classically assumed in the fractal / TDA literature [21, 35, 22]. In particular, this assumption is valid for the usual $0 - 1$ loss. In [21], it is shown that the proposed theory seems to be experimentally valid even for unbounded losses. Our experimental findings suggest that this observation also applies to our work.

### 3.2 Persistent homology related generalization bounds

In contrast to all existing fractal dimension-based bounds [78, 10, 13, 21], we propose new generalization bounds that apply to practical discrete stochastic optimizers with a finite number of iterations. To this end, our key idea involves replacing the intrinsic dimension with intermediary quantities that are used to compute them numerically. Following [10, 3], this points us towards the two quantities, $\boldsymbol{E}_\alpha$ and $\mathbf{Mag}$, defined in section 2. We are now ready to state the first generalization bound in terms of the $\alpha$-weighted lifetime sums, where we denote $\boldsymbol{E}_\alpha^\rho$ for $\boldsymbol{E}_\alpha^\rho(\mathcal{W}_{t_0 \to T})$.

**Theorem 3.4.** *Let $\rho$ be a pseudometric on $\mathbb{R}^d$. Supposes that Assumption 1 holds and that $\ell$ is $(q, L, \rho)$-Lipschitz, for $q \geq 1$. Then, for all $\alpha \in [0, 1]$, with probability at least $1 - \zeta$, we have:*

$$\sup_{t_0 \leq i \leq T} G_S(w_i) \leq 2B \sqrt{\frac{2 \log\left(1 + K_{n,\alpha} \boldsymbol{E}_\alpha^\rho\right)}{n}} + \frac{2B}{\sqrt{n}} + 3B \sqrt{\frac{\mathrm{I}_\infty(S, \mathcal{W}_{t_0 \to T}) + \log(1/\zeta)}{2n}},$$

*with $K_{n,\alpha} := 2 \left(2L\sqrt{n}/B\right)^\alpha$.*

The term $\mathrm{I}_\infty(S, \mathcal{W}_{t_0 \to T})$ is the total mutual information (MI) term that is defined in Sections 2 and A.1. It measures the statistical dependence between the random set $\mathcal{W}_{t_0 \to T}$ and the data $S \sim \mu_z^{\otimes n}$. Such MI terms appear in previous works related to fractal-based generalization bounds [78, 13, 21, 35]. Our proof technique, presented in Appendix B.5, makes use of a recently introduced PAC-Bayesian framework for random sets [22] to introduce this MI term. It is also shown in [22] that the MI term $\mathrm{I}_\infty(S, \mathcal{W}_{t_0 \to T})$ is tighter than those appearing in the aforementioned works.

We highlight the fact that Theorem 3.4 is fundamentally different from the persistent homology dimension (PH-dim) based bounds studied in [10, 21]. Indeed, while the growth of $\boldsymbol{E}_\alpha$ for increasing

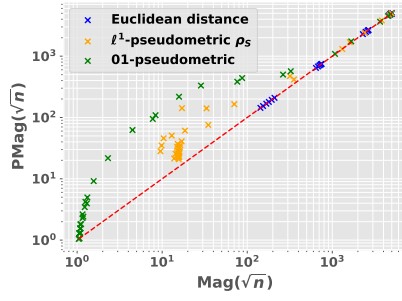
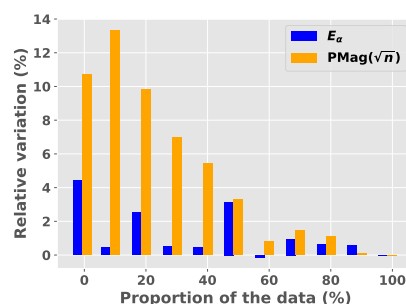

(a) Comparison of **Mag** and **PMag**.

(b) Relative variation of $E_1$ and **Mag**.

Figure 2: *Left:* Comparison of **Mag** and **PMag** (for $s = \sqrt{n}$), for different (pseudo)metrics (ViT on CIFAR10). *Right:* relative variation of the quantities $E_\alpha(\mathcal{W}_{t_0 \to T})$ and $\mathbf{Mag}(\sqrt{n}\mathcal{W}_{t_0 \to T})$, with respect to the proportion of the data used to estimated $\rho_S^{(1)}$ (ViT on CIFAR10).

finite subsets of the trajectory are used in [10] to estimate the PH-dim, it does not provide any formal link between the generalization error and the value of $E_\alpha$. Therefore, the above theorem could not be cast as a corollary of these previous studies. Another important characteristic of the above theorem (as well as the results of section 3.3) is to be non-asymptotic, *i.e.*, it is true for every $n \in \mathbb{N}^*$. This is an improvement over the fractal dimensions-based bounds presented in [78, 10, 21, 22].

### 3.3 Positive magnitude (PMag) and related generalization bounds

Recent preliminary experimental results displayed a correlation between the generalization error of DNNs and magnitude [3]. To provide a theoretical justification for this behavior, it would be tempting to mimic the proof of Theorem 3.4 and build on existing covering arguments. However, while lower bounds of magnitude in terms of covering numbers have been derived in [56], they appear to be impractical in our case. Another possibility would be to use the magnitude dimension bounds of [3]. Yet, this could not apply to our finite and discrete setting where the dimension is 0. Hence, we identify a new quantity, closely related to magnitude, while being more relevant to learning theory. With the notations of section 2, we fix a finite metric space $(X, \rho)$ and a weighting $\beta_s : X \longrightarrow \mathbb{R}$ of $(X, s\rho)$, where $s > 0$ is a "scale" parameter. We define the positive magnitude as

$$\forall s > 0, \ \mathbf{PMag}^\rho(sX) := \sum\nolimits_{a \in X} \beta_s(a)_+, \tag{7}$$

where $x_+ := \max(x, 0)$ denotes the positive part of $x$. To avoid harming the readability of the paper, we refer to Appendix B.3 for the extension of **PMag** to the pseudometric case. Based on a new theoretical approach, we prove that the positive magnitude can be used to upper bound the generalization error (see the proof in Appendix B.7). This leads to the following theorem:

**Theorem 3.5.** *Let $\rho$ be a pseudometric such that $(\mathcal{W}, \lambda\rho)$ admits a positive magnitude (according to Definition B.5) for every $\lambda > 0$. We assume that $\ell$ is $(q, L, \rho)$-Lipschitz continuous with $q \geq 1$. Then, for any $s > 0$, we have with probability at least $1 - \zeta$ that*

$$\sup_{t_0 \leq i \leq T} G_S(w_i) \leq \frac{2}{s} \log \mathbf{PMag}^\rho\left(Ls\mathcal{W}_{t_0 \to T}\right) + s\frac{B^2}{n} + 3B\sqrt{\frac{\mathrm{I}_\infty(S, \mathcal{W}_{t_0 \to T}) + \log(1/\zeta)}{2n}}.$$

We now present a quick sketch of the proof of Theorem 3.5, in order to highlight its key elements.

*Proof.* (Sketch) Let $\mathcal{W}$ be a data-dependent random compact set (*e.g.*, $\mathcal{W}_{t_0 \to T}$). The proof is based on two technical elements. The first is a framework recently proposed in [22] for uniform generalization bounds for random sets. These results give that with high probability we have a bound of the form:

$$\sup_{w \in \mathcal{W}} G_S(w) \lesssim \mathrm{Rad}(\ell, \mathcal{W}_{t_0 \to T}) + \sqrt{\frac{\mathrm{I}_\infty(S, \mathcal{W}_{t_0 \to T}) + \log(1/\zeta)}{n}},$$

where $\mathrm{Rad}(\ell, \mathcal{W}_{t_0 \to T})$ is the celebrated Rademacher complexity [5], whose definition is given in Appendix A.2. The second technical element is a new link between the Rademacher complexity of a compact set and its positive magnitude. This result is discussed in Appendix B.7. □

| MODEL-DATASET | VIT-CIFAR10 | | | | SWIN-CIFAR100 | | | | GRAPHSAGE-MNIST | | | | GATEDGCN-MNIST | | | |
|---|---|---|---|---|---|---|---|---|---|---|---|---|---|---|---|---|
| COMPL.-METRIC | $\psi_{LR}$ | $\psi_{BS}$ | $\Psi$ | $\tau$ | $\psi_{LR}$ | $\psi_{BS}$ | $\Psi$ | $\tau$ | $\psi_{LR}$ | $\psi_{BS}$ | $\Psi$ | $\tau$ | $\psi_{LR}$ | $\psi_{BS}$ | $\Psi$ | $\tau$ |
| $\dim_{PH}$ - $\rho_S$ [21] | 0.93 | -0.67 | 0.13 | 0.61 | 0.69 | -0.47 | 0.11 | 0.50 | -0.28 | -0.26 | -0.27 | -0.35 | 0.15 | 0.07 | 0.11 | -0.06 |
| $\mathbf{Mag}(\sqrt{n})$ - $\rho_S$ | 0.68 | 0.62 | 0.65 | 0.64 | 0.56 | 0.47 | 0.51 | 0.53 | 0.69 | 0.71 | **0.70** | **0.79** | 0.85 | 0.97 | **0.91** | **0.88** |
| $\mathbf{Mag}(0.01)$ - $\rho_S$ | 0.41 | 0.58 | 0.50 | 0.47 | 0.31 | 0.47 | 0.39 | 0.33 | 0.24 | 0.10 | 0.17 | 0.36 | 0.35 | 0.35 | 0.35 | 0.49 |
| $\mathbf{PMag}(\sqrt{n})$ - $\rho_S$ | 0.91 | 0.67 | 0.79 | 0.85 | 0.69 | 0.47 | 0.58 | 0.62 | 0.59 | 0.46 | 0.53 | 0.59 | 0.73 | 0.97 | 0.85 | 0.84 |
| $\mathbf{PMag}(0.01)$ - $\rho_S$ | 0.86 | 0.40 | 0.50 | 0.80 | 0.71 | 0.58 | **0.64** | **0.68** | 0.24 | 0.10 | 0.17 | 0.36 | 0.35 | 0.35 | 0.35 | 0.49 |
| $\boldsymbol{E}_\alpha$ - $\rho_S$ | 0.95 | 0.67 | **0.81** | **0.86** | 0.69 | 0.47 | 0.58 | 0.62 | 0.67 | 0.74 | **0.70** | 0.77 | 0.48 | 0.97 | 0.72 | 0.74 |
| $\dim_{PH}$ - $\|\cdot\|_2$ [10] | 0.93 | -0.67 | 0.13 | 0.61 | 0.69 | -0.47 | 0.34 | 0.51 | 0.32 | 0.81 | 0.56 | 0.51 | -0.12 | 0.70 | 0.29 | 0.33 |
| $\mathbf{Mag}(\sqrt{n})$ - $\|\cdot\|_2$ [3] | 0.95 | -0.59 | 0.13 | 0.73 | 0.71 | -0.57 | 0.07 | 0.53 | 0.75 | 0.77 | **0.76** | **0.61** | 0.77 | 0.76 | 0.77 | 0.52 |
| $\mathbf{Mag}(0.01)$ - $\|\cdot\|_2$ [3] | 0.95 | -0.60 | 0.17 | 0.72 | 0.69 | -0.44 | 0.12 | 0.53 | 0.75 | 0.74 | 0.74 | 0.60 | 0.77 | 0.42 | 0.60 | 0.47 |
| $\mathbf{PMag}(\sqrt{n})$ - $\|\cdot\|_2$ | 0.95 | -0.59 | 0.18 | 0.73 | 0.71 | -0.57 | 0.07 | 0.53 | 0.75 | 0.74 | 0.74 | 0.60 | 0.77 | 0.93 | **0.85** | 0.54 |
| $\mathbf{PMag}(0.01)$ - $\|\cdot\|_2$ | 0.55 | 0.71 | **0.63** | 0.58 | 0.64 | 0.51 | 0.58 | 0.46 | 0.75 | -0.05 | 0.35 | 0.51 | 0.60 | -0.47 | 0.06 | 0.26 |
| $\boldsymbol{E}_\alpha$ - $\|\cdot\|_2$ | 0.95 | -0.31 | 0.32 | **0.76** | 0.63 | 0.75 | **0.74** | **0.74** | 0.75 | 0.74 | 0.74 | 0.60 | 0.77 | 0.93 | 0.84 | **0.54** |
| $\dim_{PH}$ - 01 [21] | 0.95 | -0.20 | 0.37 | 0.72 | 0.64 | 0.04 | 0.34 | 0.51 | 0.0 | -0.13 | -0.07 | 0.0 | 0.14 | 0.00 | 0.07 | 0.00 |
| $\mathbf{Mag}(\sqrt{n})$ - 01 | 0.95 | 0.67 | **0.81** | 0.88 | 0.69 | 0.47 | **0.58** | **0.62** | 0.64 | 0.68 | **0.66** | **0.75** | 0.78 | 0.85 | **0.82** | **0.82** |
| $\mathbf{Mag}(0.01)$ - 01 | 0.84 | 0.33 | 0.59 | 0.75 | 0.61 | 0.27 | 0.44 | 0.50 | 0.13 | 0.11 | 0.12 | 0.26 | 0.10 | 0.10 | 0.10 | 0.25 |
| $\mathbf{PMag}(\sqrt{n})$ - 01 | 0.95 | 0.64 | 0.80 | **0.89** | 0.69 | 0.47 | **0.58** | **0.62** | 0.63 | 0.65 | 0.64 | 0.74 | 0.76 | 0.83 | 0.79 | 0.80 |
| $\mathbf{PMag}(0.01)$ - 01 | 0.84 | 0.36 | 0.60 | 0.76 | 0.65 | 0.49 | 0.57 | 0.54 | 0.13 | 0.11 | 0.12 | 0.26 | 0.10 | 0.10 | 0.10 | 0.25 |
| $\boldsymbol{E}_\alpha$ - 01 | 0.95 | 0.67 | **0.81** | 0.87 | 0.69 | 0.47 | **0.58** | 0.61 | 0.63 | 0.68 | **0.66** | 0.74 | 0.78 | 0.85 | **0.82** | **0.82** |

Table 1: Correlation coefficients associated with the different topological complexities.

The IT term ($I_\infty$) in the above result is the same as in Theorem 3.4. Given a fixed (finite) set $\mathcal{W}$ and a big enough $s$, we establish $\mathbf{Mag}(s\mathcal{W}) = \mathbf{PMag}(s\mathcal{W})$. Moreover, we present in Figure 2a an empirical comparison of $\mathbf{Mag}$ and $\mathbf{PMag}$, showing a small and almost monotonic relation between both quantities. Therefore, Theorem 3.5 may be seen as the first theoretical justification of the empirical relationship between magnitude and the generalization error observed in [3].

A natural choice for the scale $s$ would be $s \approx \sqrt{n}$, ensuring a convergence rate in $n^{-1/2}$. However, our empirical evaluations (see section 5, in particular, Table 1) revealed that small values of $s$ (we typically use $s = 10^{-2}$) can also provide good correlation with the generalization error. This could be explained by the fact that $\mathbf{PMag}(s\mathcal{W}) \to 1$ as $s \to 0$, *i.e.*, the bound may not diverge when $s \to 0$. For our topological complexities to be computationally efficient, we focus our experiments on fixed values of $s$ (in $\{\sqrt{n}, 10^{-2}\}$). We further analyze the sensitivity of part of our experiments to the value of $s$ in Appendix D.2.2. We will omit the trajectory and denote $\mathbf{Mag}(s)$ and $\mathbf{PMag}(s)$.

*Remark* 3.6. As it is explained in Appendix B.7, a key element in the proof of Theorem 3.5 is a newly discovered link between the celebrated Rademacher complexity [5] and positive magnitude. This is an additional contribution of our work, which might be of independent interest. Moreover, this relation extends beyond the case of finite sets and applies in particular to compact trajectories (or hypothesis sets) $\mathcal{W}$. We refer the reader to Remark B.15 and lemma B.16 for more details.

## 4 Computational Considerations

We now detail the numerical estimation of the topological complexities mentioned above.

**Computation of $\boldsymbol{E}_\alpha$.** We compute $\boldsymbol{E}_\alpha$ by using the `giotto-ph` library introduced in [65, 7]. This setup is inspired by PH frameworks used in [10, 21]. This technique uses the equivalent formulation of $\boldsymbol{E}_\alpha$ in terms of PH (see Appendix A.3 for details). Theorem 3.4, and its proof (presented in Appendix B.6) suggest that the relevant value of $\alpha$ is 1; similar to [10], this is what we used in our experiments.

**Computation of $\mathbf{Mag}$ and $\mathbf{PMag}$.** Different methods exist to evaluate magnitude [47]. We use the Krylov approximation method [72], which is based on pre-conditioned conjugate gradient iteration, implemented in the Python library `krypy.linsys.Cg` to solve for the magnitude weights. We then sum over the weights to compute $\mathbf{Mag}$, and sum over the positive weights to obtain $\mathbf{PMag}$.

**Distance matrix estimation..** Given a finite set (*i.e.*, a trajectory) $\mathcal{W} \subset \mathbb{R}^d$, the calculation of our topological complexities requires computing the *distance matrix* $D_\rho := (\rho(w, w'))_{w, w' \in \mathcal{W}}$. For large DNNs, this may become challenging. Depending on $\rho$, we propose the following solutions.

- Case 1: If $\rho$ is the Euclidean distance, for large DNNs (in our case for the transformer experiments) storing the whole trajectory is challenging. In that case, we use sparse random projections inspired by the Johnson-Lindenstrauss lemma [82] to project the trajectories onto a lower-dimensional

subspace. We use the implementation in `scikit-learn` [63] so that, with high probability, the relative variation of the distance matrices is at most $5\%$, see Appendix A.7 for details.

- Case 2: If $\rho$ is of the form $\rho_S^{(q)}$ as in Example 3.2, then the computation of $D_\rho$ requires the evaluation of the model on the entire dataset at each iteration, which becomes intractable for large DNNs. In [21, Figure 3], the authors show that the PH-dim based on the pseudometric $\rho_S = \rho_S^{(1)}$ is very robust to a random subsampling of a training dataset, *i.e.* when $\rho_S$ is replaced by $\rho_B$ with $B \subseteq S$ and $|B|/|S| \ll 1$. Figure 2b shows that $\boldsymbol{E}_\alpha$ and positive magnitude are also robust to this subsampling. We mainly used $|B|/|S| = 10\%$. We refer the reader to Appendix C.2 for details.

**Generalization error**. Our theory, like many trajectory-based studies [78, 10, 21, 3] predicts upper bounds on the worst-case generalization error over the trajectory $\mathcal{W}_{t_0 \to T}$. Yet, experiments in previous works mainly reported the error at the last iteration. To estimate the worst-case error in a computationally feasible way, we periodically evaluated the test risk between times $t_0$ and $T$ (every 100 iterations) and reported (`worst test risk - final train risk`) as the error in our experiments. This is consistent as we start the trajectory $\mathcal{W}_{t_0 \to T}$ from a weight $w_{t_0}$ already in a local minimum. Our main conclusions are still valid if the final generalization gap is used. This observation, which is to the best of our knowledge new, is briefly discussed in Appendix D.2.1.

## 5 Empirical Analysis

In what follows, we study our bounds on a variety of datasets and model architectures. We first explain the setup and the evaluation metrics before delving into the results and analysis.

**Setup**. Given a DNN and a dataset, we start from a pre-trained weight vector $w_{t_0}$, yielding high training accuracy on classification tasks. By varying the learning rate ($\eta$) and the batch size ($b$), we define a grid of $6 \times 6$ hyperparameters. For each pair ($\eta, b$), we compute the training trajectory $\mathcal{W}_{t_0 \to T}$ for $5 \times 10^3$ iterations. Unless specified, we use the ADAM optimizer [40]. Based on $\mathcal{W}_{t_0 \to T}$, we estimate distance matrices as described in section 4. For the sake of clarity, we focus on 3 relevant pseudometrics: (i) the Euclidean distance $\|\cdot\|_2$ as in [10], (ii) the data-dependent pseudometric $\rho_S$, used in [21, 3], and (iii) the 01-loss distance. For (ii), $\rho_S$ is computed based on the *surrogate* loss used in training (*e.g.*, the cross-entropy loss), while the reported generalization error is always based on *accuracy gap* (01-loss), which is of interest in most applications (see section 4). For the last one (iii) $\rho$ is defined as in Example 3.2, but with $\ell$ being the 01-loss; we call it 01-pseudometric and denote it by 01 in the tables. This last setup matches exactly our theoretical requirements.

In terms of DNN architectures, we focus on practically relevant models, while previous studies mainly considered small networks [10, 35, 21, 76]. We examine two different families of architectures. The first family consists of vision transformers (ViT [79], CaiT [80], Swin [48], see Table 2), each evaluated on both the CIFAR10 [44] and CIFAR100 [43] datasets. Moreover, we also tested our theory on graph neural networks (GNN) architectures, namely GatedGCN [12] and GraphSage [32] trained on the Super-pixel MNIST dataset [23]. To the best of our knowledge, this is the first time these kinds of topological complexities have been evaluated on transformers and GNNs. We ran the experiments on 18 NVIDIA 2080Ti (11 GB) GPUs.

**Granulated Kendall's coefficients.**. We assess the correlation between our complexities and the generalization error by using the granulated Kendall's coefficients (GKC) [37]. While the classical Kendall's coefficients (KC) [38] (denoted $\tau$) measures the correlation between two quantities, it may fail to capture their causal relationship. Instead, one "granulated" coefficient is defined in [37] for each hyperparameter (*i.e.*, $\psi_{\mathrm{LR}}$ for $\eta$ and $\psi_{\mathrm{BS}}$ for $b$); it measures the correlation when only this hyperparameter is varying. In Table 1, we report $\tau$, $\psi_{\mathrm{LR}}$ and $\psi_{\mathrm{BS}}$, and the averaged GKC, $\boldsymbol{\Psi} := (\psi_{\mathrm{LR}} + \psi_{\mathrm{BS}})/2$, for several models, datasets and topological complexities. In Figures 4a and 4b, we represent our topological complexities in the plane ($\psi_{\mathrm{BS}}, \psi_{\mathrm{LR}}$); the red square indicates the region of best correlation (the coefficients are in $[-1, 1]$, their sign is the sign of the correlation). It should be noted that a scaling of this constant $B$, coming from Assumption 1, would not impact the correlation between generalization and topological complexities that is observed in our experiments.

### 5.1 Analysis

As explained above, we focus our main experiments on the quantities $\boldsymbol{E}_1$, $\mathbf{Mag}(\sqrt{n})$, $\mathbf{PMag}(\sqrt{n})$, $\mathbf{Mag}(10^{-2})$ and $\mathbf{PMag}(10^{-2})$, each computed for the 3 pseudometrics discussed above ($\|\cdot\|_2$, $\rho_S$, 01). In the interest of comparison, we also compute the PH-dim (proposed in [10] for the $\|\cdot\|_2$ and in [21] for $\rho_S$), which is thus tested for the first time on transformers and GNNs.

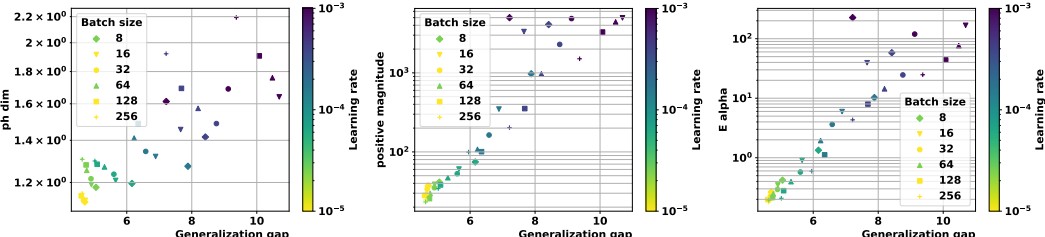

Figure 3: $\rho_S$-based complexity measures vs. generalization gap for a ViT trained on CIFAR10: $\dim_{PH}$ (*left*), $\mathbf{PMag}(\sqrt{n})$ (*middle*), and $\boldsymbol{E}_1$ (*right*).

**Performance on vision transformers**. We see in Table 1 and Figure 3 (additional graphical representation is given in Appendix D.1) that our proposed topological complexities consistently outperform the PH dimensions across several vision transformer models and datasets. This suggests that PH-dim, previously tested only on small architectures, is less scalable to industry-standards models with more parameters. Figure 4a, including all (`model`, `dataset`) pairs for the pseudometric $\rho_S$, reveals important observations. First, we notice that the GKC of our topological complexities are both positive and close to 1, indicating that they are indeed good measures of generalization. We note that for most models and datasets, $\dim_{PH}$ has a small or negative $\psi_{BS}$, indicating that it has less ability to explain generalization for varying batch-sizes. As it was observed in [21] for PH-dim, our complexities computed from the pseudometric $\rho_S$ correlate very well with the generalization gap while this gap is based on the 01 loss.

**Performance on GNNs**. An important aspect of our framework is the ability to seamlessly encapsulate different data domains. In particular, the possibility of using different pseudometrics can help define topological complexities that naturally take into account the internal symmetries of GNNs, without any model-specific analysis [39, 8]. The results of Table 1 and Figure 4a confirm that our proposed topological complexities outperform PH-dim and correlate strongly with the generalization error for GNNs. Additionally, it may be observed that $\mathbf{Mag}(\sqrt{n})$ performs significantly well for GNNs, and in particular better than $\mathbf{PMag}(\sqrt{n})$. This points us towards the idea that further theory would be desirable to formally relate magnitude to the generalization error in that case[4].

**Comparison of the topological complexities**. In Table 1 and Figures 3 and 4a, it can be seen that $\boldsymbol{E}_1$ and $\mathbf{PMag}(\sqrt{n})$ perform equally well for the image and graph experiments across multiple datasets, models, and data domains. We see in Table 1 that most topological complexities perform better with data-dependent metrics (*i.e.*, $\rho_S$ and 01) than with the Euclidean distance, for transformer-based experiments. This extends results obtained for PH-dim in [21], for smaller architectures. However, the poor performance of Euclidean-based complexities may also be partially caused by the projections applied to the Euclidean distance matrices to make them memory-wise computable (see section 4). This is a remaining limitation of our algorithms. On the other hand, the 01 and $\rho_S$ data-dependent pseudometrics seem to yield similar performance in all experiments.

**Ablations**. In Figure 4b, we reveal that changing the optimizer has little effect on the observed correlation (for the same model and dataset). Interestingly, we note that the PH-dim, computed with pseudometric $\rho_S$ and obtained from the SGD trajectories, exhibits high GKCs. This observation agrees with the results in [21]. Figure 3 further displays the typical behavior of several topological complexities for ViT and CIFAR10. In addition to the correlation of our proposed complexities being stronger than for the PH-dim, we observe that $\boldsymbol{E}_\alpha$ and $\mathbf{PMag}(\sqrt{n})$ seem to better correlate with the generalization gap for small learning rates. Finally, it is consistently observed in Table 1 and Figures 4a and 4b that using a relatively high value of the (positive) magnitude scale ($s = \sqrt{n}$) yields better correlations than small values ($s = 10^{-2}$). However, both cases still provide satisfying correlation, comforting the robustness of magnitude as a generalization indicator.

Due to limited space, we present all the correlation coefficient of one transformer model ViT for CIFAR10 and Swin for CIFAR100 in Table 1 as illustrative examples for each dataset. The remaining results appear in the Appendix, Tables 4, 6, 3 and 5, and they all follow a similar trend. Further empirical results and illustrations of this behavior are provided in Appendix D.

---

[4]We shall underline that, while $\mathbf{Mag}$ with the Euclidean distance was empirically proposed as a complexity measure in [3], a theoretical justification for $\mathbf{Mag}$ results in Table 1 is still missing for moderate values of $s$.

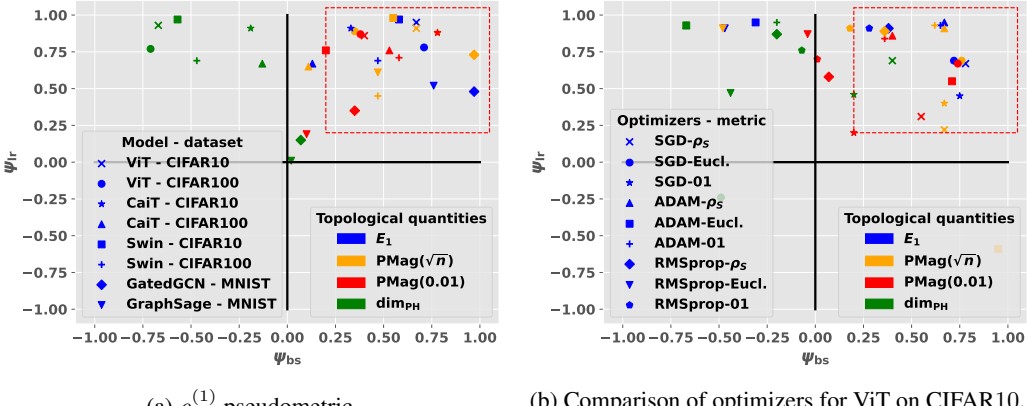

(a) $\rho_S^{(1)}$ pseudometric

(b) Comparison of optimizers for ViT on CIFAR10.

Figure 4: Granulated Kendall coefficients for several models, datasets and topological quantities. Note that our framework is directly applicable to graph networks.

## 6 Conclusion

In this paper, we proved novel generalization bounds based on several topological complexities coming from TDA, namely $\alpha$-weighted lifetime sums and a new variant of metric space magnitude, which we called positive magnitude. Compared to previous studies, we require fewer assumptions and operate in a discrete setting in which our proposed quantities are fully computable. Our algorithms are flexible enough to be seamlessly integrated with diverse data domains and tasks. These advantages of our framework allowed us to create a computationally cheap experimental setup, as close as possible to the theoretical setup. We thus provided a comprehensive suite of experiments with several industry-relevant architectures across vision transformers and graph neural networks, which have not been explored yet in this literature. We show that our proposed topological complexities correlate well with the generalization error, outperforming the previously studied intrinsic dimensions.

**Limitations & future work**. The main limitation of our theory is the lack of understanding of the IT terms, while they are still smaller than most prior works. The presence of this term renders our bounds not fully computable in practice. Indeed, we are not aware of existing techniques to evaluate the MI between random sets and the dimensionality of $\mathcal{W}_{t_0 \to T}$ (billions of parameters) could make a direct computation intractable. Nevertheless, our work focuses on improving the topological part of the existing bounds. Our main goal is to demonstrate a correlation with the generalization error rather than directly quantifying the generalization. Our experiments show that the introduced complexities are important and meaningful in addition to being amplified in the first part of the bound, as the dependence is explicit. Moreover, a better understanding of the behavior of positive magnitude for small values of the scale factor $s$ would be a necessary improvement. Regarding our experiments, a refinement of the estimation techniques of the topological complexities would be beneficial. Despite experimenting with practically relevant architectures, our future works also include scaling up our empirical analysis to include larger models and datasets, in particular large language models, which are still beyond the scope of this study.

## Acknowledgments

We would like to thank the reviewers of Neurips 2024, who helped to significantly improve this paper. R.A. is supported by the United Kingdom Research and Innovation (grant EP/S02431X/1), UKRI Centre for Doctoral Training in Biomedical AI at the University of Edinburgh, School of Informatics. U.Ş. is partially supported by the French government under management of Agence Nationale de la Recherche as part of the "Investissements d'avenir" program, reference ANR-19-P3IA-0001 (PRAIRIE 3IA Institute). B.D. and U.Ş. are partially supported by the European Research Council Starting Grant DYNASTY – 101039676. T.B. is partially supported by the Royal Society Research Grant RG\R1\241402. TB was supported by a UKRI Future Leaders Fellowship [grant number MR/Y018818/1].

**Broader impact**. Certifying generalization is key for safe and trusted AI systems, hence we believe that our study may have a positive societal impact.

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

## Appendix

We now provide additional technical details and proofs that are omitted from the paper, followed by experimental evidence complementing our main paper. We organize the appendix as follows:

- Appendix A presents additional technical background related to information theory, Rademacher complexity, and the various topological quantities that appear in our work.
- In Appendix B, we present the omitted proofs of all our theoretical results, as well as a few additional theoretical contributions.
- In Appendix C, we show the experimental details needed to reproduce our experiments.
- Finally, Appendix D is dedicated to additional empirical results.

## A  Additional technical background

### A.1  Information-theoretic quantities

The following definition is a precise definition of the total mutual information term that appears in our main theoretical results. The reader may consult [81, 35, 22] for further information on this notion.

**Definition A.1** (Total mutual information). Let $X$ and $Y$ be two random elements defined on a probability space $(\Omega, \mathcal{F}, \mathbb{P})$ (note that the codomains of $X$ and $Y$ may be distinct). We define the total mutual information between $X$ and $Y$ by the following formula:

$$\mathrm{I}_\infty(X, Y) = \log \left( \sup_A \frac{\mathbb{P}_{X,Y}(A)}{\mathbb{P}_X \otimes \mathbb{P}_Y(A)} \right).$$

Such a term has already been used in the fractal-based generalization literature [35, 22]. Other works used intricate variants of this total mutual information term [21, 10, 3, 13]. We stress the fact that our proposed bounds are simpler.

### A.2  Rademacher complexity

Rademacher complexity [6, 77] is a central tool in learning theory. As part of our theory uses this notion, we now provide its definition and introduce some notation.

**Definition A.2** (Rademacher complexity on a hypothesis set). Let us fix a dataset $S \in \mathcal{Z}^n$, a set $\mathcal{W} \subset \mathbb{R}^d$ and $\epsilon = (\epsilon_1, \ldots, \epsilon_n)$ some iid Rademacher random variable.[5] Whenever it is defined, we will call Rademacher complexity of $\ell$ over $\mathcal{W}$ the following quantity:

$$\mathrm{Rad}(\ell, \mathcal{W}, S) := \frac{1}{n} \mathbb{E}_\epsilon \left[ \sup_{w \in \mathcal{W}} \sum_{i=1}^n \epsilon_i \ell(w, z_i) \right].$$

Rademacher complexity has already been used in [21, Theorem 3.4] to relate the generalization error to the so-called data-dependent fractal dimension. Part of our theory is based on a recent extension of such arguments in the data-dependent setting [22].

### A.3  Persistent homology

The goal of this short subsection is to present a few notions of persistent homology, which is necessary for a better understanding of our contributions.

Persistent homology [24, 14, 11] is an important subfield of TDA, capable of providing myriad of new insights for analysing data by extracting meaningful topological features. It has demonstrated its usefulness in a very diverse set of applications from biology [60, 25], to materials science [34], finance [45], robotics [9], sensor networks [18] and a lot more [61]. The types of datasets which are amenable to this kind of analysis are finite metric spaces (known as point-cloud datasets), images, networks and also level-sets of functions. More recently, several studies have brought to light empirical links between persistent homology and DNNs [70, 17, 66]. In particular, recent studies have related the

---

[5]A Rademacher random variable is defined by $\mathbb{P}(\epsilon_i = 1) = \mathbb{P}(\epsilon_i = -1) = 1/2$.

worst-case generalization error to several concept of intrinsic dimensions defined through persistent homology [10, 21]. As mentioned in the introduction, our goal is to extend these last studies to more practical settings.

In general, persistent homology is defined for any degree $k \in \mathbb{N}$ (denoted $\mathrm{PH}^k$). Intuitively, $\mathrm{PH}^k$ keeps track of the number of "holes of dimension $k$" in a set when looked at different scales. However, in our work and as in [10, 21], we only use $\mathrm{PH}^0$, whose presentation is simpler. In this section, to avoid harming the readability of the paper, we only present a high-level introduction to $\mathrm{PH}^0$ that is sufficient to understand our work. The interested reader may consult [11, 15, 87] for a more in-depth introduction to persistent homology.

We first start by introducing briefly homology, which is a classical concept in algebraic topology. We only introduce the most essential concepts for understanding persistent homology. For a more detailed introduction, please consult [33].

**Definition A.3.** A simplicial complex is a set $K$ of finite sets closed under the subset relation: if $\sigma \in K$ and $\tau \subset \sigma$, then $\tau \in K$.

In the above definition, $\sigma$ is a simplex (plural simplices) and $\tau$ is a face of $\sigma$, its coface.

**Definition A.4.** An abstract simplicial complex $\mathcal{K}$ is a finite collection of simplices where a face of any simplex $\sigma \in \mathcal{K}$ is also a simplex in $\mathcal{K}$.

**Definition A.5.** A simplicial $k$-chain is the formal sum of $k$-simplices,

$$\sum_{i=1}^{N} = r_i \sigma_i, \tag{8}$$

where each $r_i \in R$, where $R$ is a fixed commutative ring with additive identity $0$ and multiplicative identity $1$, and $\sigma_i \in \mathcal{K}$.

$\mathcal{K}_k$ is the set of simplicial $k$-chains with addition over $R$, which is an $R$-module. Then, the set of all $k$-simplices of the complex $\mathcal{K}$ is a set of generators for $\mathcal{K}_k$. For each generator $\sigma$, the boundary of $\sigma$ is the sum of all $(k-1)$-faces of $\sigma$.

**Definition A.6.** The *boundary* of a $k$-simplex $\sigma = (x_0, \ldots, x_k)$ is the $(k-1)$-chain

$$\partial_k(\sigma) = \sum_{i=0}^{k} (-1)^i (x_0, \ldots, \hat{x}_i, \ldots, x_k), \tag{9}$$

where $(x_0, \ldots, \hat{x}_i, \ldots, x_k)$ is the $(k-1)$-simplex spanned by all vertices without $x_i$.

It is common that the coefficients for homology are considered to be restricted to $\mathbb{Z}_2$, which is the field with 2 elements, 0 and 1, where $1 + 1 = 0$. However, the theory extends to homoogy with coefficeints in any field (and since every field is a ring, the definitions in terms of rings are more general).

**Definition A.7.** A chain complex is a sequence of abelian groups $A_k$ with homomorphisms (called boundary maps) $\partial_k : A_k \to A_{k-1}$, such that $\partial_{k-1} \circ \partial_k = 0$ for all $k$.

We should note that when considering coefficients in $\mathbb{Z}_2$, a $k$-chain can be seen as a finite collection of $k$-simplices.

Introduce topological invariants: simplicial homology groups and Betti numbers.

**Definition A.8** (Simplicial Homology group). The $n$-th (simplicial) homology group of a finite simplicial complex $\mathcal{K}$ is

$$H_n = \ker \partial_n / \mathrm{im} \partial_{n+1}, \tag{10}$$

where $\ker$ and $\mathrm{im}$ are the kernel and image respectively of the boundary operator.

In order to define the simplicial complexes of use in TDA, we need to first understand what a nerve is.

**Definition A.9** (Nerve). A simplicial complex associated to a collection of sets is called a nerve. The sets are the vertices of the complex, and a simplex belongs to a complex iff its vertices have a non-empty intersection, $\mathrm{Nrv} = \{\alpha \subseteq S \mid \cap_{A \in \alpha} A \neq \emptyset\}$.

**Definition A.10** (Čech complex)**.** The Čech complex of $X$ for radius $r$ is $\text{Čech}_r(X) = \text{Nrv}\{B(x, r) \mid x \in X\}$, where $B(x, r)$ is the closed ball of radius $r \geq 0$, centered at $x$.

In other words, the Čech complex is the nerve of the ball neighbourhoods of a set of points $X \subseteq \mathbb{R}^n$. The Čech complex faithfully captures the topology of the space, but it is not computed in practice due to its high computational cost. Instead, a different complex called *Vietoris-Rips* (VR) is used due to ease of construction for higher dimensions. It can be shown that the VR complex is not always homotopy equivalent to the Čech complex, and therefore it can be seen as an approximation.

We first need to introduce the notion of a clique complex to explain what the VR is.

**Definition A.11** (Clique complex)**.** The *clique complex* for a graph $G = (V, E)$ consists of all cliques of $G$, which are all simplices $\alpha \subseteq V$ for which $E$ contains all edges of $\alpha$.

Now we have explicitly states all the necessary components in order to define the main complex used in TDA, the *Vietoris-Rips complex*.

**Definition A.12** (Vietoris-Rips complex)**.** The *Vietoris-Rips complex* of $X$ for radius $r$ is the clique complex of the 1-skeleton of the Čech complex of $X$ and $r$, $\text{Rips}_r(X) = \{\alpha \in X \mid ||u - v|| \leq 2r\}$ for all $u, v \in \alpha$.

Now that we have defined the most important complex in TDA, we proceed to explain how we can derive important topological information at multiple scales by introducing the concept of a filtration.

**Definition A.13.** Given a simplicial complex $\mathcal{K}$, a filtration is a totally ordered set of subcomplexes $\mathcal{K}^i$ of $\mathcal{K}$, indexed by nonnegative integers, such that for $i \leq j$, $\mathcal{K}^i \subseteq \mathcal{K}^j$.

**Definition A.14** (Filtered simplicial complex)**.** A simplicial complex, $\mathcal{K}$, together with a filtration (function $f : \mathcal{K} \to \mathbb{R}$ such that $f(\sigma) \leq f(\tau)$ whenever $\sigma$ is a face of $\tau$). The sublevel set at a value $r \in \mathbb{R}$ is $f^{-1}(-\infty, r]$, which is a subxomplex of $\mathcal{K}$. Let $r_0 < r_1 < \cdots < r_m$ be the values of the simplices, and $\mathcal{K}_i = f^{-1}(-\infty, r_i]$, then we call $\mathcal{K}_0 \subseteq \mathcal{K}_1 \subseteq \cdots \subseteq \mathcal{K}_m$ the *sublevel set filtration* of $f$.

When you start with a simplicial complex $\mathcal{K}$ and you filter it according to a filtration $f$, it is clear that the homology of $\mathcal{K}_r$ evolves as the radius $r$ increases. For example, new connected components can be formed, loops can appear or disapper, cavities can form. What persistent homology does, and where the importance of the filtering comes in is that now we have the tools to track the topological changes associated with the different stages of the filtering process, and to associate a lifetime to them (track when a topological feature has first appeared and at which stage of the filtration it will disappear). This essential topological information is recorded in a set of intervals known as barcodes, which can be represented as a multiset of points in $\mathbb{R}^2$, where the coordinates correspond to the birth and death points of each interval.

### A.3.1 Persistent homology of degree $0$ (alternative approach)

For the rest of the this section, we only focus on homology in dimension $0$, and provide an alternative and perhaps easier to understand interpretation. Please note that the following definition is a *simplified* and non-standard (though equivalent) definition of $\text{PH}^0$.

**Definition A.15** (Persistent homology of degree $0$ ($\text{PH}^0$))**.** Let $(X, \rho)$ be a finite metric space and $N$ its cardinality. For each time[6] $t \geq 0$, we construct an undirected graph $G_t$, whose edges are given by:

$$\forall x, y \in X, \ \{x, y\} \in G_t \iff \rho(x, y) \leq \delta.$$

There exists a finite set of times $0 < t_1 \cdots < t_k < +\infty$ such that the number of connected components in $G_{t_i}$ changes compared to $G_t$ for $t < t_i$. Let $c_i$ be the number of connected components in $G_{t_i}$. By convention we set $c_0 = N$ and $t_0 = 0$ and define $n_i := c_i - c_{i-1}$. $\text{PH}^0$ is then defined as the following multiset (the notation $\{\{\cdot\}\}$ denotes multisets):

$$\text{PH}^0 := \left\{\left\{\underbrace{t_1, \ldots, t_1}_{n_1 \text{ times}}, t_2, \ldots, \underbrace{t_k, \ldots, t_k}_{n_k \text{ times}}\right\}\right\}.$$

---

[6]We use the term time for the scalar $t$, as it is classically done in the study of persistent homology. Note that this has nothing to do with the number of iterations appearing in the rest of the paper.

*Remark* A.16 (Vietoris-Rips filtration). The above is a simplified high-level definition of $\mathrm{PH}^0$. More formally, the construction of the family of graphs $G_t$ corresponds to the construction of the so-called Vietoris-Rips filtration of $X$, of which we only kept the simplices of dimension $1$, see [11] for more details.

We now use $\mathrm{PH}^0$ to give the definitions of the quantities of interest in our work. The following is a definition of the quantity $\boldsymbol{E}_\alpha$ already mentioned in section 2, but seen through the lens of persistent homology. As it will be explained in Appendix A.4, these definitions are equivalent.

**Definition A.17** ($\alpha$-weighted lifetime sums). With the same notations as in Definition A.15, we define the $\alpha$-weighted lifetime sums as:

$$\forall \alpha \geq 0, \ \boldsymbol{E}_\alpha^\rho(X) := \sum_{t \in \mathrm{PH}^0} t^\alpha.$$

*Remark* A.18 ("birth" and "death" times). $\mathrm{PH}^k$ is usually defined as a multiset of birth and death times, tracking the appearance and disappearance of "holes of dimension $k$" during the construction of the Vietoris-Rips filtration of $X$. In the particular case of $\mathrm{PH}^0$, all birth times are $0$ and the times that we constructed correspond to the death times.

We end this section by giving the definition of the PH dimension, which has been shown to be theoretically and empirically related to the generalization error of neural networks in prior works [10, 21].

**Definition A.19** (Persistent homology dimension of degree 0). Given a compact metric space $(X, \rho)$, we define the PH dimension of degree $0$ by:

$$\dim_{\mathrm{PH}}^\rho(X) := \inf \left\{ \alpha \geq 0, \ \exists C > 0, \forall A \subseteq X \text{ finite}, \ \boldsymbol{E}_\alpha(A) \leq C \right\}.$$

It has been shown in [42, 73] that for any compact metric space, the PH dimension defined above is equal to the celebrated upper box-counting dimension [26, 54].

## A.4 Minimum spanning tree

The persistent homology dimension used in existing generalization bounds [10, 21] is closely related to another notion of intrinsic dimension, called minimum spanning tree (MST) dimension [42], in the sense that the PH and MST dimensions of bounded metric spaces are identical. The link between persistent homology and MST is even deeper than the equality between the induced dimensions, as noted by [73]. In this section, we define quantities related to MSTs which will play an important role in our proofs.

In this section let us fix a finite metric space $(X, \rho)$. Let us first specify our notations for trees. A tree $\mathcal{T}$ on $X$ is a connected undirected graph. We represent $\mathcal{T}$ by its set of edges, which are denoted $a \to b$ (or equivalently $b \to a$ as the graph is undirected). For an edge $e$ of the form $a \to b$, we define its length by $|e| = \rho(a, b)$.

**Definition A.20** (Minimum spanning tree). Let us define the cost of a tree by the sum of the length of its edges, *i.e.*,

$$\boldsymbol{E}_1^{\mathrm{MST}}(\mathcal{T}) := \sum_{e \in \mathcal{T}} |e|.$$

An MST of $X$ is defined as a tree with minimal cost. A consequence of the greedy algorithm to find such an MST [16] is that an MST $\mathcal{T}$ is also minimal for any of the following costs:

$$\boldsymbol{E}_\alpha^{\mathrm{MST}}(\mathcal{T}) := \sum_{e \in \mathcal{T}} |e|^\alpha,$$

with $\alpha \geq 0$.

Our interest in this notion comes from several results that are summed up in the following theorem. The reader can refer to [1, 73, 11] for more details.

**Theorem A.21** (Link between MST and persistent homology). *There is a bijection between the two following multisets:*

- *The multiset of the lifetimes in the persistent homology of degree $0$ of the Vietoris-Rips complex of $X$.*

- *The multiset of the length of the edges of an MST of $X$.*

*Therefore, if we fix some $\alpha \geq 0$, the weighted $\alpha$-sum associated to the persistent homology of degree $0$ of the Vietoris-Rips complex of $X$ is equal to the cost $\boldsymbol{E}_\alpha$ of an MST of $X$, ie:*

$$\boldsymbol{E}_\alpha^{\mathrm{MST}}(\mathcal{T}) = \boldsymbol{E}_\alpha(X).$$

*In all the following, we will use the notation $\boldsymbol{E}_\alpha$ to denote both quantities.*

## A.5 Magnitude

Let us restate formally a few standard definitions. The reader may refer to [46, 55, 56] for more details on the notions of magnitude, weighting, and positive definite metric spaces. In this section, we fix a finite *metric* space $(X, \rho)$. Some of the presented concepts will be later extended to pseudometric spaces in Appendix B.2.

As before, the *similarity matrix* [46] of $X$ is defined by $M(a, b) = e^{-\rho(a,b)}$, for $a, b \in X$. We now define weightings and magnitude of $X$, according to [46, Section 2.1].

**Definition A.22** (Weighting and magnitude). A weighting of $X$ is a function $\beta : X \longrightarrow \mathbb{R}$ such that

$$\forall a \in X, \ \sum_{b \in X} e^{-\rho(a,b)} \beta(b) = 1.$$

If such a weighting exists, the magnitude of $X$ is defined by:

$$\mathbf{Mag}(X) := \sum_{b \in X} \beta(b).$$

It is easily seen that this definition is independent of the choice of weighting $\beta$. When a weighting exists, we say that $X$ "has magnitude".

Based on such a definition, it is natural to inquire, whether such a weighting exists. This question has been studied by several authors [46, 55, 56]. This question appears to be related to the notion of positive definite space, which we now define, according to [46].

**Definition A.23** (Positive definite space). $X$ is positive definite if the similarity matrix $M$ is positive definite.

It is clear that positive definite spaces have magnitude. More interestingly, we have the following result, which ensures that most metric spaces considered in this study are positive definite.

**Theorem A.24** ([46, 55]). *Let $p \in [1, 2]$ and $d \geq 1$, every finite subset of $(\mathbb{R}^d, \|\cdot\|_p)$ is positive definite.*

## A.6 Covering and packing numbers

In this section, we fix a compact pseudometric space $(X, \rho)$ and give definitions of covering and packing numbers. These quantities have long been of primary interest in learning theory, in particular through the classical covering arguments for Rademacher complexity [77, 69]. More recently, limits of covering arguments have been leveraged by several authors to derive uniform generalization bounds in terms of fractal dimensions [78, 35, 13, 21, 22], which we aim to improve in this study.

For $x \in X$ and $r > 0$, we denote the closed ball centered at $x$ and or radius $r$ by $\bar{B}_r(x) := \{y \in X \ , \rho(x, y) \leq r\}$. We can now define covering and packing.

**Definition A.25** (Covering number). Let $\delta > 0$, the covering number $N_\delta^\rho(X)$ is the cardinality of a minimal set of points $N$ such that:

$$X \subseteq \bigcup_{x \in N} \bar{B}_\delta(x).$$

*Remark* A.26. There exist several conventions for the definition of such numbers [26, 53, 82], all of which are equivalent up to absolute constants and in particular induce the same fractal dimensions on $X$ (see [26]).

**Definition A.27** (Packing number). Let $\delta > 0$, the covering number $N_\delta^\rho(X)$ is the cardinality of a maximal set of disjoint closed balls with centers in $X$.

### A.7 About Johnson-Lindenstrauss lemma

In our implementation of Euclidean-based topological quantities, we use sparse random projections to project the weight vectors from $\mathbb{R}^d$ to a lower dimensional subspace. This is necessary because of memory constraints. Indeed, storing the full trajectory $\mathcal{W}_{t_0 \to T} \subset \mathbb{R}^d$ (in our experiments $T - t_0 = 5 \times 10^3$) can become intractable for large models.

Given a finite set of points $\mathcal{W} \subset \mathbb{R}^d$ and $\epsilon > 0$. Let $N \geq \mathcal{O}\left(\frac{\log |\mathcal{W}|}{\epsilon^2}\right)$, Johnson-Lindenstrauss lemma [82, 27] ensures the existence of a linear map $P : \mathbb{R}^d \longrightarrow \mathbb{R}^N$ such that:

$$\forall w, w' \in \mathcal{W}, \ (1 - \epsilon) \left\| w - w' \right\|^2 \leq \left\| Pw - Pw' \right\|^2 \leq (1 + \epsilon) \left\| w - w' \right\|^2 .$$

In practice, the linear maps suggested by this result can be obtained through subgaussian random projections [82, Section 9.3].

In our work, as the purpose of Johnson-Lindenstrauss embeddings is mainly memory optimization, we have to rely on sparse random projections. We use the implementation provided in `scikit-learn` [63]. More precisely, we used a relative variation $\epsilon$ of $5\%$.

Finally, it should be noted that these projection techniques were only used for the vision transformer experiments, as the GNNs that we used have a small enough number of parameters to avoid the use of random projections.

### A.8 A note on the connection to Topological Deep Learning

Topological deep learning (TDL) is a rapidly evolving field that uses topological features to understand and design deep learning models [62, 31]. Our topological complexity measures can be seen as a direction towards addressing the Open Problem 7 mentioned in [62] concerning the discovery of topological properties of internal representations that are linked to generalization.

## B Omitted proofs of the theoretical results

In this section, we present the proofs of our main theoretical contributions. We divide our proofs into two groups of subsections:

- Sections B.1,B.2 and B.3 focus on the extension (in a very natural way) of the quantities appearing in our bounds in pseudometric spaces. The main outcome of this analysis is the definition of positive magnitude in the pseudometric case. Note that Appendix B.1 is not a contribution of this paper. We placed it in this section to improve the readability of the paper.
- In sections B.4, B.5, B.6 and B.7, we present the proof of our main theoretical results.

Before, proving our main results, we define the notion of *metric identification*, which will be used in several of the following subsections. This is the same setting that was used in [21] to naturally extend the persistent homology dimension to pseudometric spaces.

**Definition B.1** (Metric identification)**.** Let $(X, \rho)$ be a pseudometric space. We can define an equivalence relation on $X$ by $a \sim b \iff \rho(a, b) = 0$. The associated quotient space, which is denoted $X/\sim$ is a metric space for the naturally induced metric, which we still denote $\rho$.[7] We will also use the canonical projection,

$$\pi : X \longrightarrow X/\sim.$$

These notations will be used throughout the text.

### B.1 Persistent homology and MST in pseudometric spaces

In this short subsection, we first restate results proven in [21], regarding persistent homology in pseudometric spaces. The main result is the following proposition, which has been proven inside the proof of [21, Lemma B.9].

---

[7]Indeed, if $a \sim b$, then we have $\forall c \in X, \ \rho(a, c) = \rho(b, c)$.

**Proposition B.2** ([22])**.** *Let $(X, \rho)$ be a finite pseudometric space and $\alpha \geq 0$, then we have:*

$$\boldsymbol{E}_\alpha(X) = \boldsymbol{E}_\alpha \left( {}^X\!/\!{\sim} \right)$$

*where the pseudometric $\rho$ (and its metric identification) have been omitted from the notation.*

Based on Theorem A.21, the above result is also true when $\boldsymbol{E}_\alpha$ represents the cost of a MST of $X$.

## B.2 Magnitude in pseudometric spaces

In this section, we fix $(X, \rho)$ a finite pseudometric space. We denote by ${}^X\!/\!{\sim}$ its metric identification and by $\pi : X \longrightarrow {}^X\!/\!{\sim}$ the canonical projection.

We directly extend Definition A.22 to the pseudometric case. In order for this definition to make sense in our context, we first need to verify that it provides a well-posed definition of magnitude. This follows from the following lemma.

**Lemma B.3.** *We assume that the finite pseudometric space $(X, \rho)$ has magnitude. Then magnitude is independent of the choice of weighting.*

*Proof.* The proof is straightforward and identical to the metric case. Let $\beta, \beta'$ be two weightings, we have:

$$\sum_{a \in X} \beta(a) = \sum_{a \in X} \sum_{b \in X} e^{-\rho(a,b)} \beta'(b)\beta(a) = \sum_{b \in X} \beta'(b) \sum_{a \in X} e^{-\rho(a,b)} \beta(a) = \sum_{b \in X} \beta'(b).$$

$\square$

In the following theorem, we show that magnitude is invariant through metric identification.

**Theorem B.4** (Invariance of magnitude through metric identification)**.** *$X$ has magnitude if and only if ${}^X\!/\!{\sim}$ has magnitude, in which case we have:*

$$\mathbf{Mag}(X) = \mathbf{Mag} \left( {}^X\!/\!{\sim} \right).$$

*Proof.* We decompose $X$ into equivalence classes as:

$$X = \coprod_{\bar{a} \in {}^X\!/\!{\sim}} \bar{a} =: \coprod_{i \in I} \bar{a}_i,$$

where $\coprod$ denotes disjoint union and the points $(a_i)_{i \in I} \in X^I$ represent each equivalence class. We denote by $\bar{a}$ the equivalence class of $a \in X$.

Let $\beta : X \longrightarrow \mathbb{R}$ be any function. We have:

$$\forall a \in X, \ \sum_{b \in X} e^{-\rho(a,b)} \beta(b) = \sum_{i \in I} e^{-\rho(\bar{a}, \bar{a}_i)} \sum_{b \in \bar{a}_i} \beta(b). \tag{11}$$

$\Longrightarrow$ : If $X$ has magnitude, then we take $\beta$ to be a weighting of $X$, we define:

$$\forall \bar{a} \in {}^X\!/\!{\sim}, \ \bar{\beta}(\bar{a}) := \sum_{b \in \bar{a}} \beta(b).$$

By Equation (11), $\bar{\beta}$ is a weighting of ${}^X\!/\!{\sim}$.

$\Longleftarrow$ : if $\bar{\beta}$ is a weighting of ${}^X\!/\!{\sim}$, then we define:

$$\forall a \in X, \ \beta(a) := \frac{1}{|\bar{a}|} \bar{\beta}(\bar{a}),$$

where $|\bar{a}|$ denotes the cardinality of $\bar{a}$. By Equation (11), $\beta$ is a weighting of $X$. $\square$

## B.3 Definition of positive magnitude in the pseudometric case

Let us extend our new notion of *positive magnitude* in finite pseudometric spaces. This is a rather complicated task. Indeed we need to ensure that the positive magnitude is independent of the choice of weighting, which is not true in general. For this reason, we restrict our definition to pseudometric spaces whose metric identification is positive definite and we choose one particular weighting.

**Definition B.5** (Positive magnitude in finite pseudometric spaces). Let $(X, \rho)$ be a finite pseudometric space whose metric identification $X/\sim$ is positive definite. Let $\bar{\beta} : X/\sim \longrightarrow \mathbb{R}$ be a weighting of $X/\sim$, then we define the positive magnitude of $X$, denoted $\mathbf{PMag}$, by:

$$\mathbf{PMag}(X) = \sum_{\bar{x} \in X/\sim} \bar{\beta}(\bar{x})_+,$$

where $x_+ := \max(x, 0)$ denotes the positive part of $x$. We will say that $X$ admits a positive magnitude if its metric identification $X/\sim$ is positive definite.

Note that $X/\sim$ admits a unique weighting because it is positive definite. However, $X$ still admits several weightings in general. The above definition ensures that the definition of positive magnitude is independent of any choice of weighting. For the need of our proofs, we will need to introduce weightings in pseudometric spaces, whose sums of positive parts yield the positive magnitude. This is possible by using the following definition, which corresponds to a "good" choice of weighting in finite pseudometric spaces.

**Definition B.6** (Canonical weighting). Let $(X, \rho)$ be a finite pseudometric space whose metric identification $X/\sim$ is positive definite. Let $\bar{\beta} : X/\sim \longrightarrow \mathbb{R}$ be a weighting of $X/\sim$, we define the *canonical weighting* $\beta^0 : X \longrightarrow \mathbb{R}$ on $X$ by:

$$\forall a \in X, \ \beta^0(a) := \frac{1}{|\pi(a)|} \bar{\beta}(\pi(a)),$$

where $\pi : X \longrightarrow X/\sim$ is the canonical surjection.

The following lemma is then obvious but crucial to some of our theoretical results.

**Lemma B.7.** *With the notation of the previous definition, we have:*

$$\mathbf{PMag}(X) = \sum_{x \in X} \beta^0(x)_+.$$

The next proposition is a consequence of Theorem A.24, it shows that the pseudometrics considered in practice in our work (and in our experiments) admit a positive magnitude.

**Proposition B.8.** *Let $p \in [1, 2]$ and $S \in \mathcal{Z}^n$, then every finite subset of $(\mathbb{R}^d, \rho_S^{(p)})$ admits a positive magnitude, and therefore it also has a canonical weighting.*

*Proof.* Let $\mathcal{W} := \{w_1, \ldots, w_N\}$ be a finite set in $\mathbb{R}^d$. We have

$$\|\boldsymbol{L}_S(w) - \boldsymbol{L}_S(w')\|_p = n^{1/p} \rho_S^{(p)}(w, w').$$

Therefore, if we denote by $\bar{w}$ the equivalence class of $w$ in the metric identification, it is clear that $\bar{w} = \bar{w'} \iff \boldsymbol{L}_S(w) = \boldsymbol{L}_S(w')$. Hence, the map $\varphi_S := n^{-1/p} \boldsymbol{L}_S$ naturally extends to an isometry between metric spaces:

$$\mathcal{W}/\sim \xrightarrow{\sim} \varphi_S(\mathcal{W}) \underset{\text{finite}}{\subset} \mathbb{R}^n.$$

By Theorem A.24, the finite set $\varphi_S(\mathcal{W})$ is positive definite, hence it is also the case of $\mathcal{W}/\sim$. Therefore $\mathcal{W}$ admits a positive magnitude by definition. $\square$

## B.4 Warm-up: covering bounds

The following is deduced from the transcription of the results of [22] to our setting. It is the starting point of our persistent homology-based analysis.

**Theorem B.9.** *Let $\rho$ be a pseudometric on $\mathbb{R}^d$. Suppose that Assumption 1 holds and that $\ell$ is $(q, L, \rho)$-Lipschitz, for $q \geq 1$. Then, for all $\delta > 0$, with probability at least $1 - \zeta$ over $\mu_z^{\otimes n} \otimes \mu_u^{\otimes \infty}$,*

$$\sup_{t_0 \leq i \leq T} G_S(w_i) \leq 2L\delta + 2B\sqrt{\frac{2 \log N_\delta^\rho(\mathcal{W}_{t_0 \to T})}{n}} + 3B\sqrt{\frac{\mathrm{I}_\infty(S, \mathcal{W}_{t_0 \to T}) + \log(1/\zeta)}{2n}}.$$

The proof of this theorem will be given in the next subsection. Before discussing this proof, a few remarks are in order.

Covering bounds, such as B.9 have been used in [78, 13, 10, 21] to introduce fractal dimensions (more precisely through the notion of upper box-counting dimension) into the generalization bounds. This is done via the following definition of the aforementioned upper box-counting dimension:

$$\overline{\dim}_{\mathrm{B}}^\rho(X) := \limsup_{\delta \to 0} \frac{\log N_\delta^\rho(X)}{\log(1/\delta)}.$$

By using a similar procedure, we see that our framework could be used to introduce intrinsic dimensions associated to a wide range of pseudometrics, as soon as they satisfy a $(q, L, \rho)$-Lipschitz continuity assumption.

However, arguments based on these intrinsic dimensions only make sense in the limit $T \to \infty$, which makes little sense in practical settings. To address this issue, we take inspiration from two other notions that are equal to the upper box-counting dimension (and therefore lay the ground of the numerical approximation of this dimension), namely the PH-dimension [42, 73, 10, 21] and the magnitude dimension [55, 3]. Our approach is to replace the intrinsic dimensions by the "intermediary quantities" used to define them. This leads to the results presented in the next two subsection.

## B.5    Proof of Theorem B.9

Before going to the proof of Theorem B.9, we specify our theoretical setup, which is the one introduced in [22]. In this section, we prove our results in the case $T < +\infty$. However, note that one could consider $T = +\infty$ without much technical difficulties.

The setup is the following: let $(F(\mathbb{R}^d), \mathcal{T})$ denote the set of all finite subsets of $\mathbb{R}^d$, endowed with a $\sigma$-algebra $\mathcal{T}$.

We consider the following probability distribution on $F(\mathbb{R}^d)$:

$$\forall A \in \mathcal{T}, \ \pi(A) := \int_{\mathcal{Z}^n} \rho_S(A) d\mu_z^{\otimes n}(S). \tag{12}$$

As it is discussed in [22, Section 5.4], we make the following technical measure-theoretic assumption.

**Assumption 2.** The probability measure $\mu_z^{\otimes n}$ is a strictly positive Borel measure. Moreover, for every $A \in \mathcal{T}$, the map $S \mapsto \rho_S(A)$ is continuous.

The following example highlights the fact this is a very mild assumption.

*Example* B.10. If the data space $\mathcal{Z}$ is countable and the data distribution $\mu_z$ has no null mass, then the above assumption is automatically satisfied with respect to the discrete topology.

**Theorem B.9.** *Let $\rho$ be a pseudometric on $\mathbb{R}^d$. Suppose that Assumption 1 holds and that $\ell$ is $(q, L, \rho)$-Lipschitz, for $q \geq 1$. Then, for all $\delta > 0$, with probability at least $1 - \zeta$ over $\mu_z^{\otimes n} \otimes \mu_u^{\otimes \infty}$,*

$$\sup_{t_0 \leq i \leq T} G_S(w_i) \leq 2L\delta + 2B\sqrt{\frac{2 \log N_\delta^\rho(\mathcal{W}_{t_0 \to T})}{n}} + 3B\sqrt{\frac{\mathrm{I}_\infty(S, \mathcal{W}_{t_0 \to T}) + \log(1/\zeta)}{2n}}.$$

*Proof.* Let us fix some $\zeta \in (0, 1)$. First note that thanks to Assumption 2, we have that $\rho_S$ is absolutely continuous with respect to $\pi$, $\mu_z^{\otimes n}$-almost surely. Therefore, we can introduce its Radon-Nykodym derivative, denoted by $d\rho_S/d\pi$.

Thanks to the above notation, we can apply the data-dependent Rademacher complexity bound of [22, Theorem 10] to obtain that with probability at least $1 - \zeta$, we have, for any $\lambda > 0$:

$$\sup_{t_0 \leq i \leq T} \left( \mathcal{R}(w_i) - \widehat{\mathcal{R}}_S(w_i) \right) \leq 2\mathrm{Rad}(\ell, \mathcal{W}_{t_0 \to T}, S) + \frac{1}{\lambda} \left( \frac{d\rho_S}{d\pi}(\mathcal{W}_{t_0 \to T}) + \log(1/\zeta) \right) + \lambda \frac{9B^2}{8n},$$

with $\mathrm{Rad}(\ell, \mathcal{W}_{t_0 \to T}, S)$ a Rademacher complexity term, defined by:

$$\mathrm{Rad}(\ell, \mathcal{W}_{t_0 \to T}, S) := \mathbb{E}_{\boldsymbol{\epsilon}} \left[ \sup_{w \in \mathcal{W}_{t_0 \to T}} \frac{1}{n} \sum_{i=1}^{n} \epsilon_i \ell(w, z_i) \right],$$

where $\boldsymbol{\epsilon} := (\epsilon_1, \dots, \epsilon_n)$ is a vector of independent centered Bernoulli random variables.

By [22, Lemma 16], we have almost surely that:

$$\frac{d\rho_S}{d\pi}(\mathcal{W}_{t_0 \to T}) \leq \mathrm{I}_\infty(\mathcal{W}_{t_0 \to T}, S).$$

Therefore, by optimizing the choice of the parameter $\lambda$ in the above equation, we have that:

$$\sup_{t_0 \leq i \leq T} \left( \mathcal{R}(w_i) - \widehat{\mathcal{R}}_S(w_i) \right) \leq 2\mathrm{Rad}(\ell, \mathcal{W}_{t_0 \to T}, S) + 3B\sqrt{\frac{\mathrm{I}_\infty(S, \mathcal{W}_{t_0 \to T}) + \log(1/\zeta)}{2n}}. \quad (13)$$

We now perform a covering argument very similar to classical covering arguments for Rademacher complexity[77]. Let us fix some $\delta > 0$ and introduce $(x_1, \dots, x_{N_\delta^\rho(\mathcal{W}_{t_0 \to T})})$ the centers of a minimal $\delta$-covering of $\mathcal{W}_{t_0 \to T}$ for pseudometric $\rho$. For any $w \in \mathcal{W}_{t_0 \to T}$, there exists $j$ such that $\rho(w, x_j) \leq \delta$. Therefore we have:

$$
\begin{aligned}
\sup_{w \in \mathcal{W}_{t_0 \to T}} \frac{1}{n} \sum_{i=1}^{n} \epsilon_i \ell(w, z_i) &\leq \sup_{1 \leq j \leq N_\delta^\rho(\mathcal{W}_{t_0 \to T})} \frac{1}{n} \sum_{i=1}^{n} \epsilon_i \ell(x_j, z_i) + \frac{1}{n} \sum_{i=1}^{n} \epsilon_i (\ell(w, z_i) - \ell(x_j, z_i)) \\
&\leq \sup_{1 \leq j \leq N_\delta^\rho(\mathcal{W}_{t_0 \to T})} \frac{1}{n} \sum_{i=1}^{n} \epsilon_i \ell(x_j, z_i) + \frac{1}{n} \sum_{i=1}^{n} |\ell(w, z_i) - \ell(x_j, z_i)| \\
&\leq \sup_{1 \leq j \leq N_\delta^\rho(\mathcal{W}_{t_0 \to T})} \frac{1}{n} \sum_{i=1}^{n} \epsilon_i \ell(x_j, z_i) + n^{-1/q} \left\| \boldsymbol{L}_S(w) - \boldsymbol{L}_S(x_j) \right\|_q,
\end{aligned}
$$

where the last line comes from Hölder's inequality.

We can now apply Massart's lemma on the first term and the $(q, L, \rho)$-Lipschitz continuity of $\ell$ on the second term, this gives us:

$$\mathrm{Rad}(\ell, \mathcal{W}_{t_0 \to T}, S) \leq L\delta + B\sqrt{\frac{2 \log N_\delta^\rho(\mathcal{W}_{t_0 \to T})}{n}},$$

which concludes the proof.

$\square$

## B.6 Persistent homology bounds

We now present the proofs of our persistent homology-based bounds, ie, the results of section 3.2.

The following lemma is a pseudometric version of a classical result of fractal geometry [26].

**Lemma B.11** (Covering and packing in pseudometric spaces)**.** *Let $(X, \rho)$ be a pseudometric space, $\delta > 0$, and $\{x_1, \dots, x_{P_\delta(X)}\}$ a maximal $\delta$-packing of $X$ for pseudometric $\rho$. Then we have:*

$$N_{2\delta}^\rho(X) \leq P_\delta^\rho(X).$$

*Proof.* Let us fix $\delta > 0$ and let $(x_1, \dots, x_{P_\delta^\rho(X)})$ be centers of a maximal packing of $X$ with closed $\delta$-balls. Let us assume that:

$$X \backslash \bigcup_{1 \leq i \leq P_\delta^\rho(X)} \bar{B}_{2\delta}(x_i) \neq \emptyset,$$

so that we can take some $x_0$ belonging to the above non-empty set. Now let us fix $i \in \{1, \dots, P_\delta^\rho(X)\}$ and $w \in \bar{B}_\delta(x_i)$. By the triangle inequality and the definition of $w$ and $x_0$, we have:

$$\underbrace{\rho(x_0, x_i)}_{>2\delta} \leq \rho(x_0, w) + \underbrace{\rho(w, x_i)}_{\leq \delta}.$$

Therefore, we have $\rho(x_0, w) > \delta$, and hence $\bar{B}_\delta(x_i) \cap \bar{B}_\delta(x_0)$, so that we construct a bigger $\delta$-packing, by adding $x_0$ to $(x_1, \ldots, x_{P_\delta^\rho(X)})$, which is absurd.

Therefore, we have $X \setminus \bigcup_{1 \le i \le P_\delta^\rho(X)} \bar{B}_{2\delta}(x_i) = \emptyset$, hence the result. $\qquad\square$

The next lemma asserts that $E_\alpha$ is increasing (with respect to the inclusion of sets), if and only if $\alpha \le 1$. This is the reason why we require $\alpha \in [0, 1]$ in Theorem 3.4.

**Lemma B.12.** *Let $(X, \rho)$ be a non-empty finite pseudometric space, $\alpha \in [0, 1]$ and $\delta > 0$. Then we have:*

$$E_\alpha^\rho(X) \ge \frac{P_\delta^\rho(X)}{2} \delta^\alpha$$

*Proof.* We refer to Figure 5 for a graphical illustration of the main technical elements of this proof.

In the case where $P_\delta^\rho(X) = 1$, the result is obvious. In the rest of the proof we assume that $P_\delta^\rho(X) \ge 2$

In all the following, we fix $\alpha \in [0, 1]$ and $\delta > 0$. We also denote $P := P_\delta^\rho(X)$. Without loss of generality, we can assume $P \ge 2$.

We fix $\mathcal{T}$ an MST of $X$, represented by a set of edges denoted $x \to y$, with $x, y \in X^2$ (note that we identify $x \to y$ and $y \to x$). It is a classical result that there are $|X| - 1$ edges. For an edge $e$ of the form $a \to b$, we denote its length by $|e| := \rho(a, b)$.

For $a, b \in X$, with $a \ne b$, we denote by $\{a \to b\}$ the shortest path between $a$ and $b$. More precisely, we represent it as a list of edges, denoted $a = a_0 \to a_1 \cdots \to a_K = b$, for some $K$. When the context is clear, we identify $\{a \to b\}$ to the set of its edges $a \to b$.

Let us introduce $(x_1, \ldots, x_P)$ a maximal $\delta$-packing of $X$ by closed.

For every $i \in \{1, \ldots, P\}$, as $\mathcal{T}$ is connected, there exists $y_i \in X$ such that $y_i \notin \bar{B}_\delta(x_i)$ and $y_i$ is the only point in the path $\{x_i \to y_i\}$ that does not belong to the ball $\bar{B}_\delta(x_i)$.

For each $i$, we denote $e_i$ the only edge in $\{x_i \to y_i\}$ to which $y_i$ belongs, *i.e.* $e_i$ is of the form $z_i \to y_i$, with $z_i \in \bar{B}_\delta(x_i)$. By construction, those edges $e_i$ are the only ones that can be shared by several paths $\{x_i \to y_i\}$.

Let us introduce the following set of indices:

$$I := \{i \in \{1, \ldots, P\}, \ \forall j \ne i, \ e_i \notin \{x_j \to y_j\}\}, \quad K := \{1, \ldots, P\} \setminus I.$$

Let us consider $i \in K$. Let us assume that we have $j, j' \in \{1, \ldots, P\}$ such that $e_i \in \{x_j \to y_j\}$ and $e_i \in \{x_{j'} \to y_{j'}\}$. If we denote $e_i$ as $z_i \to y_i$, we have that $z_i \in \bar{B}_\delta(x_i)$, by definition of $y_i$. Therefore, by definition of $y_j$, we have $z_i = y_j$ (because $\bar{B}_\delta(x_i) \cap \bar{B}_\delta(x_j) = \emptyset$). We have similarly $z_i = y_{j'}$ and thus $y_j = y_{j'}$. By definition of $y_j$ and $y_{j'}$ we also have $y_i \in \bar{B}_\delta(x_j) \cap \bar{B}_\delta(x_{j'})$, which is absurd, by definition of packing. We conclude the following:

$$\forall k \in K, \ \exists! j \ne i, \ e_i \in \{x_j \to y_j\}.$$

For $k \in K$, we denote the corresponding $j$ by $\varphi(k)$.

By definition of $K$, it is clear that $\varphi(k) \in K$. Moreover, as $y_{\varphi(i)} = z_i \in \bar{B}_\delta(x_i)$, this implies that $\varphi^2(i) = i$. Therefore, we have constructed an involution,

$$\varphi : K \longrightarrow K,$$

such that $\forall k \in K$, $\varphi(k) \ne k$. This implies that the cardinality of $K$ is even and that we can write $K = K_1 \coprod K_2$, with:

$$|K_1| = |K_2|, \quad \varphi(K_1) = K_2.$$

The outcome of this construction is that we now have disjoint paths given by the $(x_i \to y_i)_{i \in I}$ and the $(x_k \to x_{\varphi(k)})_{k \in K_1}$. Therefore, we get the following lower bound on $E_\alpha(X)$.

$$E_\alpha(X) \ge \sum_{i \in I} \sum_{e \in \{x_i \to y_i\}} |e|^\alpha + \sum_{k \in K_1} \sum_{e \in \{x_k \to x_{\varphi(k)}\}} |e|^\alpha.$$

As $\alpha \in [0,1]$, we have that:

$$\boldsymbol{E}_\alpha(X) \geq \sum_{i \in I} \left( \sum_{e \in \{x_i \to y_i\}} |e| \right)^\alpha + \sum_{k \in K_1} \left( \sum_{e \in \{x_k \to x_{\varphi(k)}\}} |e| \right)^\alpha .$$

By the triangle inequality, and by definition of packing, we have:

$$\boldsymbol{E}_\alpha(X) \geq \sum_{i \in I} \delta^\alpha + \sum_{k \in K_1} \delta^\alpha = \delta^\alpha(|I| + |K_1|) \geq \frac{1}{2} P_\delta^\rho(X)\delta^\alpha,$$

which concludes the proof. $\qquad\square$

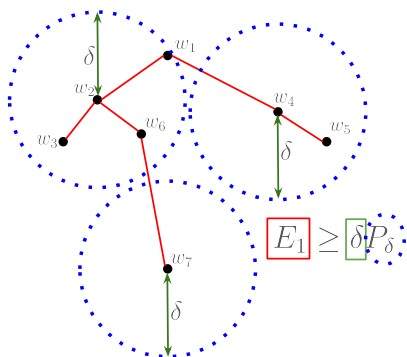

Figure 5: Geometric representation of the proof of Lemma B.12. It represents a point cloud $(w_i)_i$, the centers of the 3 packing balls (blue) and the minimum spanning tree $T$ (red), so that the sum of the lengths of the edges of $T$ is exactly $E_1$, see Appendix A.

**Theorem 3.4.** *Let $\rho$ be a pseudometric on $\mathbb{R}^d$. Supposes that Assumption 1 holds and that $\ell$ is $(q, L, \rho)$-Lipschitz, for $q \geq 1$. Then, for all $\alpha \in [0,1]$, with probability at least $1 - \zeta$, we have:*

$$\sup_{t_0 \leq i \leq T} G_S(w_i) \leq 2B\sqrt{\frac{2\log\left(1 + K_{n,\alpha}\boldsymbol{E}_\alpha^\rho\right)}{n}} + \frac{2B}{\sqrt{n}} + 3B\sqrt{\frac{I_\infty(S, \mathcal{W}_{t_0 \to T}) + \log(1/\zeta)}{2n}},$$

*with $K_{n,\alpha} := 2\left(2L\sqrt{n}/B\right)^\alpha$.*

*Proof.* For better clarity, we assume $T < +\infty$. Let us fix some $\zeta \in (0,1)$, $\delta > 0$, and $\alpha \geq 0$. By Theorem B.9, we have, with probability at least $1 - \zeta$:

$$\sup_{t_0 \leq i \leq T} \left( \mathcal{R}(w_i) - \widehat{\mathcal{R}}_S(w_i) \right) \leq 2L\delta + 2B\sqrt{\frac{2\log N_\delta^\rho(\mathcal{W}_{t_0 \to T})}{n}} + 3B\sqrt{\frac{I_\infty(S, \mathcal{W}_{t_0 \to T}) + \log(1/\zeta)}{2n}}.$$

We now bound the covering number appearing in the above equation. By Lemma B.12, we have:

$$\boldsymbol{E}_\alpha^\rho(\mathcal{W}_{t_0 \to T}) \geq 2^{-\alpha-1} \left[ P_{\delta/2}^\rho(\mathcal{W}_{t_0 \to T}) - 1 \right] \delta^\alpha.$$

Moreover, by Lemma B.11, we have:

$$\boldsymbol{E}_\alpha^\rho(\mathcal{W}_{t_0 \to T}) \geq 2^{-\alpha-1} \left[ N_\delta^\rho(\mathcal{W}_{t_0 \to T}) - 1 \right] \delta^\alpha.$$

We now combine this with our generalization bound by choosing the value:

$$\delta := \frac{B}{L\sqrt{n}},$$

and we get that with probability at least $1 - \zeta$, we have:

$$\sup_{t_0 \leq i \leq T} \left( \mathcal{R}(w_i) - \widehat{\mathcal{R}}_S(w_i) \right) \leq \frac{2B}{\sqrt{n}} + 2B\sqrt{\frac{2\log(1 + K_{n,\alpha}\boldsymbol{E}_\alpha^\rho(\mathcal{W}_{t_0 \to T}))}{n}}$$
$$+ 3B\sqrt{\frac{I_\infty(S, \mathcal{W}_{t_0 \to T}) + \log(1/\zeta)}{2n}},$$

with:

$$K_{n,\alpha} := 2 \left( \frac{2L\sqrt{n}}{B} \right)^{\alpha}.$$

It leads to the desired result. $\qquad\square$

## B.7 Proof of the magnitude-based generalization bounds

**Lemma B.13.** *Let $\mathcal{W} \subset \mathbb{R}^d$ be a finite set and $\epsilon := (\epsilon_1, \ldots, \epsilon_n)$ and $\rho$ a pseudometric such that $(\mathcal{W}, \lambda\rho)$ admits a positive magnitude (according to Definition B.5) for every $\lambda > 0$. We assume that $\ell$ is $(L, q, \rho)$-Lipschitz continuous with $q \in [1, 2]$. Then, for any $\lambda > 0$, we have:*

$$\mathbb{E}_{\epsilon} \left[ \exp \left\{ \frac{\lambda}{n} \sup_{w \in \mathcal{W}} \sum_{i=1}^{n} \epsilon_i \ell(w, z_i) \right\} \right] \le e^{\frac{\lambda^2 B^2}{2n}} \mathbf{PMag} \left( (L\lambda)\mathcal{W} \right).$$

*where $\mathbf{PMag}$ is the positive magnitude, see Appendix B.3*

*Proof.* We first remark that, by Hölder's inequality and the $(L, q, \rho)$-Lipschitz condition, we have:

$$\forall w, w' \in \mathcal{W}, \ \rho_S(w, w') \le n^{-1/q} \left\| \boldsymbol{L}_S(w) - \boldsymbol{L}_S(w') \right\|_q \le L\rho(w, w').$$

Let us fix some $\lambda > 0$. As $(\mathcal{W}, \lambda\rho)$ admits a positive magnitude, we can introduce a canonical weighting $\beta : \mathcal{W} \longrightarrow \mathbb{R}$. By definition of a weighting, we have

$$\forall a \in \mathcal{W}, \ \sum_{b \in \mathcal{W}} e^{-\lambda\rho(a,b)} \beta(b) = 1.$$

Moreover, for any $\epsilon \in \{-1, 1\}^n$, we introduce:

$$a_\epsilon := \operatorname{argmax}_{a \in \mathcal{W}} \sum_{i=1}^{n} \epsilon_i \ell(a, z_i).$$

With those notations, we can compute:

$$1 \le \sum_{b \in \mathcal{W}} e^{-\lambda\rho(a_\epsilon, b)} \beta_+(b)$$

$$\le \sum_{b \in \mathcal{W}} e^{-\frac{\lambda}{L}\rho_S(a_\epsilon, b)} \beta_+(b)$$

$$= \sum_{b \in \mathcal{W}} \exp \left\{ -\frac{\lambda}{Ln} \sum_{i=1}^{n} |\ell(a_\epsilon, z_i) - \ell(b, z_i)| \right\} \beta_+(b)$$

$$\le \sum_{b \in \mathcal{W}} \exp \left\{ -\frac{\lambda}{Ln} \sum_{i=1}^{n} \epsilon_i (\ell(a_\epsilon, z_i) - \ell(b, z_i)) \right\} \beta_+(b)$$

$$= \exp \left\{ -\frac{\lambda}{Ln} \sum_{i=1}^{n} \epsilon_i \ell(a_\epsilon, z_i) \right\} \sum_{b \in \mathcal{W}} \exp \left\{ \frac{\lambda}{Ln} \sum_{i=1}^{n} \epsilon_i \ell(b, z_i) \right\} \beta_+(b).$$

Therefore, by dividing by the first term on the right-hand side and using the independence of the $\epsilon_i$, we deduce that:

$$\mathbb{E}_{\epsilon} \left[ \exp \left\{ \frac{\lambda}{Ln} \sup_{w \in \mathcal{W}} \sum_{i=1}^{n} \epsilon_i \ell(w, z_i) \right\} \right] \le \mathbb{E}_{\epsilon} \left[ \sum_{b \in \mathcal{W}} \exp \left\{ \frac{\lambda}{Ln} \sum_{i=1}^{n} \epsilon_i \ell(b, z_i) \right\} \beta_+(b) \right]$$

$$= \sum_{b \in \mathcal{W}} \prod_{i=1}^{n} \mathbb{E}_{\epsilon} \left[ e^{\frac{\lambda}{Ln} \epsilon_i \ell(b, z_i)} \right] \beta_+(b).$$

By Hoeffding's lemma, we have:

$$\mathbb{E}_{\epsilon} \left[ \exp \left\{ \frac{\lambda}{Ln} \sup_{w \in \mathcal{W}} \sum_{i=1}^{n} \epsilon_i \ell(w, z_i) \right\} \right] \le e^{\frac{\lambda^2 B^2}{2nL^2}} \sum_{b \in \mathcal{W}} \beta_+(b)$$

$$= e^{\frac{\lambda^2 B^2}{2nL^2}} \mathbf{PMag} \left( \lambda\mathcal{W} \right).$$

The result follows by the change of variable $\lambda = \Lambda L$. $\qquad\square$

**Theorem 3.5.** *Let $\rho$ be a pseudometric such that $(\mathcal{W}, \lambda\rho)$ admits a positive magnitude (according to Definition B.5) for every $\lambda > 0$. We assume that $\ell$ is $(q, L, \rho)$-Lipschitz continuous with $q \geq 1$. Then, for any $s > 0$, we have with probability at least $1 - \zeta$ that*

$$\sup_{t_0 \leq i \leq T} G_S(w_i) \leq \frac{2}{s} \log \mathbf{PMag}^\rho \left(Ls\mathcal{W}_{t_0 \to T}\right) + s\frac{B^2}{n} + 3B\sqrt{\frac{\mathrm{I}_\infty(S, \mathcal{W}_{t_0 \to T}) + \log(1/\zeta)}{2n}}.$$

*Proof.* The beginning of the proof is completely similar to the proof of B.9 up to Equation (13). More precisely, we have that with probability at least $1 - \zeta$:

$$\sup_{t_0 \leq i \leq T} \left(\mathcal{R}(w_i) - \widehat{\mathcal{R}}_S(w_i)\right) \leq 2\mathrm{Rad}(\ell, \mathcal{W}_{t_0 \to T}, S) + 3B\sqrt{\frac{\mathrm{I}_\infty(S, \mathcal{W}_{t_0 \to T}) + \log(1/\zeta)}{2n}}.$$

By Jensen's inequality, we have, for all $\lambda > 0$:

$$\mathrm{Rad}(\ell, \mathcal{W}_{t_0 \to T}, S) \leq \frac{1}{\lambda} \log \mathbb{E}_\epsilon \left[\exp\left\{\frac{\lambda}{n} \sup_{w \in \mathcal{W}_{t_0 \to T}} \sum_{i=1}^n \epsilon_i \ell(w, z_i)\right\}\right].$$

Therefore, we can apply Lemma B.13 to write that, for all $s > 0$:

$$\mathrm{Rad}(\ell, \mathcal{W}_{t_0 \to T}, S) \leq s\frac{B^2}{2n} + \frac{1}{s} \log \mathbf{PMag}\left(Ls\mathcal{W}_{t_0 \to T}\right).$$

We deduce that for all $s > 0$, we have with probability at least $1 - \zeta$ that:

$$\sup_{t_0 \leq i \leq T} \left(\mathcal{R}(w_i) - \widehat{\mathcal{R}}_S(w_i)\right) \leq s\frac{B^2}{n} + \frac{2}{s} \log \mathbf{PMag}\left(Ls\mathcal{W}_{t_0 \to T}\right) + \sqrt{\frac{\mathrm{I}_\infty(S, \mathcal{W}_{t_0 \to T}) + \log(1/\zeta)}{2n}}.$$

$\square$

*Remark* B.14 (Link between magnitude and positive magnitude). Let $\mathcal{W} \subset \mathbb{R}^M$ be a finite set (for some $M$), of cardinality $N$, and $\rho$ a metric on $\mathcal{W}$. If we denote the similarity matrix, for a given value of $s > 0$, by $M_s(a, b) = e^{-\rho(a,b)}$, then it is clear that:

$$M_s \xrightarrow[s \to \infty]{} I_N.$$

Moreover, by continuity of the inverse, this implies that the weighting associated to $s > 0$, *i.e.* $\beta_s : \mathcal{W} \to \mathbb{R}$, satisfy:

$$\forall a \in \mathcal{W}, \ \beta_s(a) \xrightarrow[s \to \infty]{} 1.$$

From this, we first deduce that, for $s \to \infty$, we have $\mathbf{Mag}^\rho(s\mathcal{W}) \to N$. Moreover, by continuity of the inverse, this means that, up to a certain $s$, the weighting $(\beta_s(a))_{a \in \mathcal{W}}$ only has positive elements. Therefore, this implies that, for $s$ big enough, one has $\mathbf{Mag}^\rho(s\mathcal{W}) = \mathbf{PMag}^\rho(s\mathcal{W})$.

Thanks to our definitions for positive magnitude in pseudometric spaces, given in Appendix B.3, this observation extends to the pseudometric case.

*Remark* B.15 (Extension to infinite sets). There exist extensions of the definition of magnitude beyond finite sets [55, 56]. More specifically, weightings are then represented by measures on the set. It is clear from the above proofs that we can extend the positive magnitude in this setting and that the proof would follow similar lines. Therefore, our theory provides upper bounds of Rademacher complexity in terms of positive magnitude in more general cases than the one we use in this work.

In particular, the present reasoning could be extended to compact random sets $\mathcal{W}$. The next lemma is the extension of Lemma B.13 to the compact setting. The proof follows very similar lines as Lemma B.13.

**Lemma B.16.** *Let us fix $S \in \mathcal{Z}^n$ and consider a set $W \subset \mathbb{R}^d$ that is compact[8] with respect to the pseudometric $\rho_S$. For every $\lambda > 0$, we assume that $A$ possesses a weighting $\mu_\lambda$ (which is a finite*

---

[8]As we did in Appendix B.3, the positive magnitude of compact spaces should be properly extended to pseudometric spaces, we omit the details as it is very similar to the case treated in this paper.

*measure on A) with respect to pseudometric $\lambda \rho_S$, in the sense of [56, Definition 3.3]. Then we have, for any $\lambda > 0$:*

$$\mathbb{E}_\epsilon \left[ \exp \left\{ \frac{\lambda}{n} \sup_{a \in A} \sum_{i=1}^{n} \epsilon_i \ell(w, z_i) \right\} \right] \le e^{\frac{\lambda^2 B^2}{2n}} \mathbf{PMag}\left(\lambda A\right).$$

*where $\mathbf{PMag}$ is the positive magnitude of the compact space $(A, \lambda \rho_S)$ which, according to [56], can be defined as:*

$$\mathbf{PMag}\left(\lambda A\right) = \left(\mu_\lambda\right)_+ (A),$$

*where $\left(\mu_\lambda\right)_+$ denotes the positive part of the measure $\mu_\lambda$.*

## C    Additional Experimental Details

In this section, we give additional details regarding the models, datasets, and hyperparameters used in our experiments.

### C.1    Experimental setting

#### C.1.1    Vision Transformers Architecture and implementation details

Table 2: Architecture details for the vision transformers (taken from [29]). *WS refers to Window Size.*

| MODEL | DATASET | DEPTH | PATCH SIZE | TOKEN DIM | HEADS | MLP-RATIO | WS | #PARAMS |
|---|---|---|---|---|---|---|---|---|
| VIT [79] | CIFAR10 | 9 | 4 | 192 | 12 | 2 | - | 2697610 |
| VIT [79] | CIFAR100 | 9 | 4 | 192 | 12 | 2 | - | 2714980 |
| SWIN [48] | CIFAR10 | [2,4,6] | 4 | 96 | [3,6,12] | 2 | 4 | 7048612 |
| SWIN [48] | CIFAR100 | [2,4,6] | 4 | 96 | [3,6,12] | 2 | 4 | 7083262 |
| CAIT [80] | CIFAR10 | 24 | 4 | 192 | 4 | 2 | - | 8053450 |
| CAIT [80] | CIFAR100 | 24 | 4 | 192 | 4 | 2 | - | 8070820 |

The design of the ViT has been modified to accommodate for the small datasets as per [68]. Our implementation is based on the [29], which is based on the `timm` library with the architecture parameters presented in Table 2. The implementation of Swin is based on the Swin-Transformer libarary and the implementation of CaiT is predominantly based on the `timm` library with some modifications. The full version can be found in the supplementary code.

Instead of training from scratch, which is extremely time-consuming, we used the pre-trained weights available from the GitHub repository of the paper [29], we further finetuned them for 100 epochs on the dataset CIFAR10 or CIFAR100 to achieve the optimum performance reported in the paper [29]. Then we verified that the finetuned weights achieved 100% training performance, and then they were the starting point of our computational framework. We ran the transformer experiments on 18 NVIDIA 2080Ti GPUs, and the graph experiments on 18 Intel Xeon Silver 4114 CPUs.

#### C.1.2    GNN Architecture and implementation details

We will briefly talk about the details of GraphSage [32] and GatedGCN [12], prior works we use in our experiments. GraphSage [32] is an improvement over the GCN (Graph ConvNets) model [41] and it incorporates each node's own features from the previous layer in an explicit way by the update equation:

$$h_i^{l+1} = \text{ReLU}(U^l \text{Concat}(h_i^l, \text{Mean}_{j \in N_i} h_j^l)),$$

where $N_i$ is the neighbourhood of node $i$, $h_i^l$ is the feature vector and $U^l \in \mathbb{R}^{d \times 2d}$. We use the graph-pooling version of GraphSage, with the following update equation:

$$h_i^{l+1} = \text{ReLU}(U^l \text{Concat}(h_i^l, \text{Max}_{j \in N_i} \text{ReLU}(V^l h_j^l))),$$

where $V^l \in \mathbb{R}^{d \times d}$. GatedGCN (Gated Graph ConvNet) [12] uses the following update equation:

$$h_i^{l+1} = h_i^l + \mathrm{ReLU}(\mathrm{BN}(U^l h_i^l + \sum_{j \in N_i} e_{ij}^l \odot V^l h_i^l)),$$

where $U^l, V^l \in \mathbb{R}^{d \times d}$, $\odot$ is the Hadamard product, and the edge gates $e_{ij}^l$ have the following definitions:

$$e_{ij}^l = \frac{\sigma(\hat{e}_{ij}^l)}{\sum_{j' \in N_i} \sigma(\hat{e}_{ij}^l) + \epsilon},$$

$$\hat{e}_{ij}^l = \hat{e}_{ij}^{l-1} + \mathrm{ReLU}(\mathrm{BN}(A^l h_i^{l-1} + B^l h_i^{l-1} + C^l \hat{e}_{ij}^{l-1})),$$

where $\sigma$ is the sigmoid funciton, $\epsilon$ is a small constant for numerical stability, $A^l, B^l, C^l \in \mathbb{R}^{d \times d}$, and BN stands for Batch Normalization.

We used the code provided by [23], which relies on the `dgl` library implementation of GraphSage and GatedGCN. We trained GraphSage and GatedGCN until 100% training accuracy, following the setup in [23]. All experiments were ran on 18 Intel Xeon Silver 4114 CPUs. Each experiment (one fixed batch size and learning rate) was run on a single CPU and 18 experiments were run on the server at any given time (on different CPUs).

### C.2 Hyperparameter details

**Hyperparameters shared among experiments.**. For the Vision Transformers experiments, we varied the learning rate range $[10^{-5}, 10^{-3}]$, and batch size in the range $[8, 256]$. For the graph experiments, $[10^{-6}, 10^{-4}]$, and batch size in the range $[8, 256]$. For all experiments, we used 0.1 proportion of the training data for the computation of the pseudo matrix, apart from CaiT and Swin on CIFAR100, where we used 0.09 proportion of the training data due to memory constraints. All experiments use a $6 \times 6$ grid of hyperparameters which is specified as follows.

**ViT on CIFAR10.** We selected 6 values for the learning rate in the range $[10^{-5}, 10^{-3}]$, and the batch size between $[8, 256]$, and data proportion for the computation of the pseudo-distance ($\rho_S$) of $10\%$ (see section 4).

**ViT on CIFAR100.** We selected 6 values for the learning rate in the range $[10^{-5}, 10^{-3}]$, and the batch size between $[8, 256]$, and data proportion for the computation of the pseudo-distance ($\rho_S$) of $10\%$ (see section 4).

**CaiT on CIFAR10.** We selected 6 values for the learning rate in the range $[10^{-5}, 10^{-3}]$, batch size between $[8, 256]$, and data proportion for the computation of the pseudo-distance ($\rho_S$) of $10\%$ (see section 4).

**CaiT on CIFAR100.** We selected 6 values for the learning rate in the range $[10^{-5}, 10^{-3}]$, batch size between $[8, 256]$, and data proportion for the computation of the pseudo-distance ($\rho_S$) of $9\%$ (see section 4).

**Swin on CIFAR10.** We selected 6 values for the learning rate in the range $[10^{-5}, 10^{-3}]$, batch size between $[8, 256]$, and data proportion for the computation of the pseudo-distance ($\rho_S$) of $10\%$ (see section 4).

**Swin on CIFAR100.** We selected 6 values for the learning rate in the range $[10^{-5}, 10^{-3}]$, batch size between $[8, 256]$, and data proportion for the computation of the pseudo-distance ($\rho_S$) of $9\%$ (see section 4).

**GatedGCN.** We selected 6 values for the learning rate in the range $[10^{-6}, 10^{-4}]$, the batch size between $[8, 256]$ and data proportion for the computation of the pseudo-distance ($\rho_S$) of $10\%$ (see section 4). We note that for due to time constraints, the experiments with batch sizes of 8 and 256 for the Euclidean metric were not complete.

**GraphSage.** We selected 6 values for the learning rate in the range $[10^{-6}, 10^{-4}]$, the batch size between $[8, 256]$, and data proportion for the computation of the pseudo-distance ($\rho_S$) of $10\%$ (see section 4).

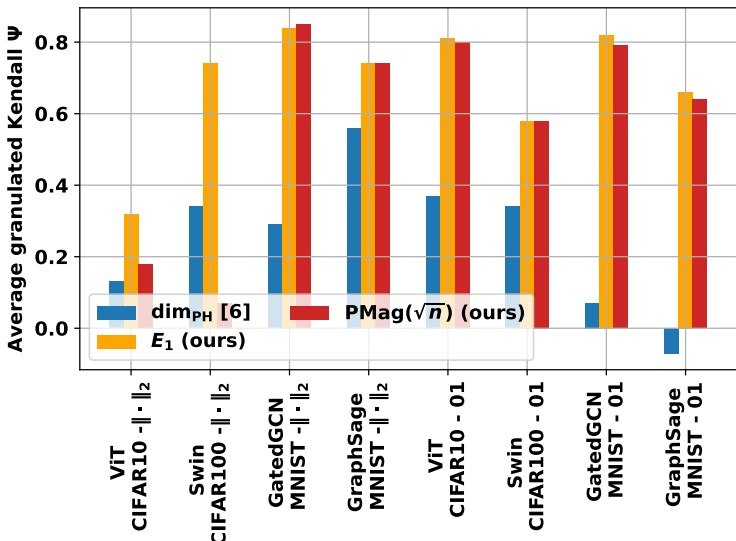

Figure 6: Comparison of topological complexities for different models, datasets, and pseudometrics. The results are a visual representation of some results from Table 1, they complete Fig. 1.c in the main par of the paper.

**ViT on CIFAR**10 **(Adam).** We selected 6 values for the learning rate in the range $[10^{-5}, 10^{-3}]$, and the batch size between $[8, 256]$, and data proportion for the computation of the pseudo-distance ($\rho_S$) of 10% (see section 4).

**ViT on CIFAR**10 **(SGD).** We selected 6 values for the learning rate in the range $[5 \times 10^{-3}, 10^{-1}]$, and the batch size between $[8, 256]$, and data proportion for the computation of the pseudo-distance ($\rho_S$) of 10% (see section 4).

**ViT on CIFAR**10 **(RMSprop).** We selected 6 values for the learning rate in the range $[10^{-6}, 10^{-3}]$, and the batch size between $[8, 512]$, and data proportion for the computation of the pseudo-distance ($\rho_S$) of 10% (see section 4).

# D    Additional experimental results

In this section, we present additional empirical results, in addition to what was already presented in the main part of this document. We divide this section into three parts. Additional graphical representation of our main experimental results is presented in Appendix D.1. Then, we quickly explore in Appendix D.2 additional ablation studies and comparison of our proposed topological complexities with a complexity notion that is more standard in the literature, namely gradient variance [37]. In Appendix D.3 we report additional experiments based on vision transformers and in Appendix D.4 we include additional illustrations of the GNN experiments.

## D.1    Additional graphical representations

In this section, we include additional bar plots, shown in Figure 6, which are meant to provide more visual support to understand the results of Table 1. Figure 6 completes the bar plots shown in Figure 1.

## D.2    Further ablations and comparison with other complexity metrics

### D.2.1    About the final accuracy gap and the worst accuracy gap

Our main theoretical results, presented in section 3, apply to the worst-case generalization error over the trajectory, *i.e.* on the quantity $\sup_{t_0 \leq k \leq T} \left( \mathcal{R}(w_k) - \widehat{\mathcal{R}}_S(w_k) \right)$. However, computing this quantity over the whole trajectory may be extremely expensive as it requires evaluating the

model on the whole dataset at each iteration (this is a similar problem to the one encountered for the computation of the data-dependent distance matrices, discussed in section 4). Previous studies on worst-case TDA-inspired generalization bounds circumvented this issue by reporting the final accuracy gap as the "generalization error" in their experiments (as it is the case in our work, most existing experiments consist of classification tasks).

In our work, we argue that the true worst-case generalization error may however have a different behavior than the final accuracy gap. In order to estimate this quantity in a computationally friendly way, we used the following procedure: we periodically estimated the test accuracy during the training, computed its minimum value $\mathrm{acc}_{\text{test-worst}}$ and substracted it from the final train accuracy ($\mathrm{acc}_{\text{train-final}}$) to obtain the "generalization gap" $\widehat{G}_S$ reported in our main experiments, *i.e.*,

$$\widehat{G}_S := \mathrm{acc}_{\text{train-final}} - \mathrm{acc}_{\text{test-worst}}.$$

Note that in addition to being a good proxy to the true error appearing in our theory, the above quantity could be of independent experimental interest.

In order to assess that our main conclusions remain valid if the final accuracy gap is used instead of $\widehat{G}_S$, we present here a few additional experiments using the final accuracy gap as a generalization measure (it is denoted `Accuracy gap` in the figures.) In the case of a ViT on CIFAR10, this is shown in Figure 7 and Figure 8. We observe that our proposed topological complexities also correlate very well with the final accuracy gap, and outperform the previously proposed PH dimensions [10, 21].

In addition to these findings, we make two additional new observations. First, the Ph dim, while outperformed by our proposed metric, has better granulated Kendall's coefficients when compared to the final accuracy gap than the worst generalization error ($\boldsymbol{\Psi}$ goes from $0.20$ to $0.36$). This may explain why we observed poor performance of PH-dim in Figure 4a. Second, we observe that the correlation seems to be slightly less good with the final accuracy gap, especially for high learning rates, which seems to be similar behavior to what was reported in [21].

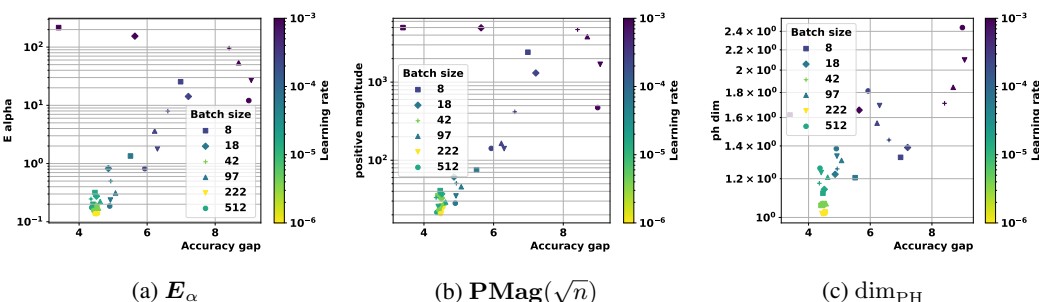

(a) $\boldsymbol{E}_\alpha$        (b) $\mathbf{PMag}(\sqrt{n})$        (c) $\dim_{\mathrm{PH}}$

Figure 7: ViT on CIFAR10 with $\rho_S$-pseudometric, using the final accuracy gap as a generalization measure.

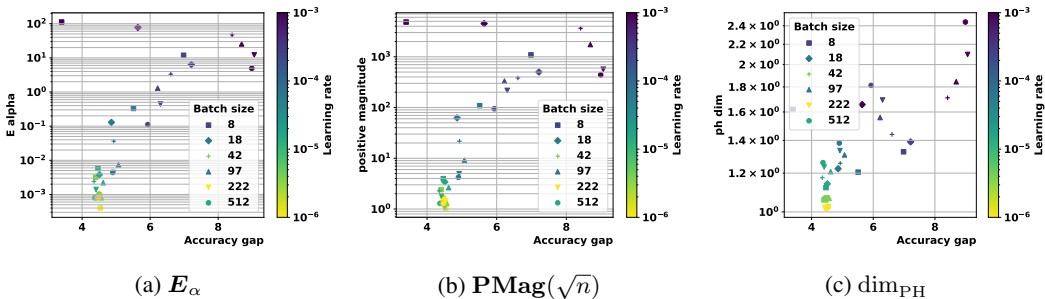

(a) $\boldsymbol{E}_\alpha$        (b) $\mathbf{PMag}(\sqrt{n})$        (c) $\dim_{\mathrm{PH}}$

Figure 8: ViT on CIFAR10 with 01-pseudometric, using the final accuracy gap as a generalization measure.

### D.2.2 Sensitivity to the scale parameter in magnitude experiments

As is explicitly shown by Theorem 3.5, using (positive) magnitude as a topological complexity requires choosing the scale parameter $s > 0$. In our main experiments, we experimented with both $s = \sqrt{n}$ (justified to obtain the expected $1/\sqrt{n}$ in the generalization bound) and $s = 0.01$ (in order to compare with using a small value for $s$). We can see in Table 1 that both settings give relatively satisfactory results. Note that in our setting we have $\sqrt{n} \approx 223.6$.

We present in Figure 9 the observed correlation between positive magnitude and generalization error for several intermediary values of $s$. This experiment was made with a ViT on the CIFAR10 dataset, using the ADAM optimizer. We observe a relative stability of the correlation $\Psi$ with respect to $s$. In this particular case, the correlation is extremely stable for higher values of $s$ while it displays more variability for smaller values of $s$. Further experiments would be necessary to understand whether this behavior is general and could then lead to the discovery of more stable magnitude-related complexities, which we leave for future work.

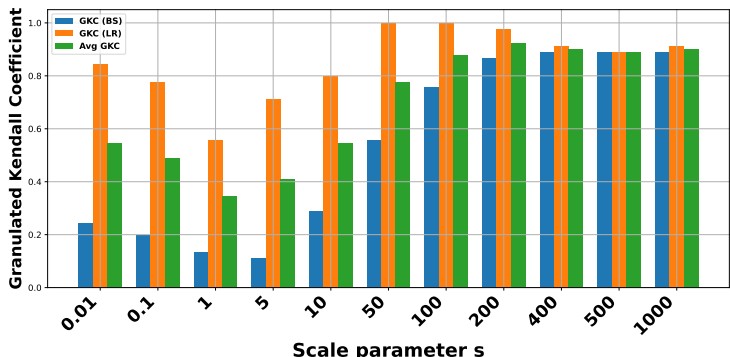

Figure 9: Sensitivity analysis of the scale parameter $s$ for positive magnitude $\mathbf{PMag}(s\mathcal{W}_{t_0 \to \tau})$. Experiment made with ViT on CIFAR10 and ADAM optimizer.

### D.2.3 Comparison with gradient variance as a generalization measure

In this short subsection, we investigate the performance comparison of our proposed topological complexities with the more widely used gradient variance, which appears for instance in [37].

In this experiment, conducted with a ViT on the CIFAR10 dataset and the ADAM optimizer, we observe very similar performance between $E_1$, $\mathbf{PMag}(\sqrt{n}\mathcal{W}_{t_0 \to \tau})$ and the gradient variance. Note that the fact that $E_1$ and $\mathbf{PMag}(\sqrt{n}\mathcal{W}_{t_0 \to \tau})$ yield similar correlation was already observed on Table 1. This tends to suggest that these three complexity measures may be able to capture similar aspects of the geometry around a local minimum.

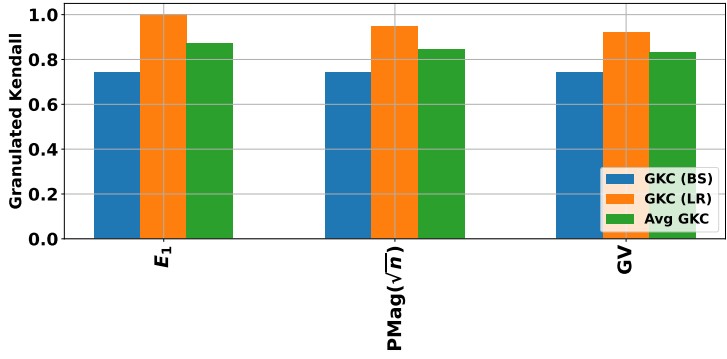

Figure 10: Comparison of the granulated Kendall coefficients of topological complexities vs gradient variance (GV). $\psi_{\mathrm{LR}}$ is denoted GKC (LR), $\psi_{\mathrm{BS}}$ is denoted GKC (BS) and the averaged coefficient $\Psi$ is denoted Avg GKC. Experiment made with ViT on CIFAR10 and ADAM optimizer.

*Remark* D.1. It should be noted that the primary goal of the introduced topological complexity measure is not to outperform existing measures such as a gradient variance but rather to demonstrate the empirical importance of the topology of the trajectory for generalization error.

## D.3 Vision Transformers - additional experiments

We compare the performance of the different metrics by using the granulated Kendall's coefficients introduced in [37]. The experiments presented here use 3 different Vision Transformers (ViT [79], CaiT [80], Swin [48]) on CIFAR10 and CIFAR100. As a baseline, we use the $\dim_{\mathrm{PH}}$ introduced in [10] and the data-dependent dimension with the pseudometric $\dim_{\mathrm{PH}}$ from [21].

Here we present the full results on each dataset and model. They can be found in Table 4 for CaiT and CIFAR10, 6 for Swin and CIFAR10, 3 for ViT and CIFAR100 and 5 for CaiT and CIFAR100. The plots from each experiment for every computed quantity can be found in (the remaining 3 quantities for ViT and CIFAR10).

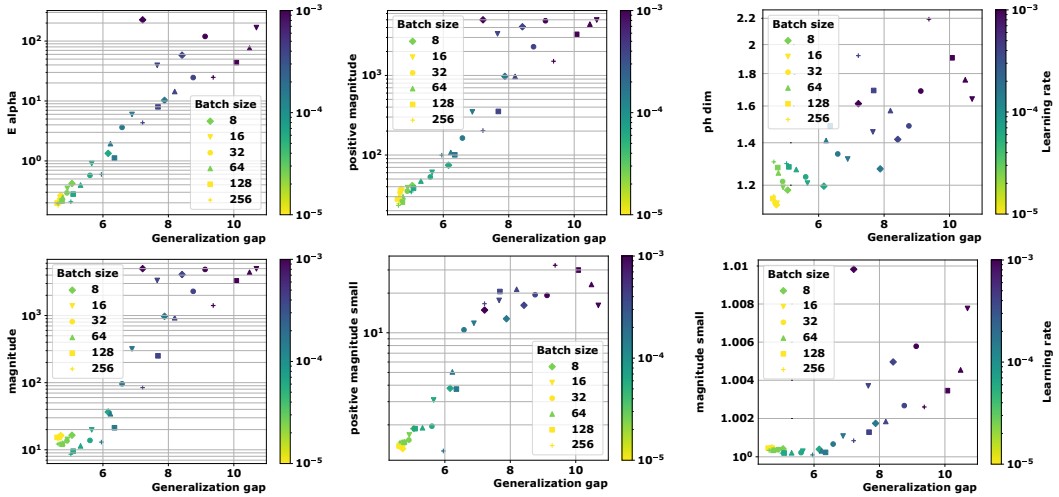

Figure 11: ViT on CIFAR10 with $\rho_S$

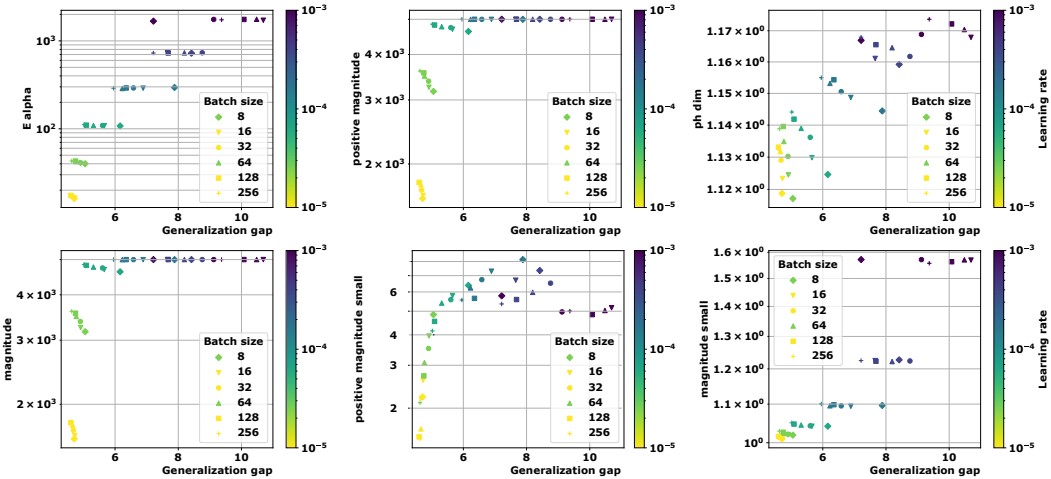

Figure 12: ViT on CIFAR10 with $\|\cdot\|_2$

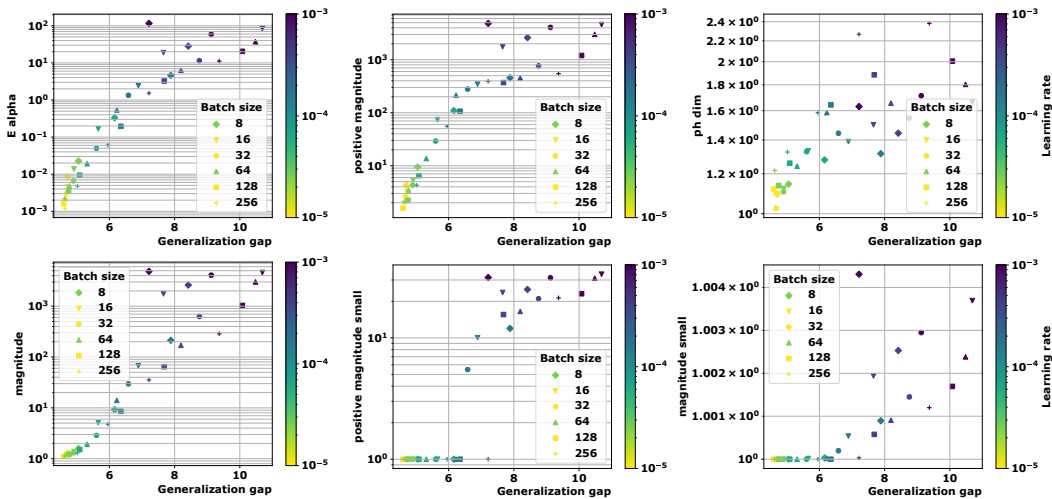

Figure 13: ViT on CIFAR10 with 01-pseudometric

Table 3: Correlation coefficients for all quantities for **ViT model** and **CIFAR100 dataset**. The corresponding plots are presented in Figures 14, Figure 15 and Figure 16.

| METRIC | COMPLEXITY | $\psi_{\mathrm{LR}}$ | $\psi_{\mathrm{BS}}$ | $\Psi$ | $\tau$ |
|---|---|---|---|---|---|
| | $E_\alpha$ | 0.78 | 0.71 | **0.74** | 0.70 |
| | $\mathbf{Mag}(\sqrt{n})$ | 0.78 | 0.71 | **0.74** | **0.72** |
| $\rho_S$ | $\mathbf{Mag}(0.01)$ | 0.15 | 0.11 | 0.13 | 0.17 |
| | $\mathbf{PMag}(\sqrt{n})$ | 0.78 | 0.71 | **0.74** | **0.72** |
| | $\mathbf{PMag}(0.01)$ | 0.60 | 0.62 | 0.61 | 0.56 |
| | $\dim_{\mathrm{PH}}$ [21] | 0.77 | -0.71 | 0.03 | 0.36 |
| | $E_\alpha$ | 0.77 | 0.51 | 0.64 | **0.67** |
| | $\mathbf{Mag}(0.01)$ [3] | 0.77 | -0.69 | 0.04 | 0.50 |
| $\|\cdot\|_2$ | $\mathbf{Mag}(\sqrt{n})$ | 0.77 | -0.45 | 0.16 | 0.54 |
| | $\mathbf{PMag}(0.01)$ | 0.82 | 0.53 | **0.68** | 0.66 |
| | $\mathbf{PMag}(\sqrt{n})$ | 0.78 | -0.45 | 0.16 | 0.54 |
| | $\dim_{\mathrm{PH}}$ [10] | 0.77 | -0.71 | 0.03 | 0.37 |
| | $E_\alpha$ | 0.77 | 0.71 | **0.74** | 0.70 |
| | $\mathbf{Mag}(\sqrt{n})$ | 0.77 | 0.71 | **0.74** | **0.71** |
| 01 | $\mathbf{Mag}(0.01)$ | 0.68 | 0.51 | 0.59 | 0.59 |
| | $\mathbf{PMag}(\sqrt{n})$ | 0.77 | 0.71 | **0.74** | 0.70 |
| | $\mathbf{PMag}(0.01)$ | 0.72 | 0.71 | 0.71 | 0.63 |
| | $\dim_{\mathrm{PH}}$ | 0.73 | 0.02 | 0.37 | 0.57 |

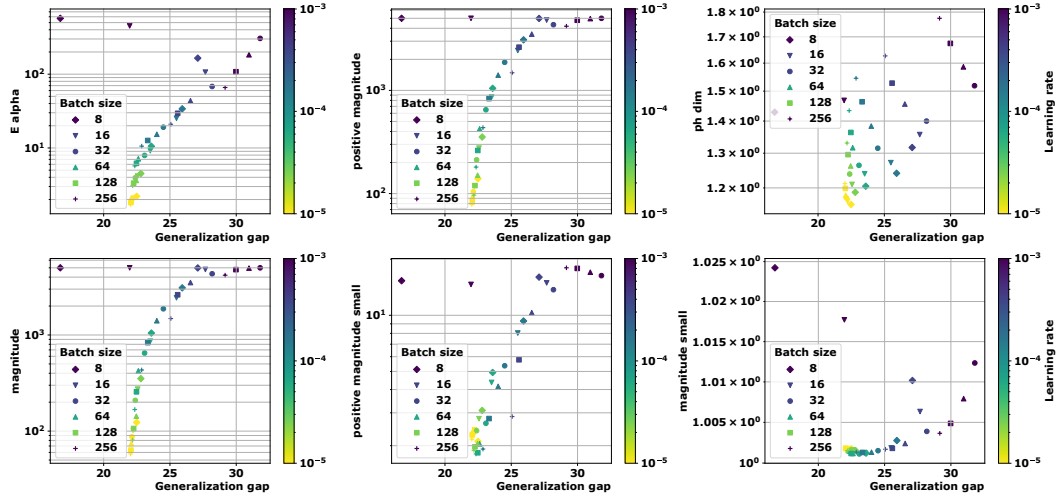

Figure 14: ViT on CIFAR100 with $\rho_S$

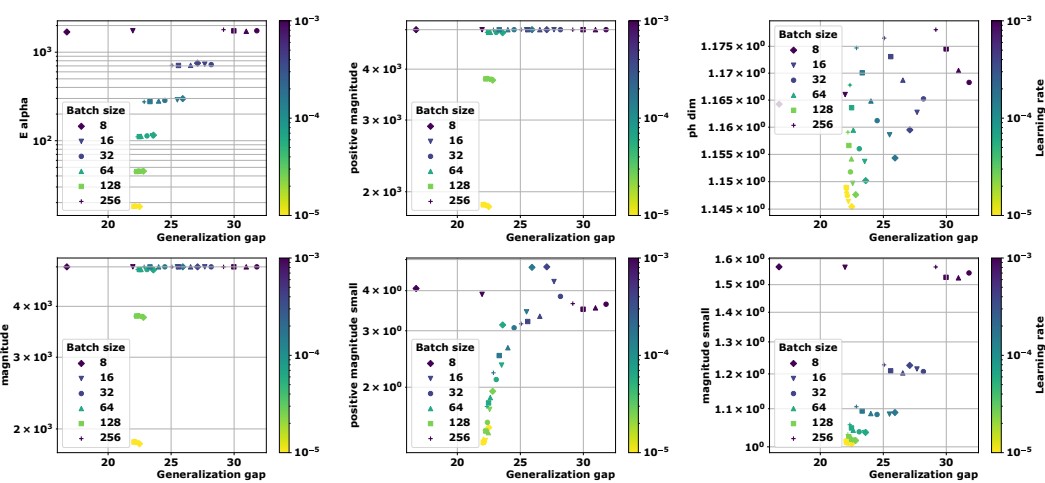

Figure 15: ViT on CIFAR100 with $\|\cdot\|_2$

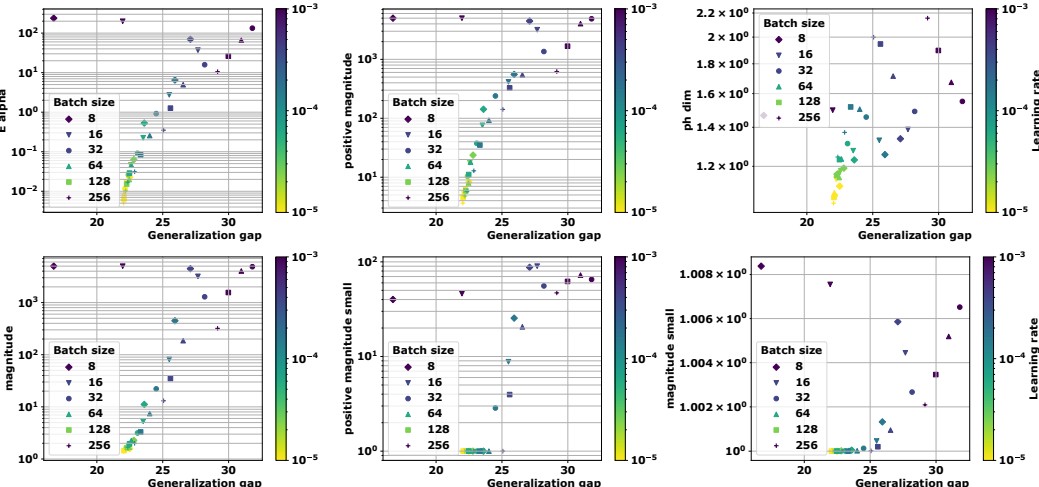

Figure 16: ViT on CIFAR100 with 01-pseudometric

Table 4: Correlation coefficients for all quantities for **CaiT model** and **CIFAR10 dataset**. The corresponding plots can be seen in Figures 17, 18 and 19.

| METRIC | COMPLEXITY | $\psi_{\text{LR}}$ | $\psi_{\text{BS}}$ | $\Psi$ | $\tau$ |
|---|---|---|---|---|---|
| $\rho_S$ | $\boldsymbol{E}_\alpha$ | 0.91 | 0.33 | **0.62** | **0.78** |
| | $\mathbf{Mag}(\sqrt{n})$ | 0.91 | 0.33 | **0.62** | 0.75 |
| | $\mathbf{Mag}(0.01)$ | 0.75 | 0.29 | 0.52 | 0.69 |
| | $\mathbf{PMag}(\sqrt{n})$ | 0.91 | 0.33 | **0.62** | 0.75 |
| | $\mathbf{PMag}(0.01)$ | 0.87 | 0.38 | **0.62** | 0.75 |
| | $\dim_{\text{PH}}$ [21] | 0.91 | -0.19 | 0.36 | 0.75 |
| $\|\cdot\|_2$ | $\boldsymbol{E}_\alpha$ | 0.91 | 0.38 | **0.64** | **0.85** |
| | $\mathbf{Mag}(\sqrt{n})$ | 0.89 | -0.42 | 0.23 | 0.73 |
| | $\mathbf{Mag}(0.01)$ [3] | 0.91 | -0.15 | 0.37 | 0.77 |
| | $\mathbf{PMag}(\sqrt{n})$ | 0.89 | -0.42 | 0.23 | 0.73 |
| | $\mathbf{PMag}(0.01)$ | 0.53 | 0.26 | 0.4 | 0.48 |
| | $\dim_{\text{PH}}$ [10] | 0.91 | -0.31 | 0.30 | 0.67 |
| 01 | $\boldsymbol{E}_\alpha$ | 0.91 | 0.33 | 0.62 | **0.84** |
| | $\mathbf{Mag}(\sqrt{n})$ | 0.91 | 0.33 | 0.62 | 0.77 |
| | $\mathbf{Mag}(0.01)$ | 0.86 | 0.33 | 0.60 | 0.76 |
| | $\mathbf{PMag}(\sqrt{n})$ | 0.91 | 0.33 | 0.62 | 0.79 |
| | $\mathbf{PMag}(0.01)$ | 0.88 | 0.44 | **0.66** | 0.71 |
| | $\dim_{\text{PH}}$ | 0.91 | -0.13 | 0.39 | 0.78 |

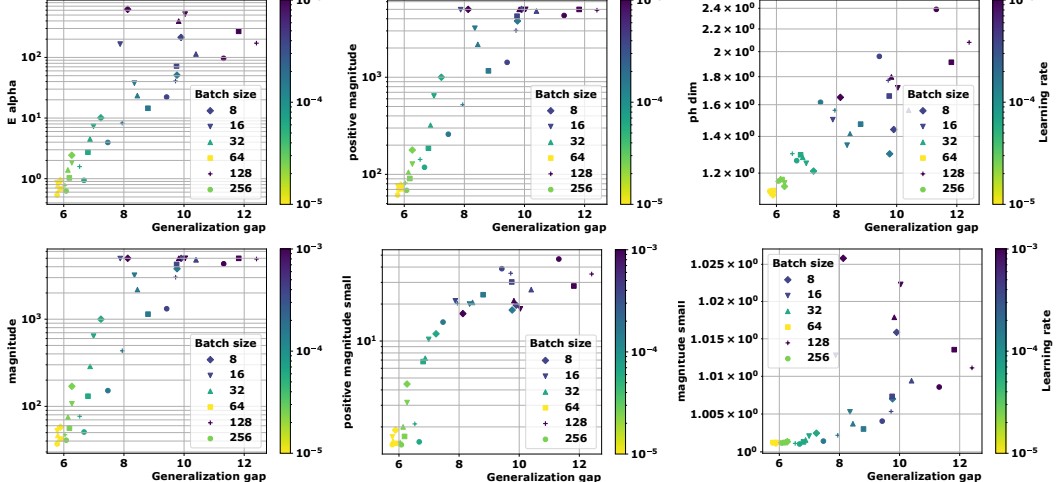

Figure 17: CaiT on CIFAR10 with $\rho_S$-pseudometric.

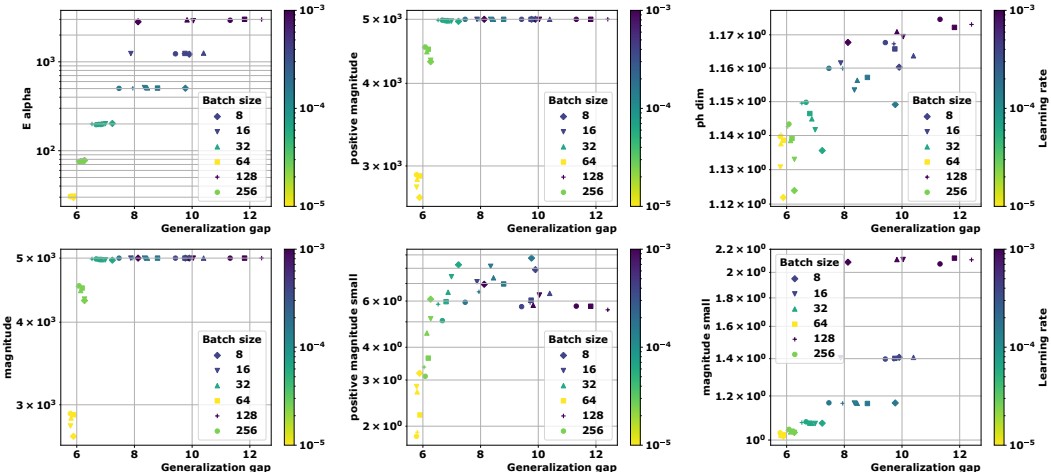

Figure 18: CaiT on CIFAR10 with $\| \cdot \|_2$ distance.

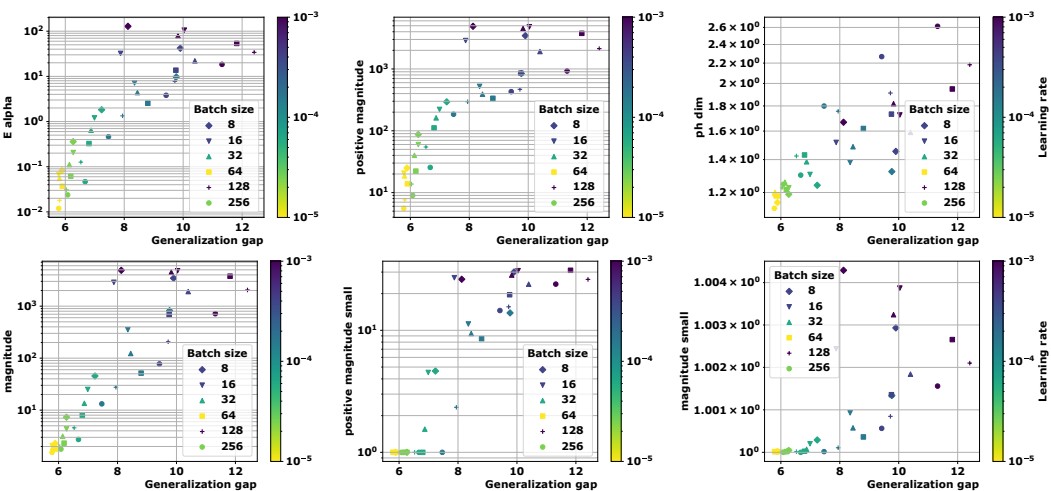

Figure 19: CaiT on CIFAR10 with 01-pseudometric.

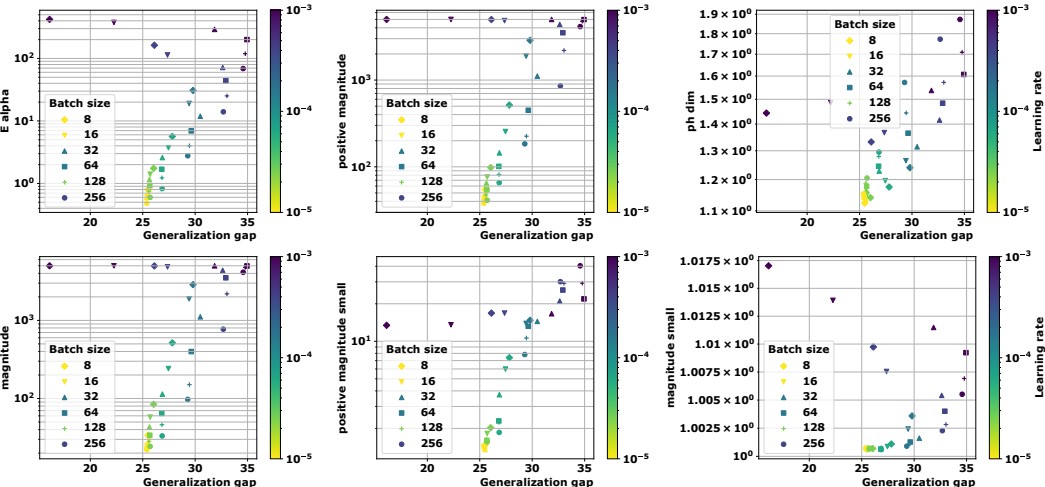

Figure 20: CaiT on CIFAR100 with $\rho_S$-pseudometric.

Table 5: Correlation coefficients for all quantities for **CaiT model** and **CIFAR100 dataset**. The corresponding plots can be seen in 20, 21 and 22

| METRIC | COMPLEXITY | $\psi_{\mathrm{LR}}$ | $\psi_{\mathrm{BS}}$ | $\Psi$ | $\tau$ |
|---|---|---|---|---|---|
| | $E_\alpha$ | 0.67 | 0.13 | 0.40 | 0.54 |
| | $\mathbf{Mag}(\sqrt{n})$ | 0.67 | 0.13 | 0.40 | 0.52 |
| $\rho_S$ | $\mathbf{Mag}(0.01)$ | 0.47 | -0.18 | 0.14 | 0.36 |
| | $\mathbf{PMag}(\sqrt{n})$ | 0.67 | 0.13 | 0.40 | 0.53 |
| | $\mathbf{PMag}(0.01)$ | 0.76 | 0.53 | **0.64** | **0.71** |
| | $\dim_{\mathrm{PH}}$ [21] | 0.67 | -0.13 | 0.27 | 0.56 |
| | $E_\alpha$ | 0.67 | 0.40 | **0.53** | 0.64 |
| | $\mathbf{Mag}(\sqrt{n})$ | 0.68 | 0.33 | 0.50 | **0.65** |
| $\|\cdot\|_2$ | $\mathbf{Mag}(0.01)$ [3] | 0.66 | -0.33 | 0.17 | 0.54 |
| | $\mathbf{PMag}(\sqrt{n})$ | 0.68 | 0.33 | 0.50 | **0.65** |
| | $\mathbf{PMag}(0.01)$ | 0.62 | 0.09 | 0.36 | 0.43 |
| | $\dim_{\mathrm{PH}}$ [10] | 0.64 | -0.09 | 0.28 | 0.50 |
| | $E_\alpha$ | 0.67 | 0.13 | 0.40 | 0.52 |
| | $\mathbf{Mag}(\sqrt{n})$ | 0.67 | 0.13 | 0.40 | **0.57** |
| 01 | $\mathbf{Mag}(0.01)$ | 0.61 | 0.18 | 0.40 | 0.43 |
| | $\mathbf{PMag}(\sqrt{n})$ | 0.67 | 0.11 | 0.39 | 0.53 |
| | $\mathbf{PMag}(0.01)$ | 0.65 | 0.41 | **0.53** | 0.48 |
| | 01 LOSS | 0.58 | 0.07 | 0.32 | **0.57** |

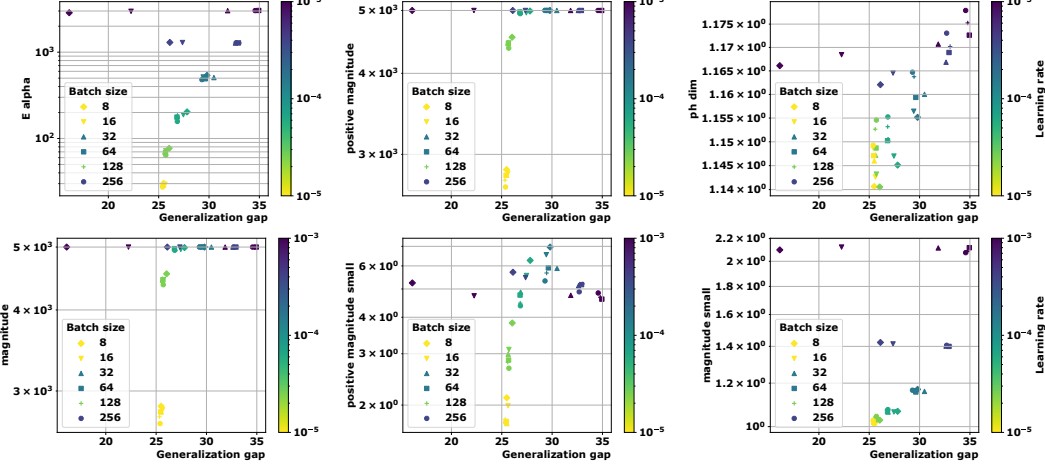

Figure 21: CaiT on CIFAR100 with $\|\cdot\|_2$.

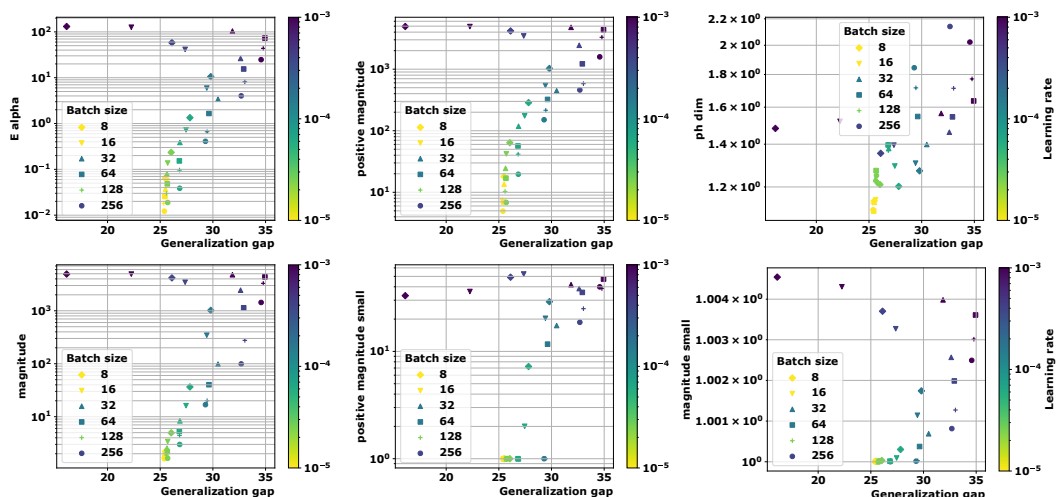

Figure 22: CaiT on CIFAR100 with 01-pseudometric.

Table 6: Correlation coefficients for all quantities for **Swin model** and **CIFAR10**. The corresponding plots are in Figure 23, 24 and 25.

| METRIC | COMPLEXITY | $\psi_{\mathrm{LR}}$ | $\psi_{\mathrm{BS}}$ | $\Psi$ | $\tau$ |
|---|---|---|---|---|---|
| $\rho_S$ | $E_\alpha$ | 0.97 | 0.58 | **0.77** | 0.86 |
| | $\mathbf{Mag}(\sqrt{n})$ | 0.97 | 0.57 | **0.77** | 0.84 |
| | $\mathbf{Mag}(0.01)$ | 0.87 | 0.58 | 0.72 | 0.75 |
| | $\mathbf{PMag}(\sqrt{n})$ | 0.98 | 0.55 | **0.77** | **0.87** |
| | $\mathbf{PMag}(0.01)$ | 0.76 | 0.20 | 0.48 | 0.65 |
| | $\dim_{\mathrm{PH}}$ [21] | 0.97 | -0.57 | 0.19 | 0.67 |
| $\|\cdot\|_2$ | $E_\alpha$ | 0.97 | -0.04 | 0.46 | **0.84** |
| | $\mathbf{Mag}(\sqrt{n})$ | 0.97 | -0.43 | 0.27 | 0.77 |
| | $\mathbf{Mag}(0.01)$ [3] | 0.98 | -0.22 | 0.38 | 0.80 |
| | $\mathbf{PMag}(\sqrt{n})$ | 0.98 | -0.43 | 0.27 | 0.77 |
| | $\mathbf{PMag}(0.01)$ | 0.51 | 0.53 | **0.52** | 0.47 |
| | $\dim_{\mathrm{PH}}$ [10] | 0.95 | -0.57 | 0.18 | 0.69 |
| 01 | $E_\alpha$ | 0.97 | 0.58 | **0.77** | 0.84 |
| | $\mathbf{Mag}(\sqrt{n})$ | 0.97 | 0.58 | 0.77 | **0.86** |
| | $\mathbf{Mag}(0.01)$ | 0.94 | 0.48 | 0.71 | 0.79 |
| | $\mathbf{PMag}(\sqrt{n})$ | 0.98 | 0.58 | 0.78 | 0.87 |
| | $\mathbf{PMag}(0.01)$ | 0.92 | 0.42 | 0.67 | 0.78 |
| | $\dim_{\mathrm{PH}}$ | 0.93 | -0.28 | 0.32 | 0.69 |

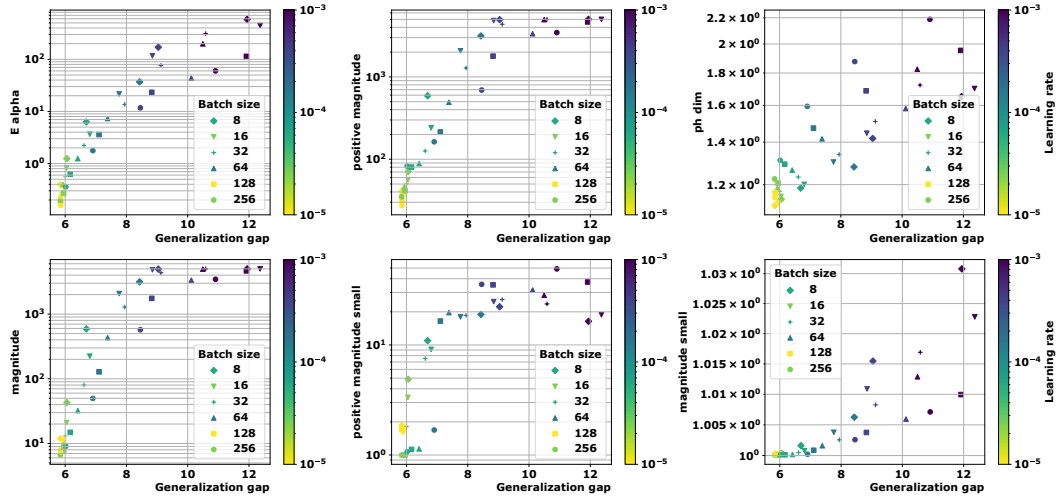

Figure 23: Swin on CIFAR10 with $\rho_S$-pseudometric.

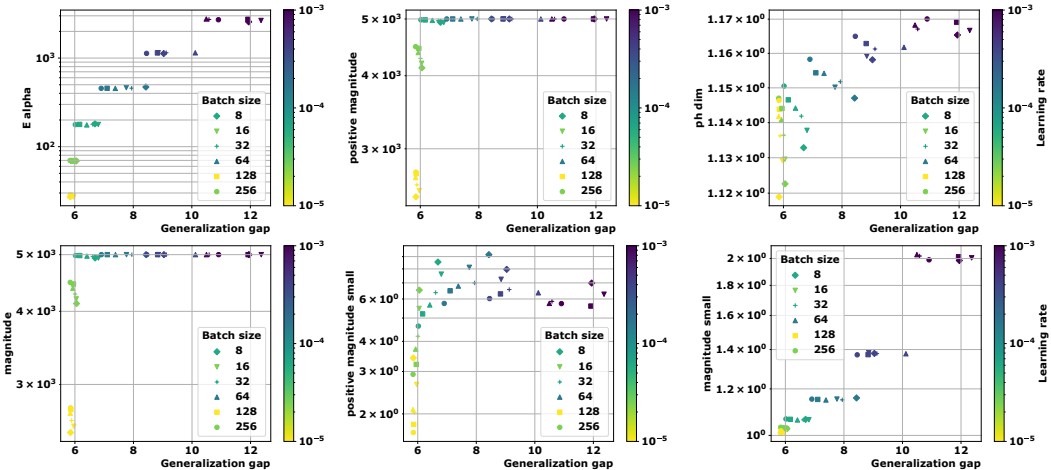

Figure 24: Swin on CIFAR10 with $\| \cdot \|_2$.

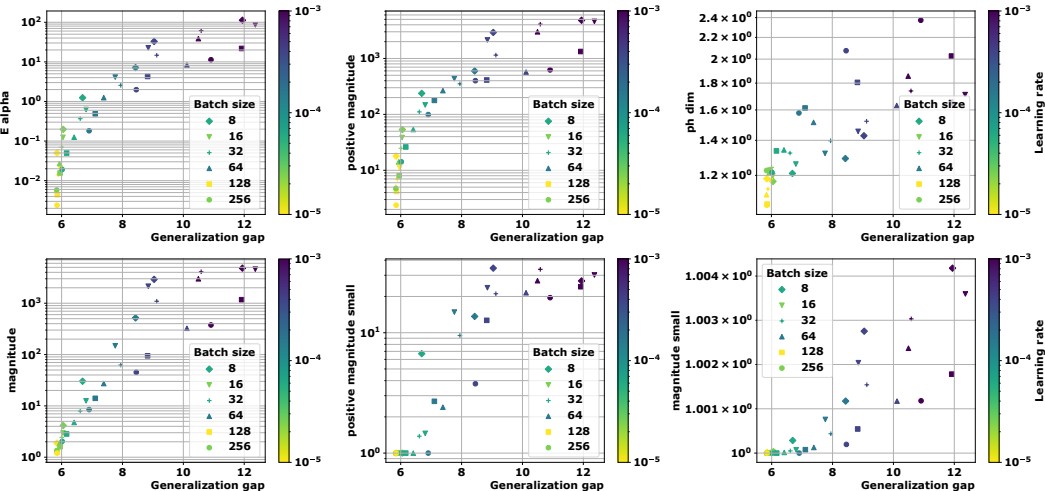

Figure 25: Swin on CIFAR10 with 01-pseudometric.

Table 7: Correlation coefficients for all quantities for **Swin model** and **CIFAR100**. See Figures 26, 27 and 28 for the corresponding plots.

| METRIC | COMPLEXITY | $\psi_{\text{LR}}$ | $\psi_{\text{BS}}$ | $\Psi$ | $\tau$ |
|---|---|---|---|---|---|
| | $E_\alpha$ | 0.69 | 0.47 | 0.58 | 0.62 |
| | $\mathbf{Mag}(\sqrt{n})$ | 0.56 | 0.47 | 0.51 | 0.51 |
| | $\mathbf{Mag}(0.01)$ | 0.31 | 0.47 | 0.39 | 0.33 |
| $\rho_S$ | $\mathbf{PMag}(\sqrt{n})$ | 0.69 | 0.47 | 0.58 | 0.63 |
| | $\mathbf{PMag}(0.01)$ | 0.71 | 0.58 | **0.64** | **0.68** |
| | $\dim_{\text{PH}}$ [21] | 0.69 | -0.47 | 0.11 | 0.50 |
| | $E_\alpha$ | 0.69 | 0.22 | 0.46 | 0.63 |
| | $\mathbf{Mag}(\sqrt{n})$ | 0.71 | -0.57 | 0.07 | 0.53 |
| | $\mathbf{Mag}(0.01)$ [3] | 0.69 | -0.44 | 0.12 | 0.53 |
| $\|\cdot\|_2$ | $\mathbf{PMag}(\sqrt{n})$ | 0.71 | -0.57 | 0.07 | 0.53 |
| | $\mathbf{PMag}(0.01)$ | 0.64 | 0.51 | **0.58** | 0.46 |
| | $\dim_{\text{PH}}$ [10] | 0.69 | -0.47 | 0.11 | 0.45 |
| | $E_\alpha$ | 0.69 | 0.47 | **0.58** | 0.61 |
| | $\mathbf{Mag}(\sqrt{n})$ | 0.69 | 0.47 | **0.58** | **0.62** |
| | $\mathbf{Mag}(0.01)$ | 0.61 | 0.27 | 0.44 | 0.50 |
| 01 | $\mathbf{PMag}(\sqrt{n})$ | 0.69 | 0.47 | **0.58** | **0.62** |
| | $\mathbf{PMag}(0.01)$ | 0.65 | 0.49 | 0.57 | 0.54 |
| | $\dim_{\text{PH}}$ | 0.64 | 0.04 | 0.34 | 0.51 |

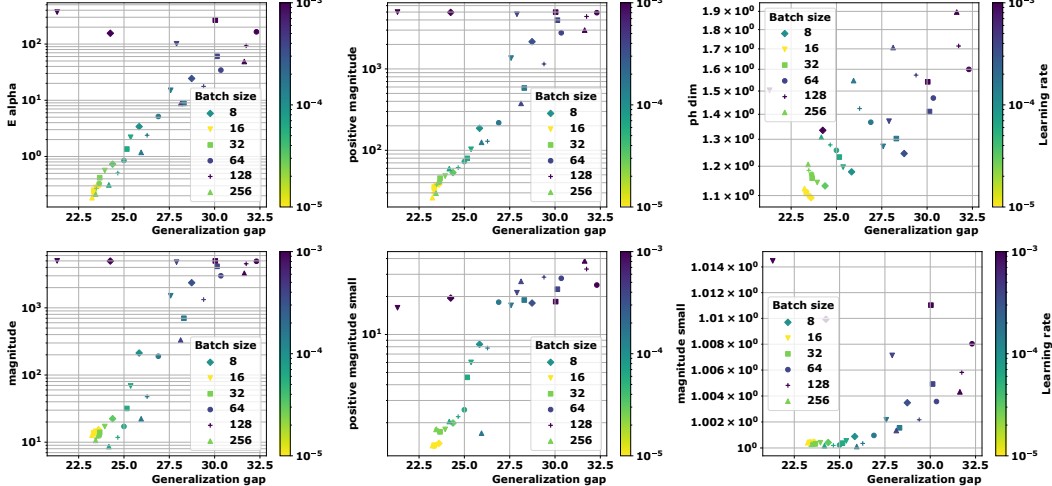

Figure 26: Swin on CIFAR100 with $\rho_S$-pseudometric.

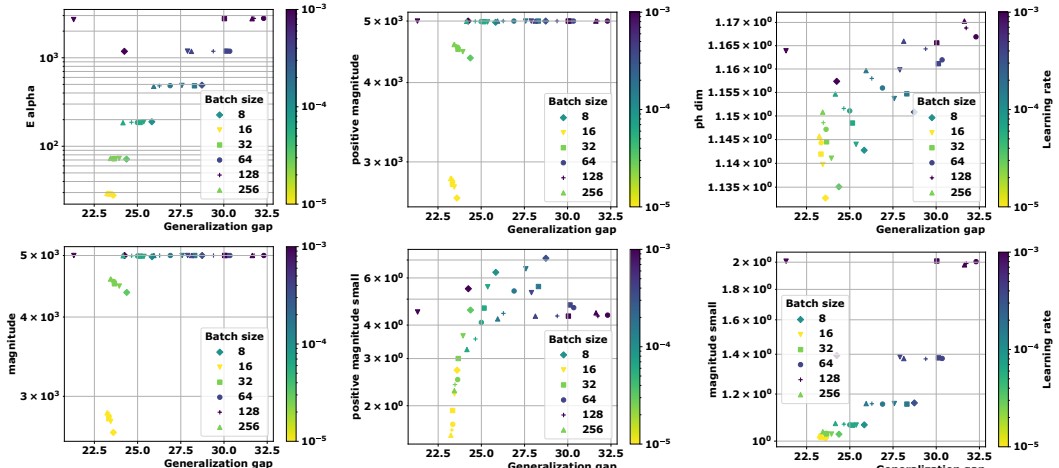

Figure 27: Swin on CIFAR100 with $\|\cdot\|_2$.

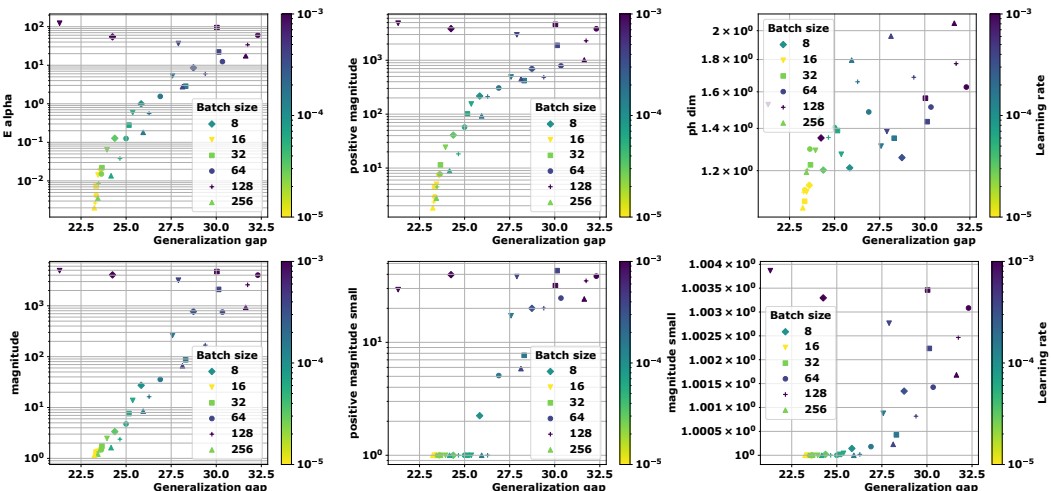

Figure 28: Swin on CIFAR100 with 01.

## D.4 Graph Neural Networks – Additional Experiments

In Table 1, we already presented the correlation coefficients for all quantities for the GNN models considered in our study (GraphSage, GatedGCN) [23] (we have selected the models which achieve 100% training accuracy)) and Graph-MNIST. We can observe a nice correlation, outperforing dim-PH in most experiments. As it was observed for the transformer-based experiments, the correlation seems to be better for the data-dependent-metrics. This is an important fact, as no sparse random projection was used to compute the Euclidean distance matrices in the GNN experiments (it was not necessary as these models have less parameters than the tramsformers considered above). This shows that the fact the data-dependent pseudometrics outperform the Euclidean distance also happens in the absence of these projections. It also shows that all quantities seem to yield better correlations in the absence of random projections, at least in the GNN expsriments.

The corresponding plots for GatedGCN can be seen in Figure 32 with the pseudometric, Figure 33 for the Euclidean and 34 for 01. The plots for GraphSage are reported in Figure 29, Figure 30 and Figure 31.

We can observe a strong correlation on these figures, outperforing dim-PH in most cases. As it was observed for the transformer-based experiments, the correlation seems to be better for the data-dependent-metrics. This is an important fact, as no sparse random projection was used to compute the Euclidean distance matrices in the GNN experiments[9]. This shows that data-dependent pseudometrics outperform the Euclidean distance also in the absence of these projections. In addition, all quantities seem to yield better correlations in the absence of random projections, at least in the GNN expsriments.

Interestingly, a few failure cases can be seen on these plots. Indeed, $\mathbf{Mag}(0.01)$ and $\mathbf{PMag}(0.01)$ seem to be almost constant and near 1. This indicates that the scale choice $s = 0.01$ was not suited for these experiments; this behavior was already reflected in Table 1 through very low Kendall's coefficients, indicating the absence of meaningful correlation. However, $\mathbf{Mag}(\sqrt{n})$ and $\mathbf{PMag}(\sqrt{n})$ provide significantly better correlation, which supports our main claims, as $s = \sqrt{n}$ has been argued in section 3.3 to be a particulary relevant choice of scale factor.

Note finally that the PH-dim plots for the 01-pseudometric failed to produce numbers in these graphs experiments (this is why they are either missing or look irrelevant). As before, we gave away this fact in Table 1 by imposing our granulated Kendall's coefficients implementation to return zeros in the absence of correlation, hence the small numbers observed in this case. That being said, this behavior should not be seen as an issue. Indeed, PH-dim with 01-pseudometric consists (in theory) in estimating the dimension of a subset of a discrete hypercube, which is always 0. The reason we still reported PH-dim for this pseudometric is for consistence and to test the implementation of [10, 21] in this non-standard setting; it is however not theoretically grounded.

---

[9]A sparse random projection was not necessary as these models have less parameters than the tramsformers considered above

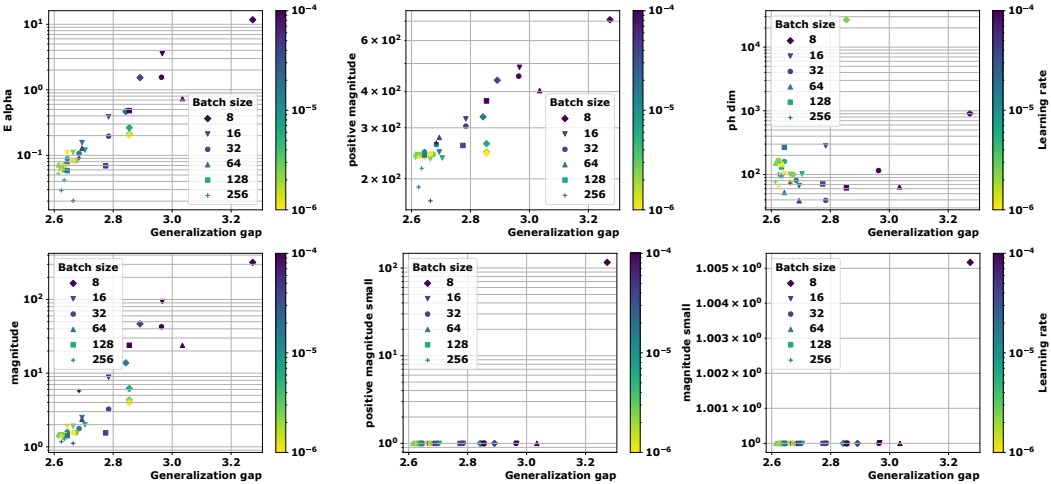

Figure 29: GraphSage on MNIST with $\rho_S$-pseudometric.

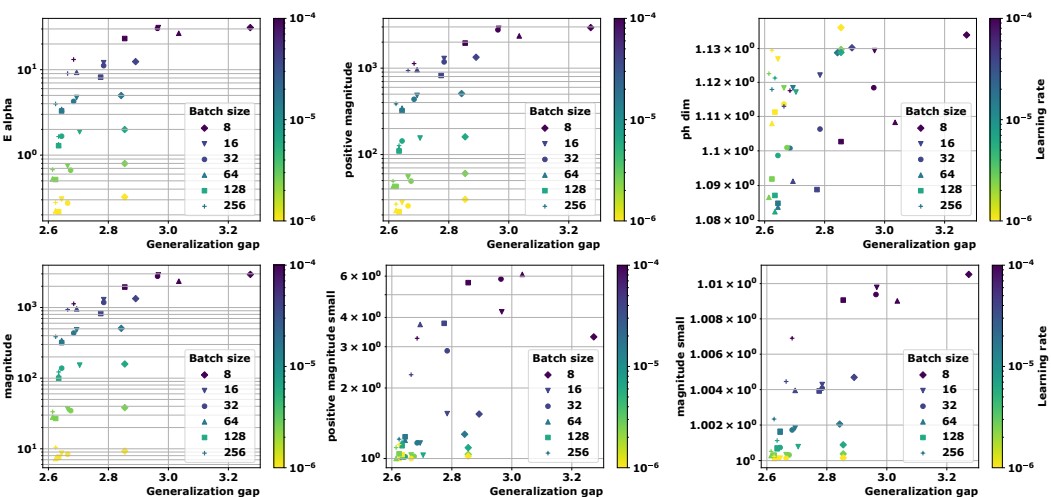

Figure 30: GraphSage on MNIST with $\| \cdot \|_2$.

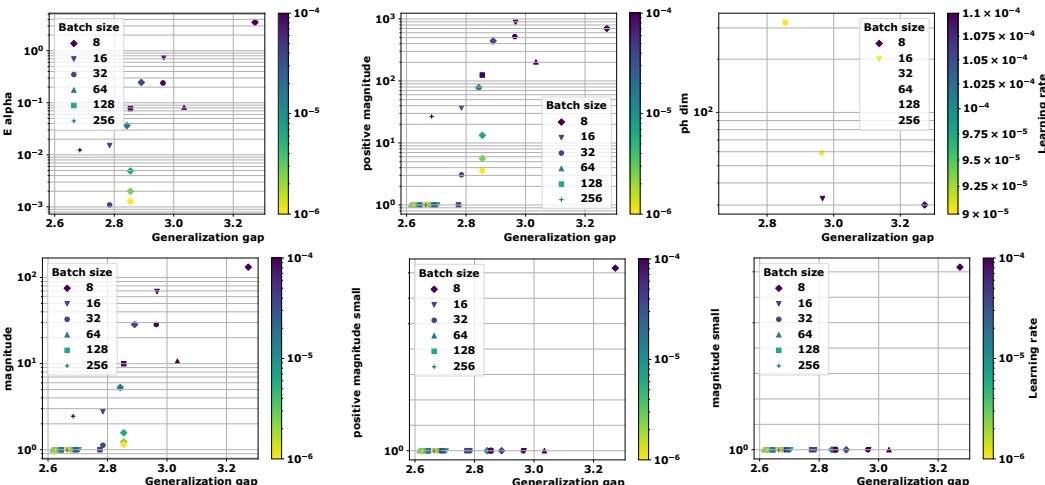

Figure 31: GraphSage on MNIST with 01.

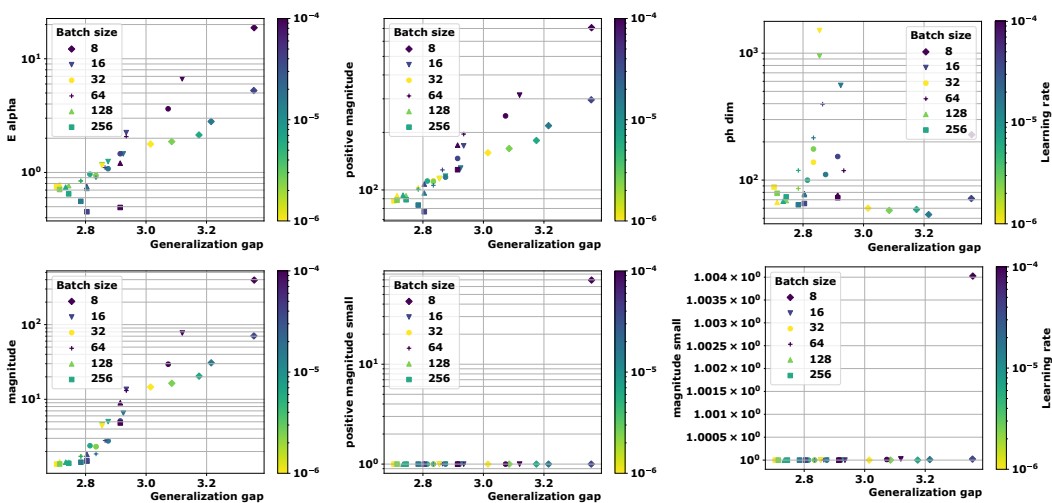

Figure 32: GatedGCN on MNIST with $\rho_S$-pseudometric.

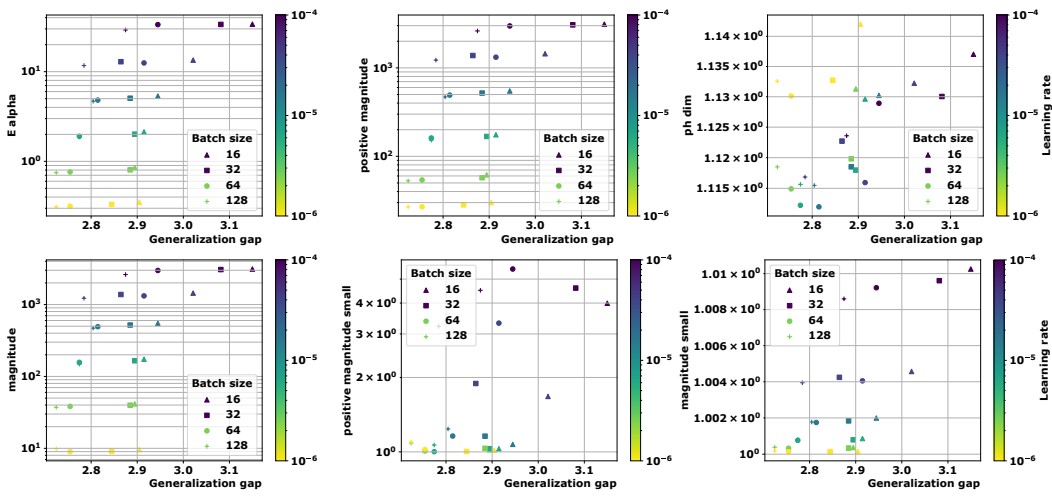

Figure 33: GatedGCN on MNIST with $\| \cdot \|_2$.

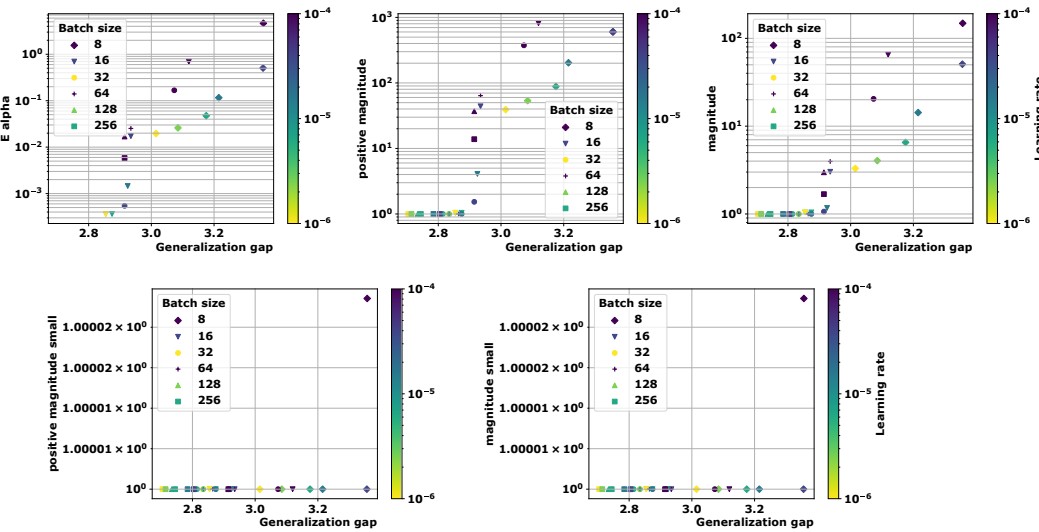

Figure 34: GatedGCN on MNIST with 01.

