# OpenReview forum: "Topological Generalization Bounds for Discrete-Time Stochastic Optimization Algorithms"
_NeurIPS.cc/2024/Conference — NeurIPS 2024 poster_

### Official Review · Reviewer_bXXR · 2024-06-13

**Soundness:** 2
**Presentation:** 2
**Contribution:** 3
**Rating:** 6
**Confidence:** 3

**Summary:**

This paper proposes a method to upper-bound the generalization gap $G(w)$ (discrepancy between the empirical risk and the theoretical risk for a given parameter $w$ of our model) by tractable _topological_ quantities.

Namely, given a set $W = \{w_\tau,\dots,w_T\}$ (typically, a sequence of iterates for a SGD between iterations $\tau$ and $T$, where $w_\tau$ is already near a local minima of the empirical risk), the authors derive bounds of the form

$$\sup_{w \in W} G(w) \leq \sqrt{\frac{\text{Topological quantity } + \text{(Information theory quantity)} + \log(\zeta^{-1})}{n}}$$

with probability $1 - \zeta$. The work focuses on building actual topological quantities that make this claim true, proposing two possible candidates:
- The _$\alpha$-weighted lifetime sums_ that (roughly) computes the maximum spanning tree of $W$ and then assign a weight to it,
- The _positive magnitude_ that (roughly) computes the positive mass of $\beta = K_s^{-1} \mathbb{1}$, where $K_s$ can be understood as a Gaussian/Laplace kernel on $W$ for some distance $\rho$ and bandwith $s$ (and $\mathbb{1}$ is a vector full of $1$s).

Contrary to previous similar works, their theoretical results directly hold in (mostly) practical situations: they do not require observing a time-continuous trajectory and the quantities involved are (at least theoretically) directly computable (in practice, this requires approximation).

Experimentally, the authors observe through an extensive set of experiments that the upper-bound they propose performs better, in terms of correlation with the actual generalization gap, than the other kind of topological upper-bound previously introduced.

**Strengths:**

The work proposes an original and promising approach to tackle an important question.

Contrary to most previous works, the experiments go way beyond "toy-models", and the results are reported on modern architectures like transformers, showing that the method is usable in practice on "real" setups. Experimental results are extensively reported, supporting the reproducibility of the work.

The presentation of related works / comparison of SotA (with which I am not completely familiar) seems to be quite comprehensive.

**Weaknesses:**

In my opinion, the main weakness is that the (main) paper is kinda far from being self-contained and it is somewhat hard to get an intuitive grasp on the proposed approach. The appendix (about 40 pages!) contains crucial information (proofs, missing definitions, etc.) that couldn't be carefully investigated (*).

While the contribution itself is seemingly solid, this prevents me from claiming that the work is theoretically sounded and limit the understanding if one stick to reading the main paper. For instance, if one looks at Theorem 3.4, I cannot understand the role of $\tau$ and $T$ (as far as I can tell, they only appear in $I_\infty$ in the rhs, but since this term is not defined I do not know what happend if, say, $T = \ŧau$ or $T = +\infty$). Similarly, without reading the proofs, I have no clue on why taking $\alpha = 1$ should be natural, what is the role played by the constant $q$ in the $(q,L,\rho)$-Lipschitz assumption, etc.

I understand that being comprehensive is impossible given the space limitation, but I believe that it is part of the contribution to produce an accessible less-than-10-pages-long presentation of the work.

(*) Note that I quickly parsed the supplementary material, which I found quite enlightening, but cannot guarantee the correctness of its content.

**Questions:**

1. Could you give (for, say, theorem 3.4) a sort of "sketch of proof" and more intuition on "why this should be true"?

2. Related to above, in Thm 3.4 (and 3.5), the r.h.s. depends on many parameters on which the l.h.s. doesn't. For instance, it seems that I could pick a different $\rho$ or $\alpha$ without affecting the l.h.s. Is that correct? Could I just try to find a $\rho$ that makes $E_\alpha^\rho$ close to $1$ in order to improve my bound?

3. Still in Thm 3.4, would it be in some sense possible that we pick $\alpha$ below the PH-dim of (roughly speaking) the possible support of the random set $W$, in which case (here again, this is informal) it may be possible that $E_\alpha$ is unbounded?

**Limitations:**

Limitations have been discussed by the authors, which is appreciated.

---

> ### Author Rebuttal · Authors · 2024-08-06
>
> We thank the reviewer for their time spent on the paper and their insightful comments.
>
> We address below the main questions (alongside weaknesses).
>
> **Could you give (for, say, theorem 3.4) a sort of "sketch of proof" and more intuition on "why this should be true"?**
>
> First, we would like to point our that even though the appendix is long, most of it is dedicated to additional empirical results and figures, while the proofs are a relatively short portion of the appendix.
>
> Including a sketch of proof is a very good suggestion from the reviewer. In the next version of the paper, we now include a sketch of proof of Theorem 3.4. The general idea is the following: we build on Corollaries 26 and 27 of the reference [18] of our paper, to get a bound of the worst-case generalization error as (this is an informal statement, in particular we omit absolute constants and less relevant terms, refer to the paper for more details):
> $$
> G \lesssim \sqrt{\frac{N_\delta}{n}} + \sqrt{\frac{\mathrm{IT}}{n}},
> $$
> where $N_\delta$ is a covering number of the trajectory and $\mathrm{IT}$ is an information-theoretic term. First, we use lemma 16 of [11] to show that $\mathrm{IT}\leq I_\infty$, where $I_\infty$ is the mutual information term appearing in Theorem 3.4. Second, we bound $N_\delta$ in terms of packing numbers, denoted $P_\delta$ (see Definition A.15), which we further bound in terms of $E_1$ (or in general $E_\alpha$ for $\alpha \leq 1$) using a geometric construction displayed on Figure 2.b (rebuttal pdf). We have now included this figure to the paper to provide more intuition on the proof technique, which is based on this geometric observation.
>
> For theorem 3.5, we can mention that the key difference is Lemma B.13, whose proof is short but contains the essential ideas behind this theorem. It relies on properties of magnitude that are detailed in the appendix.
>
>
> **Related to above, in Thm 3.4 (and 3.5), the r.h.s. depends on many parameters on which the l.h.s. doesn't. For instance, it seems that I could pick a different $\rho$ or $\alpha$ without affecting the l.h.s. Is that correct? Could I just try to find a $\rho$ that makes $E_\alpha^\rho$ close to $1$ in order to improve my bound?**
>
>  It is true that some of our bounds depend on free parameters.
> As noted by the reviewer, this is in particular true for the constant $\alpha \in [0,1]$ and the pseudometric $\rho$, as soon as $\rho$ satisfies the regularity condition given by Definition 3.1. Let us discuss these two quantities separately.
>
> > Choice of $\alpha$
>
>  First, regarding the choice of $\alpha$, the only pertinent choice in general seems to be $\alpha=1$, which is what we choose in our experiments. The reason why this choice is natural is that the $\alpha$-weighted lifetime sums tend to be higher when $\alpha$ gets smaller. In the case where the random set is finite, $E_0$ even corresponds to the cardinality of the set and does not capture any meaningful topological behavior (moreover, it goes to infinity when the size of the set goes to infinity). The reviewer may consult [61] for better intuition about the behavior of this quantity.
>
> > Choice of $\rho$ and $q$
>
>  Second, for the choice of $\rho$, the possibilities might not always be numerous. Indeed, $\rho$ must satisfy a $(q,L,\rho)$ condition as in Definition 3.1. The choice of $\rho$ impacts the bound both by affecting the topological quantity ($E_\alpha^\rho$ in the case of Theorem 3.4) and the value of $q$ and $L$. It should be noted that the pseudometrics $\rho_S^{(q)}$ (with $q\geq 1$) appear in Example 3.2 always satisfy the $(q,1,\rho_S^{(p)})$-Lipschitz condition. In that particular case, it seems that choosing $\rho_S := \rho_S^{(1)}$ is the best choice because of Hölder's inequality: $\forall q \geq 1,~ \rho_S \leq \rho_S^{(q)}$. The fact that $q=1$ is a good choice is particularly apparent in the proof of Theorem 3.5, for this exact reason. When more pseudometrics are admissible, the reviewer is right to notice that there could be an optimal choice giving the best bound. In the final version of the paper, we discuss in more detail the choice of these parameters.
>
> > I cannot understand the role of $\tau$ and $T$
>
> The meaning of these parameters is that we collect the training trajectory between iterations $\tau$ and $T$. While it is technically possible to chose $\tau=T$, it means that the trajectory would be reduced to a single point and therefore it does not contain any interesting topological information. Similarly, taking $T\to\infty$ seems to be possible under mild assumptions, even though it is not our focus.
>
> Note that our topological complexities also depends implicitly on $\tau$ and $T$, as they are evaluated on the trajectory between iterations $\tau$ and $T$.
>
>  **Still in Thm 3.4, would it be in some sense possible that we pick $\alpha$ below the PH-dim of (roughly speaking) the possible support of the random set $W$, in which case (here again, this is informal) it may be possible that $E_\alpha$ is unbounded?**
>
>  This is a pertinent question from the reviewer. In the case that is of more interest for our experiments, the random set $W$ is finite, which makes the $\alpha$-weighted lifetime sums bounded for any value of $\alpha$. That being said, if one extends our theory to infinite sets (which is technically possible for most aspects, even though some statements should be reformulated), the value of $\alpha$ should indeed be carefully chosen with respect to the dimension of the set. Even in our experimental setting, if the size of the set goes to infinity, it might be pertinent to take into account the dimension of the "limiting set" to tune the value of $\alpha$, keeping in mind that it must satisfy $\alpha \in [0,1]$. However, this setting is largely theoretical as the sets are large but finite in practice. In the final version of the paper, we added a discussion about this interesting question from the reviewer.

---

> > ### Comment · Reviewer_bXXR · 2024-08-08
> > **Thanks**
> >
> > Thank you for taking time answering my review.
> >
> > As i said, my main concern with the work is that it contains a lot of technical content deferred to the appendix that I could not review in detail. Otherwise I'm fine with it; so I must discuss with point with the AC and other reviewers in the next discussion period.

---

> > > ### Author Response · Authors · 2024-08-11
> > >
> > > We understand that the reviewer finds the appendix to be important for the grasp of our paper. So, we would like to clarify its content.  First, let us highlight that the appendix is 26 pages long, and not 40 as claimed by the reviewer.
> > > 1) Having proofs in the appendix (Appendix B) is standard for NeurIPS papers and helps readability of our paper. The grasp of the proofs are not needed for the grasp of the ideas in our paper.
> > > 2) Our exposition of the persistent homology (Appendix A) should not be seen as a set of missing definitions but rather as a little treatise of the background material included for making our paper self-contained. We could have cited the relevant resources only, but we instead opted for including the relevant content from those references as a small section. We hope that such non-crucial sections can help the uninformed reader for understanding our paper.
> > > 3) The remaining part of the appendix (Appendix C) is dedicated to experimental setting details and additional empirical results. These experiments mainly support the empirical findings reported in the main paper, and provide detailed analyses of the content reported in the main paper.
> > >
> > > In the final version of the paper, we will use the extra page to include, in the main paper, a summary of the main ideas contained in the appendix, in particular a sketch of proof and comments on the additional experiments. We hope that the reviewer reconsiders their evaluation in the light of these.

---

> > > > ### Comment · Reviewer_bXXR · 2024-08-12
> > > > **ty**
> > > >
> > > > > the appendix is 26 pages long, and not 40 as claimed by the reviewer.
> > > >
> > > > Yes indeed I must admit that I went for the lazy evaluation "length of the pdf - main body = 40 pages", not accounting for references, neurips checklist etc. The authors are right, the appendix is shorter than what my initial comment suggest, and the section dedicated to delayed technical proofs is "only" 8 pages long.
> > > >
> > > > Nonetheless, it is important to confess that I did not investigate these proofs in detail and, as such, cannot guarantee the correctness of the work. I appreciate that the authors took time to explain them in their rebuttal and I am leaning toward supporting the paper (which is reflected by my positive note and average confidence).

---

> > > > > ### Author Response · Authors · 2024-08-12
> > > > > **Thank you**
> > > > >
> > > > > We are grateful to the reviewer for their positive feedback. As we mentioned in the rebuttal, we will improve the clarity of the paper in the final version by implementing the changes we discussed.

---

### Official Review · Reviewer_4cSo · 2024-07-01

**Soundness:** 3
**Presentation:** 2
**Contribution:** 4
**Rating:** 7
**Confidence:** 2

**Summary:**

Prior work has sought to provably bound the generalization of a neural network based on a complexity measures, eg using some form of evaluation of mutual information between the data and the training path, however such proofs have relied on the topologies from the asymptotic infinite training case and other impractical assumptions. This paper seeks to extend the provable generalization bounds to a practical training regime with discrete time steps. They attempt to make such complexity measures more tractable by, rather than a full derivation of those intrinsic dimension measures utilized in prior works, looking to leverage instead those dependent underlying variables on which intrinsic dimension is based, which are more accesible during training, resulting the the measures of "alpha weighted lifetime sums" and "magnitude".

In addition to the forms of experimental validation comparing the complexity measures to a resulting generalization gap, the scope of the paper includes extensive theoretical discussions that are slightly beyond the competence of this reviewer to fully evaluate, and so this review should be considered as taking much of these aspects at "face value". That being said, relying on assumption of rigor in such aspects, I believe the scope of the paper and implications of the claims would likely merit some form of recognition from the conference, like an oral presentation or spotlight.

**Strengths:**

- Originality
I have seen some manner of complexity measures as a form of performance metric discussed in prior work, including things like sharpness measures, however I have yet to see anyone claim bounded forms of generalization guarantees accessible during intermediate stages of training, suggesting that this could be a significant improvement towards such applications.

    - Quality
The paper was ambitious in scope and for the most part appeared dense in significance.

    - Clarity
Some of the benchmarking was more intuitive as the complexity compared to generalization charts as opposed to the Table 1 for instance, I don't know if that could be simplified in some fashion (sometimes less information is better and save the half page of numbers to the appendix for instance).

    - Significance
Taken at face value the availability of a provable form of generalization guarantee efficiently available in real time during training is potentially quite significant towards mainstream practice.

**Weaknesses:**

It is hard be this reviewer to fully assess the theoretical merit of the derivations, and hope that some of the other reviewers may be more competent in this sense.

**Questions:**

Can you further clarify your use of the term "connected components"?
Should we interpret that as the number or ratio of weights impacted by a gradient update step?
Am I interpretting correctly that with increased connectivity we could expect improved generatlization or is that too simplistic? (After all for the neural tangent kernel at infinite width wouldn't all weights be updated with each gradient step?)

Do you expect that the generalization bounds are asymptotic towards scale of foundation models or can such assumptions be comparably extended to models of limited scope and scale?

Can you clarify how training may be conducted in this form, eg for a simple supervised learning setup, am I correct in assuming that it would it be possible to replace the performance metric but the labels corresponding to training data still be required?

**Limitations:**

It would be hard to fully validate the claims of "provable generalization bound" without some more significant survey of the appendices, which are outside the competence of this reviewer.

---

> ### Author Rebuttal · Authors · 2024-08-06
>
> We thank the reviewer for their time and their insightful comments on our work.
>
> > Some of the benchmarking was more intuitive as the complexity compared to generalization charts as opposed to the Table 1 for instance, I don't know if that could be simplified in some fashion
>
> We believe that Table 1 is quite informative (though compact) as it summarizes a lot of our empirical findings. In order to provide more visual representations of our results, we provided in Fig. 4.c a graphical representation of the performance of our topological complexities in the plane $(\psi_{\mathrm{bs}},\psi_{\mathrm{lr}})$. This figure contains part of the information of Table 1 along with additional pair (model, dataset). Fig. 1.c is also a graphical representations of the values of $\Psi$ reported in the table. These two figures therefore contain most of the information from the table. In the next version of the paper, we have completed them by additional plots similar to Fig. 1.c (see Fig. 2.a in the rebuttal pdf for an example). Note moreover that a wide range of plots in the appendix represent the topological complexities against the generalization error.
>
> > It is hard be this reviewer to fully assess the theoretical merit of the derivations
>
> In order to improve the clarity of the paper, we now added to the main text (thanks to the additional page) a sketch of proof for our main results in the next version of the paper.
>
> > Can you further clarify your use of the term "connected components"?
>
> Indeed, our use of the term "connected components" in Section 2 might not have been clear enough. It should be understood as follows: given a distance parameter $\delta>0$ and a point cloud $W$ (which could be one of our training trajectories for instance), we can construct a graph by connecting each pair of points at distance at most $\delta$ and count the number of connected components of the obtained graph. What we meant by "tracking the “connected components” of a finite set at different scales" consists in observing the evolution of these connected components when varying $\delta$. While this is just an informal introduction to the concept of persistent homology, we provide in Appendix A a more formalized description. In the next version of the paper, we are now more precise about it.
>
> > Should we interpret that as the number or ratio of weights impacted by a gradient update step? Am I interpretting correctly that with increased connectivity we could expect improved generatlization or is that too simplistic?
>
> The reviewer is right in noticing that if only a ratio of weights are affected by the gradient steps, then the trajectory will be lower dimensional, which might make our topological complexities smaller. However, this is only a particular case and a too simple view of what is happening during training. The topological properties of the trajectory are the consequence of a complex iterative algorithm (see for instance reference [11] in the paper), and capture information that is more complicated than the ratio of updated weights.  The emergence of connected components is considered in a complex recursive way, it is not related to the ratio of weights that are updated. Finally, the NTK is a specific limit with a particular scaling scheme, we do not believe that it can be compared to the observed behavior in our paper.
>
> > Do you expect that the generalization bounds are asymptotic towards scale of foundation models or can such assumptions be comparably extended to models of limited scope and scale?
>
> While our work was the first to compute topological generalization measures on neural architecture with real practical interest, applying the same procedures on large language models or other foundation models would still be far too computationally expensive.
>
> However, one could imagine a setup where neural networks are embedded in a lower dimensional representation and compute the topological quantities associated with the trajectories in this lower dimensional space. While additional theory would be necessary to understand the true impact of this compression procedure, this might open the door to the computation of topological measures for foundation models. This could be a direction for future research.
>
> > Can you clarify how training may be conducted in this form, eg for a simple supervised learning setup, am I correct in assuming that it would it be possible to replace the performance metric but the labels corresponding to training data still be required?
>
> Regardless of the training procedure, the dataset, and the model, we need two things to compute our topological complexities: a training trajectory and a pseudometric satisfying the condition of Definition 3.1. If the metric is chosen carefully, it could theoretically be applied to any large model as soon as suitable computational resources are available. An interesting setting happens when a large model (eg, a foundation model) is finetuned on a dataset different from the dataset used for pretraining. In that case, to compute our topological complexities, one would only need the dataset used for the fine-tuning procedure and the training data for the inital large model would not be needed.

---

> > ### Comment · Reviewer_4cSo · 2024-08-08
> >
> > Thank you for response. I retain my recommendation for accept pending any discussions with other reviewers.

---

### Official Review · Reviewer_9Fve · 2024-07-08

**Soundness:** 3
**Presentation:** 3
**Contribution:** 3
**Rating:** 7
**Confidence:** 4

**Summary:**

- The authors provided a novel topological-complexity-based uniform generalization error bound, constructed on the $\alpha$-weighted lifetime sums or positive magnitude. This bound shows better correlation with the generalization error compared to existing bounds.
- The authors proposed an implementation scheme based on dissimilarity measures between neural networks, enabling the quantification of generalization across different model architectures without the need for domain or problem-specific analysis.
- The authors confirmed that their topological complexity term exhibits better correlation with the actual generalization gap in large-scale neural networks, such as ViT and GNN.

**Strengths:**

- More Realistic Generalization Error Bound
  - This paper addresses a significant limitation of previous topological complexity-based generalization error bounds, which only hold under discrete parameter updates based on SGD or Adam, or when the number of iterations is taken to infinity. By overcoming this limitation, the bound provided in this paper offers a more practical and realistic understanding of generalization performance.
  - Additionally, the paper presents a method to compute this complexity for large-scale neural networks such as ViTs and GNNs, enhancing its practical applicability.
- Innovative Combination of Two Interesting Topological Metrics for Generalization Error Bound
  - The authors leverage the fact that the $\alpha$-weighted lifetime sums based on minimum spanning trees, a quantity from persistent homology, are related to a pseudo metric, and apply this to generalization error analysis.
  - They also focus on a topological quantity called Magnitude and its connection to metric spaces, successfully providing another generalization error bound.
  - These achievements are the result of effectively combining techniques from topological data analysis, which goes beyond mere application and demonstrates significant originality.
- Correlation with Generalization Performance Confirmed through Numerical Experiments on Large-Scale Neural Networks
  - The proposed topological complexity is computationally feasible even for large-scale neural networks. This is crucial for understanding the generalization performance of models like LLMs in the future.

**Weaknesses:**

- Concerns Regarding Assumptions
  - This bound only holds under bounded losses. Thus, as the upper bound B increases, the bound becomes vacuous and diverges under unbounded losses.
  - Additionally, the relationship between the upper bound B of the loss and the topological complexity is unclear, making the overall interpretation of the bound difficult. For instance, under large B, the correlation of the topological complexity might be negated, resulting in a bound that does not correlate well overall. Providing an intuitive discussion on this aspect would help clarify the significance of the bound.
  - I am not well-versed in the assumptions based on pseudo metrics, so I cannot assess the validity of the (1,L,\rho)-Lipschitz continuity assumption. However, as the authors mentioned, this assumption limits the applicability of the bound to certain pseudo metrics like the Euclidean distance or data-dependent pseudo metrics.

- Concerns Regarding Mutual Information in the Bound
  - Although the authors reference literature to assert that the mutual information in their proposed bound is tighter than the information-theoretic quantities in traditional bounds, they do not provide concrete methods for calculating or estimating this quantity.
  - This means that while some terms in the proposed generalization error bound are computable, the bound as a whole is not computationally evaluable. Thus, although the topological quantities used in the bound correlate with generalization performance, this does not guarantee the overall tightness of the bound. In practice, even if the topological complexity decreases, an increase in mutual information could render the bound vacuous.
  - The paper claims that the proposed bound is a uniform generalization bound. Therefore, it is necessary to discuss whether this bound achieves uniform convergence. If the mutual information term can be bounded by a constant, uniform convergence might be guaranteed, but this is not trivial and depends on the behavior of this quantity. Otherwise, the bound might fail to ensure uniform convergence.

- Challenges in Handling Hyperparameters and Lack of Detailed Discussion
  - If I understand correctly, the evaluation of the bound and topological complexity is closely related to the iteration step \tau at which the measurement begins and the scale value s in PMag, both of which need to be determined for the bound to hold. These settings carry the risk of arbitrariness in evaluating generalization error. However, the paper does not provide sufficient discussion on the correlation between these changes and generalization performance (incidentally, there is a notation conflict between Kendall’s coefficients and \tau here). For instance, providing more candidates for s and analyzing the sensitivity of the correlation degree to changes in its value, or verifying the variation in experimental results under multiple starting point candidates \tau, would allow for a minimum level of discussion.

- Lack of Comparison with Other Information-Theoretic Quantities or Metrics Strongly Correlated with Generalization
  - The numerical experiments presented in this study focus solely on topological complexity and compare the contributions of this research with existing studies in terms of correlation with the generalization gap. As a result, the relationship between topological quantities and generalization, and their standing in the context of other discussions, such as generalization based on, e.g., gradient variance (e.g., Jiang et al. (2019)) or mutual information (e.g., Russo and Zou (2016); Harutyunyan et al. (2021); Steinke et al. (2020)), remains unclear.
  - Thus, it remains uncertain how the proposed measure compares to other evaluation metrics in terms of superiority.

Citation:
- Jiang et al. (2019):  Jiang et al. Fantastic Generalization Measures and Where to Find Them. ICLR2020. https://openreview.net/forum?id=SJgIPJBFvH
- Russo and Zou (2016): D. Russo and J. Zou. Controlling bias in adaptive data analysis using information theory. AISTATS2016. https://proceedings.mlr.press/v51/russo16.html
- Harutyunyan et al. (2021): Harutyunyan et al. Information-theoretic generalization bounds for black-box learning algorithms. NeurIPS2021. https://arxiv.org/abs/2110.01584
- Steinke et al. (2020): Steinke et al. Reasoning About Generalization via Conditional Mutual Information. COLT2020. https://arxiv.org/abs/2001.09122

**Questions:**

The following questions primarily stem from the Weaknesses section. Please review these alongside the Weaknesses section and provide your comments. If the concerns and questions raised in the Weaknesses and Questions sections are appropriately addressed, I will consider increasing the score.

- **Discussion on the Validity of Assumptions**: Could you provide a more detailed discussion on the impact of the upper bound $B$ of the bounded loss and how it affects the bound value? Specifically, for cases like the bound in Theorem 3.4, where the topological complexity is multiplied by $B$, it would be helpful to understand to what $B$ could potentially negate the correlation with the topological distance, either theoretically or numerically. Additionally, could you elaborate on the validity of the $(1,L,\rho)$-Lipschitz continuity assumption?

- **Mutual Information in the Bound**: Please provide a more detailed discussion on the mutual information term appearing in the bound. Can this quantity be upper-bounded by a constant, thus ensuring uniform convergence? Or could this potentially be unbounded and thus become a fundamental limitation?

- **Sensitivity to Hyperparameter Settings ($s$ and $\tau$)**: How do the evaluation metrics and experimental results vary with the hyperparameter settings, such as $s$ and $\tau$? If feasible within the rebuttal period, it would be beneficial to discuss this sensitivity. If there is a valid reason for not addressing this, please explain.

- **Comparison with Gradient Variance and Other Mutual Information Terms**: Does the topological complexity in the proposed bound achieve better correlation with generalization compared to gradient variance and mutual information terms, which are known to correlate well with generalization performance? If the derived topological complexity demonstrates a stronger correlation, it could significantly contribute to future research directions. Providing theoretical and experimental evidence would be beneficial. If there is a valid reason for not making this comparison, please explain.

**Limitations:**

- This paper is a theoretical study aimed at providing a more realistic evaluation of generalization error. The data used in the experiments, such as MNIST, are open-source, which indicates that potential negative social impacts are appropriately controlled.
- While limitations are discussed in Section 6, there appear to be additional potential limitations as mentioned in the questions above. Addressing these would provide a more comprehensive understanding of the research's boundaries and enhance the validity of the study.

---

> ### Author Rebuttal · Authors · 2024-08-06
>
> We thank the reviewer for their insightful review. Below we address all the raised concerns. We hope that the reviewer reconsiders their score in the light of our comments.
>
> Before delving further, we would like to point out that our main focus is to obtain not just theoretical insights but also empirically meaningful quantities. We stress the importance of the latter, the extensive suite of evaluations proving that our topological quantities are meaningful and exhibit strong correlations. Our paper presents the first topological measures with an extensive set of experiments, obtained on practical models and complex data domains.
>
> We have now included additional discussion on all the points listed below.
>
> **Validity of the assumptions.**
> > Bounded loss
>
> Let us emphasize that the bounded loss assumption is realistic in several practical settings (eg, for the $0-1$ loss) and that it has been largely used in the literature on topological bounds [17,18,29].
>
> In Thm. 3.4, the scaling of $B$ will not affect the correlation. Indeed, if we multiply the loss by $c>0$, ie, $\ell':=c\ell$, and consider the data-dependent pseudometric $\rho_S$, then $E_1$ is also multiplied by $c$, hence the observed correlation is independent of the scaling factor. We now expand on these details in the paper.
>
> >  Lipschitz assumption \& applicability
>
> First the $(1,1,\rho_S)$-Lipschitz condition is always satisfied for the data-dependent pseudometric $\rho_S$. Indeed, by definition, $\Vert L_S(w) - L_S(w') \Vert = n \rho_S(w,w')$.
> The $(1,L,\rho)$-Lipschitz condition becomes non-trivial when $\rho$ is the Euclidean distance. In that case, it requires $\ell(w,z)$ to be Lipschitz, which is relatively standard.
>
> In our paper, we encapsulated both scenarios inside a single condition to identify the more general structure and pave the way for the use of other distances between models. Thus, our Lipschitz condition should not be seen as a limitation but rather a generalization.
>
> **Mutual Information.**
> The MI term measures the dependence between the data $S$ and a random set $\mathcal{W}$. Its presence is inherited from prior topology-based generalization studies, which face the same issues [17,18,8,11,69,29]. These bounds can be decomposed as the sum of a topological and an MI part, yet, in prior works the topological part suffered from major drawbacks, which is the main focus of our paper. We aim at improving the topological part of the bound.
> Obtaining a clearer understanding of the remaining part is an important direction for our future research.
>
> > No method to estimate the MI term
>
> We agree with the reviewer that the very complex structure of the MI term raises several difficulties. (1) we are not aware of existing techniques to evaluate MI between random sets and (2) the dimensionality of $\mathcal{W}$ (billions of parameters), could make a direct computation intractable.
> While these render the MI term uncomputable (as we mention in the limitations), our work focuses on the topological part.
> Our experiments show that these complexities are important and meaningful in addition to being amplified in the first part of the bound, as the dependence is explicit.
>
> Nevertheless, as acknowledged by the reviewer, our definition of MI is much simpler than the prior works making it intuitive to understand.
>
> > Can this ensure uniform convergence?
>
> We thank the reviewer for this very pertinent question.
> Let us first highlight that our use of the term "uniform" refers to the bound being uniform over the trajectory, i.e., we look at the worst-case error over the trajectory.
> In this context, our convergence is uniform.
>
> As mentioned above, convergence can be ensured with similar information-theoretic terms (which we could use in our work) for particular algorithms, like SGLD [18]. However, in the most general case (eg, a deterministic algorithm), it could lead to vacuous bounds. We acknowledged this lack of understanding in the limitations and we will add further comments.
>
> **Sensitivity to hyperparameters.**
> We thank the reviewer for suggesting this interesting sensitivity experiment.
> We performed an analysis of the sensitivity to $s$ for the ViT and CIFAR10 in Fig. 1.a (rebuttal pdf). We observe that the correlation is relatively stable near the two values of $s$ used in our study.
>
> Note that our choice of $\sqrt{n}$ is theoretically motivated (see lines 216 to 221), and the choice of small $s$ comes from evidence in prior work. We have added these sensitivity experiments in the paper.
>
> We introduced the parameter $\tau$ for two reasons: (1) to allow the use of pretrained weights and (2) to capture the geometric properties of the trajectory near a local minimum. Therefore, $\tau$ is chosen to ensure that the training is near convergence.
> We did not try to tune the parameter $\tau$ in any way.
>
> Thank you for catching the conflict of notations, we will replace $\tau$ by $t_0$.
>
> **Comparison with other terms.**
> As suggested by the reviewer, we conducted a small-scale experiment to compare our complexities with the gradient variance [31] in Fig. 1.b (rebuttal pdf). These preliminary results indicate a comparable correlation performance while slightly favoring our method.
> We have now included this experiment with gradient variance in the final version and will extend it to further settings.
>
> On the other hand, the numerical comparison with conditional mutual information (CMI) bounds requires a rigorous re-adaptation of our models and datasets to the CMI analysis, which is not doable within the rebuttal period, but we will consider to try this experiment for the final version.
> We also point out that our complexities are expected to be much less computationally demanding than CMI. Indeed, one can evaluate them based on a single training and no supersample of data is necessary.
>
> Finally, let us reiterate that, as opposed to most existing works, our measures are meant for the worst-case generalization error (see Section 4).

---

> > ### Comment · Reviewer_9Fve · 2024-08-10
> > **Aknowledgements**
> >
> > First and foremost, I would like to express my gratitude to the authors for their detailed feedback. I also apologize for my misunderstanding regarding the nature of the generalization error bounds presented in this paper, which are worst-case generalization bounds rather than algorithm-dependent bounds. I misunderstood this point in the section discussing mutual information.
> >
> > The authors have effectively addressed my concerns. In addition to this clarification, I find the analytical approach and the insights gained from the bounds to be particularly interesting. I now believe the paper makes a sufficient contribution to warrant acceptance at NeurIPS. Therefore, I have decided to raise my score.

---

> > > ### Author Response · Authors · 2024-08-12
> > >
> > > We sincerely thank the reviewer for considering our feedback and updating their score. We are glad that our responses could address their concerns. We are incorporating these into the revision.

---

### Official Review · Reviewer_xNox · 2024-07-12

**Soundness:** 3
**Presentation:** 3
**Contribution:** 4
**Rating:** 8
**Confidence:** 4

**Summary:**

The paper makes significant contributions by establishing new theoretical connections between generalization and topological complexity measures, specifically $\alpha$-weighted lifetime sums and positive magnitude. The authors introduce these novel measures and link them to generalization error using innovative proof techniques.
Experiments show that these measures correlate highly with generalization error across various architectures and datasets. The work offers simpler, less restrictive generalization bounds, removing the need for complex geometric assumptions. These flexible measures are adaptable to different domains, tasks, and architectures, providing practical, theoretically justified tools for understanding and predicting generalization in modern deep neural networks.

**Strengths:**

The authors establishes new theoretical connections between generalization and topological complexity measures, specifically focusing on $\alpha$-weighted lifetime sums and positive magnitude. Also, introduces and elaborates on new topological complexity measures, such as the positive magnitude, which is a modified version of the magnitude measure. These measures are linked to generalization error using new proof techniques.
The paper respects the discrete-time nature of training trajectories and investigates topological quantities suitable for practical topological data analysis tools, which leads to computationally efficient measures. It proposes generalization bounds that are simpler and less restrictive compared to existing methods, removing the need for complex geometric assumptions and making them more practical.

**Weaknesses:**

Maybe I get this wrong but in line 113, is $Y\subset X$ or $Y\subset A$?

**Questions:**

1.- Have you explore the situation of use compact metric spaces?

2.- Have you explore the situation of change the metric to a non-euclidean one?

3.- In line 182, what do you mean by $\mu_z^{\otimes n}$?

**Limitations:**

Lack of understanding of IT terms.

---

> ### Author Rebuttal · Authors · 2024-08-06
>
> We thank the reviewer for their interesting comments.
>
> > Maybe I get this wrong but in line 113, is $Y\subset X$ or $Y\subset A$?
>
> We thank the reviewer for pointing out this minor typo in the definition of the PH dimension. Indeed, we have replaced $Y\subset X$ by $Y\subset A$ in the final version of the paper.
>
> > 1.- Have you explore the situation of use compact metric spaces?
>
> We thank the reviewer for this interesting question/comment. Since we are interested in discrete optimizers, the most pertinent setting is that of discrete (even finite in practice) trajectories. However, it is interesting to ask whether our theory extends to random compact sets. As we hinted in Remark B.15 in the appendix, Theorem 3.5 (for positive magnitude) could be extended to such a compact setting. On the other hand, the extension of Theorem 3.4, involving $\alpha$-weighted lifetime sums, might be more complicated as the definition of $E_\alpha$ makes an explicit use of the finiteness of the set. However, we believe the main ideas of Theorem 3.4 could still be used to obtain interesting bounds in the compact setting and leave this question for future work. We have now added a further remark about it in the revision.
>
> > 2.- Have you explore the situation of change the metric to a non-euclidean one?
>
> As we explain in Section 3.1 (about the mathematical setup), our framework encompasses a wide range of metrics and pseudometrics, as long as they satisfy the condition given by Definition 3.1. In particular, an important part of our experiments uses the so-called data-dependent pseudometrics on $\mathbb{R}^d$, which we denote by $\rho_S^{(p)}$ in the paper. These pseudometrics are typically non-Euclidean on $\mathbb{R}^d$. While we use these non-Euclidean metrics, one can see in Example 3.2 that $\rho_S^{(p)}$ results from the Euclidean distance on $\mathbb{R}^n$ applied on the embedding $L_S(w)$ of $w$. It could be interesting to explore other non-Euclidean possibilities, which we leave for future work. For instance, if the loss is supported on a Riemannian manifold, then the associated metric could serve as a non-Euclidean metric to compute the topological complexities. One of the goals of our works is to leave the door open to such possibilities.
>
> > In line 182, what do you mean by $\mu_z^{\otimes n}$?
>
> First, as defined in the introduction, the notation $\mu_z$ corresponds to the unknown data distribution. We used the notation $\mu_z^{\otimes n}$ as a shortcut for the product measure $\mu_z \otimes \dots \otimes \mu_z$. Therefore, the notation $(z_1,\dots,z_n) \sim \mu_z^{\otimes n}$ means that the data points $z_i$ are sampled i.i.d. from the distribution $\mu_z$. We now made these notations more precise in the paper.

---

> > ### Comment · Reviewer_xNox · 2024-08-13
> >
> > Thanks for your response!

---

### Author Rebuttal · Authors · 2024-08-06

The rebuttal pdf includes figures that are mentioned in some of our answers to the reviewers.

Fig. 1.a analyses the sensitivity to $s$ of the correlation between the generalization error and $\mathrm{PMag}(s\mathcal{W})$.
Fig. 1.b is a preliminary result regarding the comparison of our topological complexities with gradient variance.
Fig. 2.a is a visual representation of some results of Table 1, it completes Fig. 1.c. in the paper.
Fig. 2.b is a graphical representation of (part of) the proof of Theorem 3.4.

---

### Decision · Program_Chairs · 2024-09-25

**Decision:**

Accept (poster)

**Comment:**

First I want to thank all of the reviewers for the work they put in and for the authors on engaging with the reviewers and correcting several things in the paper following the discussion.

Given the review and the quality of work, i am more than happy to recommend this paper for acceptence to NeurIPS.

Best wishes,

    AC